# Biological and Chemical Diversity of Marine Sponge-Derived Microorganisms over the Last Two Decades from 1998 to 2017

**DOI:** 10.3390/molecules25040853

**Published:** 2020-02-14

**Authors:** Mei-Mei Cheng, Xu-Li Tang, Yan-Ting Sun, Dong-Yang Song, Yu-Jing Cheng, Hui Liu, Ping-Lin Li, Guo-Qiang Li

**Affiliations:** 1Key Laboratory of Marine Drugs, Chinese Ministry of Education, School of Medicine and Pharmacy, Ocean University of China, Yushan Road 5, Qingdao 266003, China; alisa_0701@126.com (M.-M.C.); syt@stu.ouc.edu.cn (Y.-T.S.); 21170831098@stu.ouc.edu.cn (D.-Y.S.); c17860731575@163.com (Y.-J.C.); LH1498740241@163.com (H.L.); 2Laboratory of Marine Drugs and Biological Products, National Laboratory for Marine Science and Technology, Qingdao 266235, China; 3College of Chemistry and Chemical Engineering, Ocean University of China, Songling Road 238, Qingdao 266100, China; tangxuli@ouc.edu.cn

**Keywords:** marine sponges, sponge-derived microorganisms, natural products, relationship

## Abstract

Marine sponges are well known as rich sources of biologically natural products. Growing evidence indicates that sponges harbor a wealth of microorganisms in their bodies, which are likely to be the true producers of bioactive secondary metabolites. In order to promote the study of natural product chemistry and explore the relationship between microorganisms and their sponge hosts, in this review, we give a comprehensive overview of the structures, sources, and activities of the 774 new marine natural products from sponge-derived microorganisms described over the last two decades from 1998 to 2017.

## 1. Introduction

The term “symbiosis” was first defined by the German mycologist Heinrich Anton de Bary in 1879 as “the living together of unlike organisms” [1]. Symbiosis is an intimate and long-term biological interaction between two different biological organisms, whether reciprocal, symbiotic, or parasitic. It is abundant and widespread in the sea for symbioses between microorganisms and marine organisms. Most marine animals and plants such as sponges, corals, sea squirts, worms, and algae host contain diverse and abundant symbiotic microorganisms. Among them, sponge is the most primitive type of metazoan, which has been used as an important source of marine active compounds. There are many types of sponges (between 10,000 and 15,000 species), accounting for 6.7% of all marine animal species. At present, detected sponges can be divided into four classes: Calcarea (about 400 species), Hexactinellida (about 600 species), Demospongiae (about 4000 species), and Homoscleromorpha (about 15 species). Sponges are multicellular filter feeders, in which a layer of flagellum cells arranged in the body cavity of the sponge provides nutrients and oxygen to the spongy body through the generated water flow. Its unique pore structure makes it an excellent host for many marine microorganisms, which account for a large amount of sponge biomass. Some studies have shown that the true source of secondary metabolites in sponges may be their symbiotic microbes. Therefore, a comprehensive review of the existing studies of the sponge-associated microbes is expected to reveal the potential chemical association between symbiotic microbes and their hosts.

In recent years, research on co-existing microorganisms derived from marine animals and plants has received increasing attention, especially sponge symbiotic microorganisms. Several reviews discussing “sponge-derived microorganisms” have been published, and the topics include genetics [2], ecology [3,4], and chemical diversity [5,6,7]. In 2014, Cristian et al. [8] reviewed the microbial community and its biological activity of the Irciniidae sponges, and discussed the relationship between the host and its co-existing microorganisms in combination with genetics and ecology. In 2017, Fehmida Bibi et al. [9] reviewed the latest studies on active secondary metabolites produced by sponge-derived commensal bacteria, suggesting that sponge symbiotic bacteria are one of the important sources of new drugs. Therefore, it is essential that a systematic overview for the chemical diversity especially for the new compounds of sponge-derived microorganisms were summarized.

In 1988, Stierle et al. in Montana State University in the United States obtained three diketopiperazine alkaloids—cyclo-(Ala-Pro), cyclo-(Val-Pro), and cyclo-(Leu-Pro)—from the sponge *Tedania ignis*-derived bacteria *Micrococcus* sp., which is the first natural product molecule obtained from the sponge symbiotic bacteria [10]. In the following 10 years, there have been several reports on secondary metabolites of sponge-derived microorganisms, most of which are about fungi, bacteria, and actinomycetes. The rapid development of this research is mainly after 1998. This review comprehensively focuses on the chemical diversity and biological activity of the symbiotic microorganisms derived from different sponge species, covering 774 new compounds totally from 1998 to 2017.

## 2. Sponges and Derived Microbes’ Chemical Diversity

### 2.1. Class Calcarea

Two new compounds, designated JBIR 74–75 (**1**–**2**) (Figure 1), were isolated from the fungus *Aspergillus* sp. fS14, which was isolated from the unidentified marine sponge (class, Calcarea) collected off Ishigaki Island, Okinawa, Japan. Neither of the two compounds showed cytotoxic activity against several cancer cell lines (IC_50_ > 100 μM), nor did they show antimicrobial activity against *Candida albicans*, *Micrococcus luteus*, and *Escherichia coli* [11].

#### 2.1.1. Order Baerida

##### Family Baeriidae

From a bacterium *Microbulbifer* sp. strain L4-n2 associated with the sponge *Leuconia nivea,* eight new natural parabens **3**–**10** (Figure 1) were isolated. Compounds **3, 4, 5,** and **9** appeared for the first time as natural products. Compound **5** exhibited the greatest efficiency against *S. aureus* with minimal inhibitory concentration (MIC) values of 2.8–5.6 μM [12].

#### 2.1.2. Order Clathrinida

##### Family Clathrinidae

Fractionation of the strain *Micromonospora* sp. (strain L-31-CLCO-002) from the sponge *Clathrina coriacea* collected off Sanish Fuerteventura Island afforded two new indolocarbazole alkaloids **11**–**12** (Figure 1), and it exhibited stronger cytotoxic activities against the P388D1, A549, HT-29, and SK-MEL-28 cell lines with IC_50_ values of 2–40 nM [13].

### 2.2. Class Demospongiae

Chemical investigation of a fungus *Acremonium* sp. SpF080624G1f 01 from a Demospongiae sponge (Ishigaki Island, Okinawa, Japan) led to the isolation of two novel glycosyl benzendiols, JBIR 37–38 (**13**–**14**) (Figure 1) [14]. From the fungus *Penicillium citrinum* SpI080624G1f 01, a new compound termed JBIR-59 (**15**) and new sorbicillinoid derivative designated as JBIR-124 (**16**) (Figure 1) were isolated. Compound **15** showed reduced L-glutamate toxicity in N18-RE-105 cells with EC_50_ values of 71 μM, and compound **16** had DPPH radical scavenging activity (IC_50_, 30 μM) [15,16]. A new salicylamide derivative termed JBIR-58 (**17**) [17] and two new pyrazinones JBIR 56–57 (**18**–**19**) (Figure 1) [18] were isolated from actinomyces *Streptomyces* SpD081030ME-02 and SpD081030SC-03, which originated in Demospongiae sponges, respectively. Compound **17** exhibited a weak cytotoxic effect on human cervical carcinoma (HeLa) cells (IC_50_, 28 μM).

#### 2.2.1. Order Agelasida

##### Family Agelasidae

From a strain of fungus *Trichoderma* sp. derived from the Caribbean sponge *Agelas dispar*, four novel sorbicillinoid polyketide derivatives (**20**–**23**) (Figure 1) were isolated; unfortunately, the tested compounds **20**–**23** showed no bioactivities [19]. A new pyranone derivative, trichopyrone (**24**) (Figure 1), was isolated from the fungus *Trichoderma viride*, which was originally separated from the Caribbean sponge *Agelas dispar* and shown to have very weak or no effects in a series of bioassays (radical scavenging, antioxidant, antimicrobial, inhibition of HIV-1 RT) [20]. Two new highly oxygenated hexacyclic cyclopiazonic acid (CPA), speradines B–C (**25**–**26**), together with one new related tetracyclic oxindole alkaloid, speradine D (**27**) (Figure 1), were produced in the fungal strain *Aspergillus flavus* MXH-X104 associated with the marine sponge *Agelas aff. nemoechinata* collected from the Xisha Islands of China. However, their bioassay was disappointing [21]. A sponge-associated actinomycetes *Streptomyces* sp. SBT345 from the Mediterranean sponge *Agelas oroides* provided a new cytotoxic phenoxazin analogue strepoxazine A (**28**) and a new antioxidant and antichlamydial quinolone ageloline A (**29**) (Figure 1). Compounds **28** and **29** exhibited cytotoxic activity against leukaemia cells HL-60 cells with an IC_50_ value of 8 μg/mL and inhibitory activity toward the formation and growth of Chlamydia trachomatis inclusion in a dose-dependent manner with an IC_50_ value of 9.54 ± 0.36 μM [22,23], respectively.

#### 2.2.2. Order Axinellida

From a strain of the fungus *Trichoderma atroviride* (NF16) derived from an Axinellid sponge collected from the Mediterranean Sea, eight new linear peptaibols (**30**–**37**) (Table 1) were isolated and found to have antimicrobial activity against environmental bacteria isolated from the Mediterranean coast of Israel [24].

##### Family Axinellidae

Two unique steroids possessing the bicyclo [4.4.1] A/B ring system, isocyclocitrinol A (**38**) and 22-acetylisocyclocitrinol A (**39**) (Figure 2), were isolated from *Penicillium citrinum* colonizing in a sponge (*Axinella* sp.) collected in Papua New Guinea, showing weak antibacterial activity against *Staphylococcus epidermidis* and *Enterococcus durans* [25]. Seven new compounds, bicoumanigrin (**40**), aspernigrins A–B (**41**–**42**), and pyranonigrins A–D (**43**–**46**) (Figure 2) were obtained from the strain of *Aspergillus nige*, which is a symbiont on the Mediterranean sponge *Axinella damicornis*. Compound **40** showed moderate cytotoxicity against human cancer cell lines, and compound **42** displayed a strong neuroprotective effect against stimulation caused by L-glutamic acid or quisqualic acid [26]. Two new congeners communesins C–D (**47**–**48**) (Figure 2) were isolated from the fungus *Penicillium* sp., which was derived with the Mediterranean sponge *Axinella verrucose*. Both exhibited moderate antiproliferative activity against the different leukemia cell lines [27].

Highly *N*-Methylated Linear Peptides, RHM 1–4 (**49**–**50, 53**–**54**), as well as linear pentadecapeptides efrapeptin Eα (**51**) and efrapeptin H (**52**) (Figure 3), were produced by fungus *Acremonium* sp. (UCSC coll. no. 021172 cKZ), which is associated with the marine sponge *Teichaxinella* sp. (Papua New Guinea). Compound **51** displayed very strong cytotoxic activity with an IC_50_ value of 1.3 nM against H125 cells [28,29]. Four new tetromycin derivatives, tetromycins 1–4 (**55**–**58**) (Figure 3), were produced by a strain actinomycete of *Streptomyces axinellae* Pol001T originally derived from the marine sponge *Axinella polypoides* (Banyuls-sur-Mer, France). All four compounds showed antiparasitic activities against *Trypanosoma brucei* and the time-dependent inhibition of cathepsin L-like proteases with Ki values in the low micromolar range [30]. Compound **59** (Figure 3) was isolated from a bacterium *Pseudomonas fluorescens* 4.9.3 cultivated from the sponge *Axinella damicornis* (Turkey) [31].

Two new oxaphenalenone dimers, talaromycesone A–B (**60**–**61**), and a new isopentenyl xanthenone, talaroxanthenone (**62**) (Figure 4), were found in the fungus *Talaromyces* sp. strain LF458 associated with the sponge *Axinella verrucosa* (Mediterranean Sea, Italy). Compound **60** exhibited potent antibacterial activity with an IC_50_ value of 3.70 μM against human pathogenic *Staphylococcus* strains and an inhibitory activity of phosphodiesterase PDE-4B2 (IC_50_ 7.25 μM). In addition, compound **60** and compound **62** also displayed potent acetylcholinesterase inhibitory activities with an IC_50_ of 7.49 and 1.61 μM, respectively [32]. The fungus *Talaromyces rugulosus* from the Mediterranean sponge *Axinella cannabina* (Turkey) provided three butenolides (**63**–**65**), seven (3*S*)-resorcylide derivatives (**66**–**72**), two butenolide-resorcylide conjugates (**73**–**74**), and two dihydroisocoumarins (**75**–**76**) (Figure 4). Compounds **73**–**74** exhibited potent cytotoxicity against the L5178Y murine lymphoma cell line with IC_50_ values of 3.9 and 1.3 µM, respectively [33]. The chemical investigation of the mycelia and culture filtrate of *Penicillium* sp. DRF2 from the sponge *Dragmacidon reticulatum* (Brazil) yielded seven novel curvularins that belong to macrocyclic polyketides 12-Keto-10,11-dehydrocurvularin **77** and *cis*-10,11-Epoxycurvularin **78** and the sulfur-containing curvularin derivatives **79**–**83** (Figure 4) [34]. A study on fungus *Hypocrea koningii* PF04 derived from the sponge *Phakellia fusca* (Yongxing Island, China) afforded two new furan derivatives, hypofurans A–B (**84**–**85**), and three new cyclopentenone derivatives, hypocrenones A–C (**86**–**88**) (Figure 4). Compound **84** showed modest antibacterial activity against *Staphylococcus aureus* (MIC, 32 μg/mL) and moderate (2,2-diphenyl-1-picrylhydrazyl) DPPH radical scavenging capacity (IC_50_, 27.4 µg/mL) [35].

From the fungus *Arthrinium arundinis* ZSDS1-F3 cultured from a *Phakellia fusca* marine sponge (Xisha Islands, China), four new cytochalasins, arthriniumnins A–D (**89**–**92**), a new natural product, ketocytochalasin (**93**), three new 4-hydroxy-2-pyridone alkaloids, arthpyrones A–C (**94**–**96**), and a new natural product, phenethyl 5-hydroxy-4-oxohexanoate (**97**) (Figure 5), were isolated. Compounds **94** and **96** displayed significant cytotoxicities against the selected tumor cell lines with IC_50_ values ranging from 0.24 to 45 μM. Furthermore, compound **96** displayed acetylcholinesterase (AchE) inhibitory activity with an IC_50_ value of 0.81 μM [36,37,38]. Five unusual 14-membered macrolides, gliomasolides A–E (**98**–**102**) (Figure 5), were produced in fungus *Gliomastix* sp. ZSDS1-F7-2 from the marine sponge *Phakellia fusca Thiele* (South China Sea). Only compound **98** exhibited moderate inhibitory effect with an IC_50_ value of 10.1 μM against the growth of HeLa cell lines [39]. Sponge-derived fungus *Hypocrea koningii* PF04 (from the sponge *Phakellia fusca* collected off Yongxing Island, China) resulted in one new tyrosol derivative, hypocrol A (**103**) (Figure 5) with weak antibacterial activity [40]. Three new cyclohexadepsipeptides, oryzamides A–C (**104**–**106**) (Figure 5) originated in fungus *Nigrospora oryzae* PF18 from the marine sponge *Phakellia fusca* (Yongxing Island, China) [41].

Three racemates of diorcinol monoethers, (±)-versiorcinols A–C (**107**–**109**), and 12 previously undescribed polyketide derivatives, heterocornols A–L (**110**–**121**) (Figure 6), were isolated from the fungus *Aspergillus versicolor* 16F-11 and *Pestalotiopsis heterocornis* XWS03F09 cultivated from the sponge *Phakellia fusca* (the Xisha Islands, China). The bioassays of all the compounds toward selected models were disappointing [42,43]. An examination of the fungus *Pestalotiopsis heterocornis* originated from the sponge *Phakellia fusca* (Xisha Islands of China) yielded two isocoumarins, pestaloisocoumarins A–B (**122**–**123**), one sesquiterpenoid degradation, isopolisin B (**124**), and one furan derivative, pestalotiol A (**125**) (Figure 6) [44].

##### Family Raspailiidae

Two new prenylated polyketides, epoxyphomalin A–B (**126**–**127**) (Figure 7), produced in fungus *Phoma* sp. derived from sponge *Ectyplasia ferox* (Caribbean Sea, Dominica) showed superior cytotoxicity at nanomolar concentrations toward 36 human tumor cell lines [45].

#### 2.2.3. Order Biemnida

##### Family Rhabderemiidae

A study on the fungus *Aspergillus similanensis* KUFA 0013 isolated from the sponge *Rhabderemia* sp. (Thailand) resulted in two new isocoumarin derivatives (**128**–**129**), a new chevalone derivative chevalone E (**130**), and a new natural product pyripyropene S (**131**) (Figure 7). Compound **130** was found to have synergism with the antibiotic oxacillin against methicillin-resistant *Staphylococcus aureus* (MRSA) [46].

#### 2.2.4. Order Chondrillida

##### Family Chondrillidae

Five new analogs of thiocoraline (**132**–**136**) (Figure 7) were isolated from marine *Verrucosispora* sp. isolated from sponge *Chondrilla caribensis f. caribensis* (the Florida Keys). Compounds **13****2**, **135**, and **136** demonstrated significant cytotoxicity against the A549 human cancer cell line with IC_50_ values of 0.13, 2.86, and 1.26 μM, respectively [47]. A new meroditerpene, sartorypyrone C (**137**) (Figure 7), was produced in the marine sponge-associated fungus *Neosartorya paulistensis* strain KUFC 7897 from sponge *Chondrilla australiensis* (Thailand) [48]. A chemical examination of the marine fungus *Aspergillus* sp. isolated from the sponge *Chondrilla nucula* (Turkey) yielded two new phenolic bisabolane sesquiterpenes, asperchondols A–B (**138**–**139**) (Figure 7). The antibacterial activities of compounds **138** and **139** were evaluated against eight human pathogenic bacteria [49].

#### 2.2.5. Order Chondrosiida

##### Family Chondrosiidae

Seven new polyketides with linear pentaene fragments (**140**–**146**) (Figure 8) were characterized from the fungus *Penicillium rugulosum* associated with the sponge *Chondrosia reniformis* (Elba, Italy) [50].

#### 2.2.6. Order Clionaida

##### Family Clionaidae

A new compound, namely butylrolactone-VI (**147**) (Figure 9), was metabolized by the fungus *Aspergillus* sp. (2P-22) associated with the marine sponge *Cliona chilensis* (collected in the Pacific Sea, Chile), and its antibacterial and antitumor activities were determined [51]. A unique *O*-glycosylated disubstituted Microluside A (**148**) (Figure 9) was isolated from *Micrococcus* sp. EG45 associated with the Red Sea sponge *Spheciospongia vagabunda*, which exhibited antibacterial potential against *Enterococcus faecalis* JH212 and *Staphylococcus aureus* NCTC 8325 with MIC values of 10 and 13 μM, respectively [52]. From the actinomycete *Actinokineospora* sp. EG49 cultivated in Red Sea sponge *Spheciospongia vagabunda*, eight new benzanthraquinone-like *O*-glycosylated angucyclines, actinosporin A–H (**149**–**156**) (Figure 9), were isolated. Compounds **151** and **152** showed antioxidant potential using two different approaches including cell-free and cell-based assays [53,54,55].

#### 2.2.7. Order Dictyoceratida

Three new benzolactone metabolites, chrysoarticulins A–C (**157**–**159**) (Figure 10), were produced in the fungus *Chrysosporium articulatum* obtained from an unidentified dictyoceratid sponge (Korea). The tested bioactivity for them was unfavorable [56].

##### Family Dysideidae

A novel tripeptide containing a β-amino acid (**160**) (Figure 10) was metabolized by a bacterium *Pseudomonas alteromonas* from the marine sponge *Dysidea fragilis* (collected in the Black Sea, Bulgarian) [57]. A thiazole alkaloid, neobacillamide A (**161**) (Figure 10), was isolated from the bacterium *Bacillus vallismortis* C89 purified the South China Sea sponge *Dysidea avara* [58]. Four new cyclic lipopeptides based on the carbon nuclei cyclo-(AFA-Ser-Gln-Asn-Tyr-Asn-Ser-Thr), named cyclodysidins A–D (**162**–**165**) (Figure 10), were purified from the bacterium of *Streptomyces* strain RV15 associated with the sponge *Dysidea tupha* (collected from Rovinj, Croatia) [59].

##### Family Irciniidae

Cyclotetrapeptide (**166**) (Figure 11) was yielded in bacterium *Pseudomonas* sp. from the sponge *Ircinia muscarum* (collected in the gulf of Naples) [60]. Two first reported sorbicillin-derived alkaloids, sorbicillactones A–B (**167**–**168**), and one orbicillin derivative (**169**) as well as three novel sorbicillinol derivatives of mixed origin, sorbifuranones A–C (**170**–**172**) (Figure 11), were characterized from *Penicillium chrysogenum* associated with the Mediterranean sponge *Ircinia fasciculata* (Italy). Compound **167** exhibited a high anti-HIV activity in the concentration range from 0.3 to 3.0 μg/mL, selective anti-leukemic activities, and furthermore, antiviral and neuroprotective properties [61,62]. Diketopiperazine alkaloid, an amauromine composed of two modified tryptophan units, which are cyclized forming a central diketopiperazine ring (**173**), a quinolinone derivitive, methyl-penicinoline (**174**), and one triterpene glycoside auxarthonoside (**175**) (Figure 11) were produced by the fungus *Auxarthron reticulatum* derived from the sponge *Ircinia variabilis* (the Island of Malta). Compound **173** dispalyed high affinity and selectivity for cannabinoid CB1 receptors [63,64]. The fungus *Aspergillus tubingensis* (strain OY907) isolated from the Mediterranean sponge *Ircinia variabilis* yielded a novel anhydride metabolite, tubingenoic anhydride A (**176**) (Figure 11) [65]. From the fungus *Penicillium* sp. (strain 101) derived from another Mediterranean sponge *Ircinia oros* (Turkey), a new fusarielin analogue (**177**) (Figure 11) was purified [66].

The fungus *Aspergillus insuetus* (OY-207) originated from the Mediterranean sponge *Psammocinia* sp. (collected off-shore of Sdot-Yam, Israel), and it afforded three novel meroterpenoids insuetolides A–C (**178**–**180**) and one new drimane sesquiterpene (**181**) (Figure 11). Compound **178** exhibited antifungal activity toward *Neurospora crassa* [67]. Meroterpenoids spiroarthrinols A–B (**182**–**183**) (Figure 11) were found in fungus *Arthrinium* sp. associated with the marine sponge *Sarcotragus muscarum* (Turkey) [68].

##### Family Thorectidae

Six new polyketides, engyodontochone A–F (**184**–**189**) (Figure 12), were purified in fungus *Engyodontium album* strain LF069 originally separated from the sponge tissue of *Cacospinga scalaris* sampled at the Limski Fjord, Croatia. Compounds **186**–**189** represented the first example of a 23, a 28 seco-beticolin carbon skeleton, and compounds **184**–**185** exhibited inhibitory activity that was 10-fold stronger than chloramphenicol against methicillin-resistant *Staphylococcus aureus* [69]. The strain *Pseudoalteromonas maricaloris* KMM 636^T^ derived from the Australian sponge *Fascaplysinopsis reticulata* collected at the Great Barrier Reef was found to produce an inseparable mixture of two brominated yellow main pigments, bromoalterochromide A and A′ (**190**) (Figure 12) in a ratio of 3:1. They showed cytotoxic effects on developing eggs of the sea urchin *Strongylocentrotus intermedius* (MIC, 40 μg/mL) [70].

The cultures of *Aspergillus niger* separated from a Caribbean sponge, *Hyrtios proteus*, collected in the Dry Tortugas National Park, Florida, resulted in the isolation of a tetrahydrofuran-type derivative asperic acid (**191**) (Figure 13) [71]. The fungus *Emericellopsis minima* derived from the marine sponge *Hyrtios erecta* (Thailand) afforded a new bridged cyclic sesquiterpene (**192**) (Figure 13) [72]. A new alkaloid with an unprecedented carbon skeleton, penicillivinacine (**193**) (Figure 13), was produced by the fungus *Penicillium vinaceum*, which is associated with the marine sponge *Hyrtios erectus* (collected from Yanbu, Saudi Arabia). Compound **193** exhibited higher antimigratory activity than the positive control with an IC_50_ value of 18.4 μM against the human breast cancer cell line MDA-MB-231 [73]. Examination of the fungus *Trichoderma harzinum* HMS-15-3 derived from the sponge *Petrospongia nigra* collected from South China Sea provided four pairs of new linear C_13_ lipid enantiomers with polyene and O-dinol structure, namely harzianumols A–H (**194**–**201**) (Figure 13). Their antihyperlipidemic effects in HepG2 cells were evaluated [74].

Six new aromadendrane-type sesquiterpenoids, scedogiines A–F (**202**–**207**), and a new polyketide, scedogiine G (**208**) (Figure 13), were produced by the marine-derived fungus *Scedosporium dehoogii* F41-4, which is a symbiont on the sponge *Phyllospongia foliascens* collected from Hainan Sanya, China. Two new pyripyropenes (**209**–**210**) (Figure 13) were isolated from the other fungus *Fusarium lateritium* 2016F18-1, which is associated with the same sponge [75,76].

#### 2.2.8. Order Haplosclerida

##### Family Callyspongiidae

Two novel spiciferone derivatives, spiciferol A (**211**) and a monocyclic butoxyl derivative (**212**) (Figure 14), were isolated from the fungus *Drechslera hawaiiensis*; it is associated with the marine sponge *Callyspongia aerizusa*, which is collected from the Mengangan Island, Indonesia [77]. Two new macrolide metabolites, pandangolide 3–4 (**213**–**214**), and the new phthalide herbaric acid (**215**) (Figure 14) were isolated from a fungal strain *Cladosporium herbarum*, which is associated with the marine sponge *Callyspongia aerizusa* (collected in Indonesia). However, their bioassays were disappointing [78,79]. Two new antibacterial phenazines (**216**–**217**) (Figure 14) were isolated from the bacteria *Brevibacterium* sp. KMD 003 associated with a marine purple vase sponge of the genus *Callyspongia*, which was collected in Korea. Compounds **216** and **217** showed antibacterial activities against *Enterococcus hirae* and *Micrococcus luteus* (MIC, 5 μM) [80]. The investigation of a fungus *Stachylidium* sp. from the sponge *Callyspongia* sp. cf. *C. flammea* (collected in Sydney, Australia) yielded three new phthalide derivatives, marilones A–C (**218**–**220**), four new, putatively tyrosine-derived and O-prenylated natural products, stachylines A–D (**221**–**224**), and four novel phthalimidine derivatives marilines: A1–A2 (**225**–**226**), B–C (**227**–**228**) (Figure 14). Compound **218** was found to have antiplasmodial activity against *Plasmodium berghei* liver stages with an IC_50_ of 12.1 μM. Compound **219** showed selective antagonistic activity toward the serotonin receptor 5-HT_2B_ with a K_i_ value of 7.7 μM. Both compounds **225** and **226** inhibited human leukocyte elastase (HLE) with an IC_50_ value of 0.86 μΜ [81,82,83].

Seven novel phthalide-related compounds, cyclomarinone (**229**), maristachones A–E (**230**–**234**), marilactone (**235**), and two new *N*-methylated tetrapeptides with amino acid 3-(3-furyl)-alanine, namely endolide A–B (**236**–**237**) (Figure 14), were isolated from the fungus *Stachylidium* sp., which was isolated from the sponge *Callyspongia* sp. cf. *C. flammea* (Australia). In their bioassays, compounds **229**–**235** showed no significant biological activities. Endolide A (**236**) showed affinity to the vasopressin receptor 1A with a K_i_ of 7.04 μM, and endolide B (**237**) was selective toward the serotonin receptor 5HT_2b_ with a K_i_ of 0.77 μM [84,85]. A new pyronepolyene C-glucoside (**238**) was from fungus *Epicoccum* sp. JJY40, which was isolated from the sponge *Callyspongia* sp. collected in Hainan Province, China. Compound **238** (Figure 14) showed weak NF-κB (Proteins that can regulate gene expression) inhibitory (IC_50_ 40.0 μM) and significant inhibitory effects in the cytopathic effect (CPE) inhibition assay with IC_50_ value of 91.5 μM (ribavirin as a positive control, IC_50_ 114.8 μM) [86].

Chemical examination of a marine fungus *Alternaria* sp. JJY-32, isolated from a sponge *Callyspongia* sp. (Hainan Island, China), yielded 13 meroterpenoids (**239**–**251**) (Figure 15). The NF-κB inhibitory activities of **239**–**251** were tested, and all of them, except **244** and **245** (IC_50_ > 100 μM), showed moderate to weak activities with IC_50_ values ranging from 39 to 85 μM in RAW264.7 cells [87].

The marine-derived fungus *Dichotomomyces cejpii* from the sponge *Callyspongia* sp. cf. *C. flammea* (Bear Island, Australia) afforded two new compounds, emindole SB betamannoside (**252**) and 27-*O*-methylasporyzin C (**253**), three new steroids (**254**–**256**), and a new gliotoxin derivative, 6-acetylmonodethiogliotoxin (**257**) (Figure 15). Compounds **252** and **253** may serve as lead structures for the development of GPR18- and CB receptor-blocking drugs. Compound **255** was found to be capable of preventing the enhanced production of amyloid β-42 in Aftin-5 treated cells in an Alzheimer’s disease cellular assay. Compound **257** dose-dependently down-regulated TNFα (Tumor Necrosis Factor) -induced NF-κB (Proteins that can regulate gene expression) activity in human chronic myeloid leukemia cells with an IC_50_ of 38.5 ± 1.2 µM [88,89,90]. The chemical study of a marine-derived fungus *Stachylidium* sp. 293 K04 from the sponge *Callyspongia* sp. cf. *C. flammea* (Bare Island, Australia) afforded two new tetrapeptide analogues, endolides C–D (**258**–**259**) (Figure 15) [91]. The marine sponge-derived fungus, *Aspergillus* sp. SCSIO XWS02F40, derived from a sponge *Callyspongia* sp. (Guangdong Province, China), afforded two new asteltoxins named asteltoxin E–F (**260**–**261**) and a new chromone (**262**) (Figure 15). Compound **260** showed significant activity against H3N2 and exhibited inhibitory activity against H1N1 with the prominent IC_50_ values of 6.2 ± 0.08 and 3.5 ± 1.3 µM, respectively. Compound **261** showed significant activity against H3N2 with an IC_50_ value of 8.9 ± 0.3 µM [92].

##### Family Chalinidae

One new vertinoid polyketide (**263**) (Figure 16) was characterized from a marine-derived fungus *Trichoderma longibrachiatum* associated with a sponge *Haliclona* sp. collected from Sulawesi, Indonesia [93]. Three new cyclic sesquiterpenes, hirsutanols A–C (**264**–**265**) and ent-gloeosteretriol (**266**) (Figure 16), were metabolized by an unidentified fungus separated from an Indo-Pacific sponge *Haliclona* sp. Compounds 264 and 266 were antimicrobial toward *Bacillus subtilis* [94]. Chemical investigation of the marine-derived fungus *Emericella variecolor* isolated from the marine sponge *Haliclona valliculata* (Elba/Italy) led to the isolation of two new natural products, evariquinone and isoemericellin (**267**–**268**) (Figure 16). Compound **267** showed antiproliferative activity toward KB and NCI-H460 cells at a concentration of 3.16 μg/mL [95]. Two new macrolactams, cebulactams A1–A2 (**269**–**270**) (Figure 16), were metabolized by the genus *Saccharopolyspora cebuensis*-type strain SPE 10-1 associated with the sponge *Haliclona* sp. collected offshore Cebu, Philippines [96].

From a strain of *Streptomyces* sp. NBRC 105896 derived from a marine sponge *Haliclona* sp. (collected from Chiba Prefecture, Japan), JBIR-31, a new teleocidin analog (**271**) (Figure 16), was isolated. Compound **271** showed weak cytotoxic effects against HeLa with IC_50_ value of 49 μM [97]. Two new modified indole-containing peptides, JBIR 34–35 (**272**–**273**) (Figure 16), were produced by a marine-derived actinomycete *Streptomyces* sp. (strain Sp080513GE-23) originally derived from a marine sponge, *Haliclona* sp. (collected from Chiba Prefecture, Japan). Both compounds exhibited weak DPPH radical scavenging activity [98]. Two new anthracyclines, tetracenoquinocin (**274**) and 5-iminoaranciamycin (**275**) (Figure 16), were isolated from the culture broth of *Streptomyces* sp. Sp080513GE-26 collected from a marine sponge, *Haliclona* sp. (collected from Chiba Prefecture, Japan). Only compound **274** exhibited weak cytotoxicities with IC_50_ values of 120 and 210 μM against HeLa and HL-60 cells, respectively [99]. A new compound JBIR-107 (**276**) (Figure 16) was isolated from the culture of *Streptomyces tateyamensis* NBRC105047 isolated from the marine sponge, *Haliclona* sp. [100].

##### Family Niphatidae

Chemical examination of a marine fungus, *Truncatella angustata* XSB-01-43, isolated from a finger sponge *Amphimedon* sp. (collected in Yongxin Island, China), yielded 14 new isoprenylated cyclohexanols, namely truncateols A–N (**277**–**290**), and five new a-pyrone-based analogues, namely angupyrones A–E (**291**–**295**) (Figure 17). Compounds **279**, **281**, and **289** exerted significant inhibitory effects with IC_50_ values of 8.8–63 μM against H1N1 virus (oseltamivir as a positive control, IC_50_ 46.5 μM). Compounds **291**–**295** exhibited moderate antioxidant response element activation in HepG2C8 cells [101,102].

Five new isocoumarins, namely peyroisocoumarins A–D (**296**–**299**) and isocitreoisocoumarinol (**300**) (Figure 17) were produced by the sponge-associated fungus *Peyronellaea glomerate*, which was derived from a finger sponge *Amphimedon* sp. (Yongxin Island, China). Compounds **296**, **297**, and **299** exerted potent antioxidant response element activation in HepG2C8 cells [103]. Two new cadinane-type sesquiterpenes, hypocreaterpenes A–B (**301**–**302**) (Figure 17) were isolated from the fungal strain *Hypocreales* sp. strain HLS-104, which was isolated from a sponge *Gelliodes carnosa*; however, their bioassays were disappointing [104]. Two new monoterpenoid a-pyrones, named nectriapyrones C–D (**303**–**304**) (Figure 17), were isolated from the fungal strain *Nectria* sp. HLS206, which was isolated from the marine sponge *Gelliodes carnosa* collected from the South China Sea. None of them exhibited antibacterial activity nor cytotoxic activity [105]. Chemical study of the marine-derived *Streptomyces* sp. LS298 from the marine sponge *Gelliodes carnosa* (collected from Hainan Province, China) afforded a new analogue of echinomycin quinomycin G (**305**), together with a new cyclic dipeptide, cyclo-(L-Pro-4-OH-L-Leu) (**306**) (Figure 17). Compound **305** exhibited moderate antibacterial activities against *Staphylococcuse pidermidis*, *S. aureus*, *Enterococcus faecium*, and *E. faecalis* with MIC values ranging from 16 to 64 μg/mL and displayed remarkable anti-tumor activities against the tested cell lines ACHN, 786-O, and U87 MG [106].

The fungus *Curvularia lunata* was isolated from the marine sponge *Niphates olemda* (collected in Indonesia), and it yielded the new 1,3,8-trihydroxy-6-methoxyanthraquinone, which we named lunatin (**307**) (Figure 18). Compound **307** exhibited antibacterial activities against *Bacillus subtilis*, *Staphylococcus aureus*, and *Escherichia coli* [79]. A new hexaketide, pandangolide 1a (**308**) (Figure 18), was detected in the fungal strain *Cladosporium* sp., which was originally separated from the Red Sea sponge *Niphates rowi* [107]. The examination of the sponge-associated fungus *Stachybotrys chartarum* was isolated from the sponge *Niphates* sp. (GuangXi Province, China), and it led to the isolation of eight new isoindolinone-type alkaloids named chartarutines A–H (**309**–**316**) (Figure 18). Chartarutines **310**, **315**, and **316** exhibited significant inhibitory effects against HIV-1 virus [108].

Chemical analysis of a fungus *Stachybotrys chartarum* isolated from the sponge *Niphates recondite* WGC-25C-6 (Guangxi Province, China) afforded 16 new phenylspirodrimanes, named chartarlactams A–P (**317**–**332**), four new compounds, namely chartarenes A–D (**333**–**336**), and three phenylspirodrimane-based meroterpenoids with novel scaffolds, namely chartarolides A–C (**337**–**339**) (Figure 19). Compounds **320**–**322**, **327,** and **330**–**331** exhibited moderate antihyperlipidemic activities in HepG2 cells. Compounds **333**–**339** exerted potent or selective inhibition against a panel of tumor cell lines (including HCT-116, HepG2, BGC-823, NCIH1650, and A2780) and showed strong inhibitory activities against the human tumor-related protein kinases of FGFR3, IGF1R, PDGFRb, and TrKB [109,110,111].

Six new caryophyllene-based sesquiterpenoids named punctaporonins H–M (**340**–**345**), 10 new resorcinol derivatives named hansfordiols A–J (**346**–**355**), and three new salicylic acid derivatives (**356**–**358**) (Figure 20) were isolated from the marine fungus *Hansfordia sinuosae*, which was previously isolated from the sponge of *Niphates* sp. collected from Southern China Sea. Punctaporonin K (**343**) exhibited potent effects to reduce the triglycerides and total cholesterol in the intracellular levels. Compounds **340**–**345** showed weak cytotoxic activity against a panel of tumor cell lines with IC_50_ values more than 10 μM and showed weak inhibitory effects against the bacterial strains with the MIC values more than 125 μM. Compounds **346**–**358** did not do well in their bioassays [112,113,114].

##### Family Petrosiidae

The investigation of the Mediterranean sponge *Petrosia ficiformis*-derived fungus, *Penicillium brevicompactum*, yielded two previously unknown cyclodepsipeptides, petrosifungins A–B (**359**–**360**) (Figure 21) [115]. Five new compounds, pichiafurans A–C (**361**–**363**) and pichiacins A–B (**364**–**365**) (Figure 21), were isolated from the yeast *Pichia membranifaciens* derived from a marine sponge *Petrosia* sp. collected from South Korea [116]. New α-pyrones (**366**–**367**) and cyclohexenones (**368**–**369**) (Figure 21) were isolated from the fungus *Paecilomyces lilacinus*, which is a strain derived from a marine sponge *Petrosia* sp. (collected from Jeju Island) [117]. Two new meroterpenoids, terretonins E–F (**370**–**371**) (Figure 21), were detected in the marine-derived fungus *Aspergillus insuetus* associated with the sponge *Petrosia ficiformis* collected in the Mediterranean Sea. Compounds **370**–**371** showed activity as inhibitors of the mammalian mitochondrial respiratory chain with IC_50_ values of 3.90 ± 0.4 and 2.97 ± 1.2 μM, respectively [118]. A study on the sponge-derived fungus *Aspergillus versicolor* derived from the sponge *Petrosia* sp. (collected in Jeju Island, Korea) afforded a new peptide (**372**) and a new lipopeptide named fellutamide F (**373**) (Figure 21). Both compounds showed moderate cytotoxiciy against a panel of tumor cell lines (such as the skin cancer and colon cancer cells) with IC_50_ values ranging from 0.13 to 33.1 μM [119,120]. Two naturally rare dimeric indole derivatives (**374**–**375**) (Figure 21) were isolated from the marine actinomycete *Rubrobacter radiotolerans* cultured from a marine sponge *Petrosia* sp. (collected in Xisha Islands, China) and showed moderate acetylcholinesterase (AchE) inhibitory activity with IC_50_ values of 11.8 and 13.5 μM, respectively [121]. Futher examination of the sponge-derived actinomycete *Rubrobacter radiotolerans* produced one new dimeric indole derivative (**376**) (Figure 21), which exhibited the most effective antichlamydial activity with IC_50_ values of 46.6–96.4 µM [122]. A new diphenyl ether derivative, circinophoric acid (**377**) (Figure 21), was purified from the fungus strain of *Sporidesmium circinophorum* KUFA 0043 associated with the marine sponge *Petrosia* sp., which was collected from the Gulf of Thailand. The compound was disappointing in its bioassays [123].

Four novel secondary metabolites, namely aspergillone 1–3 (**378**–**380**) and 12-acetyl-aspergillol 4 (**381**) (Figure 22), were metabolized by fungus *Aspergillus versicolor* isolated from the marine sponge *Xestospongia exigua* (Indonesia) [124]. A marine strain of *Penicillium* cf. *montanense* derived from the marine sponge *Xestospongia exigua* (collected from the Bali Sea, Indonesia) was found to metabolize three novel decalactone metabolites, xestodecalactones A–C (**382**–**384**) (Figure 22). Compound **383** exhibited antifungal activity against the yeast *Candida albicans* [125]. Seven new angular tricyclic chromone derivatives (**385**–**391**) (Figure 22) were isolated from a fungus *Aspergillus versicolor* colonized in the marine sponge *Xestospongia exigua* (Indonesia). In the selected bioassay systems, only aspergillitine (**385**) displayed moderate antibacterial activity against *Bacillus subtilis* [126]. A new tyrosine-derived metabolite, aspergillusol A (**392**), and a novel sesquiterpenoid, asperaculin A (**393**) (Figure 22), were metabolized by the fungal strain *Aspergillus aculeatus* CRI323-04 associated with the marine sponge *Xestospongia testudinaria* (specimen no. CRI323) (Krabi Province). Compound **392** selectively inhibited α-glucosidase from the yeast *Saccharomyces cerevisiae* [127,128]. Four new bisabolane-type sesquiterpenoids, aspergiterpenoid A (**394**), (−)-sydonol (**395**), (−)-sydonic acid (**396**), (−)-5-(hydroxymethyl)-2-(2′,6′,6′-trimethyltetrahydro-2*H*-pyran-2-yl) phenol (**397**), and three new phenolic bisabolane sesquiterpenoid dimers, disydonols A–C (**398**–**400**) (Figure 22), were detected in the cultures of the marine-derived fungal strain *Aspergillus* sp., which was isolated from the sponge *Xestospongia testudinaria* (South China Sea). Compounds **394**–**397** showed selective antibacterial activity against eight bacterial strains (MIC values, 1.25–20.0 µM). Compounds **398** and **400** exhibited cytotoxicity against HepG-2 and Caski human tumor cell lines [129,130].

Three naturally new C-glycosylated benz[a]anthraquinone derivatives—urdamycinone E (**401**), urdamycinone G (**402**), and dehydroxyaquayamycin (**403**) (Figure 23)—were from the marine *Streptomycetes* sp. BCC45596, which was isolated from a marine sponge, *Xestospongia* sp. (Thailand). Compounds **401**–**403** exhibited potent antiplasmodial activity toward the *Plasmodium falciparum* K1 strain with IC_50_ values in a range of 0.0534–2.93 μg/mL, anti-Mycobacterium tuberculosis with MICs in a range of 3.13–12.50 μg/mL and cytotoxicity against cancerous (KB, MCF-7, NCI-H187) and non-cancerous (Vero) cells with IC_50_ values in a range of 0.092–10.07 μg/mL [131]. The chemical examination of a marine-derived fungus *Stachybotrys chartarum* MXH-X73 from the sponge *Xestospongia testudinaris* (Xisha Island, China) afforded seven new phenylspirodrimanes, named stachybotrins D–F (**404**–**406**), stachybocins E–F (**407**–**408**), stachybosides A–B (**409**–**410**), and a new sulfate meroterpenoid, stachybotrin G (**411**) (Figure 23). Only compound **404** exhibited anti-HIV activity by targeting reverse transcriptase [132,133]. Chemical examination of a *Streptomyces* species (S.4), isolated from the sponge *Xestospongia muta* collected from the Florida Keys, yielded a new dipeptide named xestostreptin (**412**) (Figure 23). Compound **412** exhibited weak antiplasmodial activity against the Dd2 strain of *Plasmodium falciparum*, with IC_50_ values of 50 μM [134].

#### 2.2.9. Order Poecilosclerida

##### Family Acarnidae

Three novel cytotoxic polyketides, brocaenols A–C (**413**–**415**) (Figure 24), were metabolized by a marine-derived fungus *Penicillium brocae* associated with the Sponge *Zyzzya* sp. collected in Fiji. All three compounds showed moderate to weak cytotoxicity against the HCT-116 cell line with the IC_50_ values of 20, 50, and >50 μg/mL, respectively [135].

##### Family Microcionidae

Chemical investigation of the marine-derived fungus *Neosartorya quadricincta* KUFA 0081 led to the isolation of two new pentaketides, including a new benzofuran-1-one derivative (**416**) and a new isochromen-1-one (**417**), and seven new benzoic acid derivatives, including two new benzopyran derivatives (**418**–**419**), a new benzoxepine derivative (**420**), two new chromen-4-one derivatives (**421**–**422**), and two new benzofuran derivatives (**423**–**424**) (Figure 24). The strain KUFA 0081 was isolated from the marine sponge *Clathria reinwardti*, which was collected in the Gulf of Thailand. None of the isolated compounds exhibited activities in their bioassays [136]. The investigation of the marine sponge *Clathria reinwardtii* (Thailand)-derived fungus, *Neosartorya fennelliae* KUFA 0811, yielded a previously unreported dihydrochromone dimer, paecilin E (**425**) (Figure 24), whose bioassay was disappointing [137]. Three new secondary metabolites, chaetoglobosin-510 (**426**), -540 (**427**), and -542 (**428**) (Figure 24) were isolated from the fungus *Phomopsis asparagi* cultured from the sponge *Rhaphidophlus juniperina* (U.S. Virgin Island). Compound **428** displayed antimicrofilament activity and anti-tumor activity toward murine colon and leukemia cancer cell lines [138].

##### Family Mycalidae

A new compound, (*S*)-2,4-dihydroxy-1-butyl(4-hydroxy) benzoate (**429**) (Figure 25), was from the fungus *Penicillium auratiogriseum*; it was isolated from the sponge *Mycale plumose* (collected in Qingdao, China) and showed potent anti-tumor activity in tsFT210 cells, with a maximum inhibitory effect observed at 8.0 μg/mL [139]. Three new quinazoline alkaloids, aurantiomides A–C (**430**–**432**) (Figure 25), were found from the fungus *Penicillium aurantiogriseum* SP0-19, which isolated from the sponge *Mycale plumose* collected in Qingdao, China. Compounds **431** and **432** exhibited moderate cytotoxicities against HL-60, P388, and BEL-7402, P388 cell lines [140]. Chemical investigation of the marine sponge-associated bacterium *Pseudoalteromonas rubra* CMMED 294 obtained from a small piece of sponge, most likely *Mycale armata* (collected in Kaneohe Bay off Oahu, HI, United States of America), led to the isolation of a 2-Substituted Prodiginine, 2-(p-Hydroxybenzyl) prodigiosin (**433**) (Figure 25) [141]. A new aspochracin derivative JBIR-15(**434**) (Figure 25) was from a sponge-derived fungus *Aspergillus sclerotionrum Huber* Sp080903f04, which was isolated from a sponge *Mycale* sp., collected from *Ishigaki Islang*, Japan. Compound **434** did not show any cytotoxic effects in its bioassays [142]. A new diketopiperazine dimer, eurocristatine (**435**) (Figure 25), was isolated from the fungal strain *Eurotium cristatum* KUFC 7356; in turn, the fungus was isolated from the marine sponge *Mycale* sp., which was collected from Thailand. Compound **435** exhibited neither cytotoxic, antibacterial, nor antifungal activity [143]. Chemical analysis of the marine fungus *Talaromyces tratensis* KUFA 0091 derived from the marine sponge *Mycale* sp. (Thailand) afforded a new isocoumarin derivative tratenopyrone (**436**) (Figure 25), whose bioassay was disappointing [123]. Chemical investigation of the marine sponge-associated fungus *Neosartorya glabra* KUFA 0702, which was obtained from the marine sponge *Mycale* sp. (Thailand), led to the isolation of two new cyclotetrapeptides, sartoryglabramides A–B (**437**–**438**), and a new analog of fellutanine A (**439**) (Figure 25). None of them exhibited either antibacterial or antifungal activities [144].

##### Family Myxillidae

A new equisetin-like tetramic acid derivative, beauversetin (**440**) (Figure 25), was isolated from the sponge-derived fungus *Beauveria bassiana*; in turn, this was isolated from the sponge *Myxilla incrustans*, which was collected from the island of Helgoland. Compound **440** exihibited moderate anti-tumor activity against a six-cell line panel for a monolayer assay (IC_50_, 3.09 μg/mL) [145].

#### 2.2.10. Order Scopalinida

##### Family Scopalinidae

Chemical investigation of a sponge-derived *Streptomyces* sp. from the sponge *Scopalina ruetzleri* (Puerto Rico) led to the isolation of six new angucyclinone derivatives monacyclinones A–F (**441**–**446**) (Figure 26). Monacyclinones A–F (**441**–**446**) showed potent bioactivity against human rhabdomyosarcoma cancer cells (IC_50_, **446**, 0.73 μM) and were active against Gram-positive bacteria [146]. From a strain of the marine-derived fungus *Aspergillus aculeatus* strain CRI322-03 derived from a marine sponge *Stylissa flabelliformis* (Thailand), three new compounds, pre-aurantiamine (**447**), (−)-9-hydroxyhexylitaconic acid (**448**), and (−)-9-hydroxyhexylitaconic acid-4-methyl ester (**449**) (Figure 26) were isolated [147]. A new ergosterol analog, talarosterone (**450**), and a new bis-anthraquinone derivative (**451**) (Figure 26) were isolated from the marine fungus *Talaromyces stipitatus* KUFA 0207 isolated from the marine sponge *Stylissa flabelliformis* (Thailand) [148].

#### 2.2.11. Order Sphaerocladina

From a strain of the fungus *Hypoxylon monticulosum* CLL-205 derived from a Sphaerocladina sponge collected from the Tahiti coast (France), two sporothriolide-related compounds (**452**–**453**) (Figure 26) were isolated. Only compound **452** exhibited moderate cytotoxicity against the HCT-116 cell line with IC_50_ values of 18 μM [149].

#### 2.2.12. Order Suberitida

##### Family Halichondriidae

Chemical examination of secondary metabolites produced by the marine sponge-associated fungus *Gymnasella dankaliensis* (Castellani) Currah OUPS-N134, yielded 11 new compounds, gymnasterones A–B (**454**–**455**), gymnastatins D–K (**456**–**463**), gymnamide (**464**), and a structurally unique and cytotoxic steroid (**465**) (Figure 27), which was designated dankasterone. The strain OUPS-N134 was isolated from the marine sponge *Halichondria japonica*, which was collected in the Osaka Bay of Japan. Compounds **454**–**459**, **461**–**463**, and **465** exhibited significant to weak cytotoxic activity in the P388 lymphocytic leukemia test system in cell culture with ED_50_ values ranging from 0.021 to 10.8 μg/mL. Furthermore, compounds **461** and **462** showed appreciable growth inhibition against the selected human cancer cell lines [150,151,152,153,154]. Further study on the fungus *Gymnacella dankaliensis* OUPS-N134 derived from the sponge *Halichondria japonica* (collected in Japan) afforded six extremely unusual steroids, dankasterones A–B (**466**–**467**) and gymnasterones A–D (**468**–**471**), and four new metabolites, gymnastatins Q–R (**472**–**473**) and dankastatins A–B (**474**–**475**) (Figure 28). All the compounds except compound **468** exhibited significant and marginal growth inhibition against the murine P388 cell line with ED_50_ values ranging from 0.15 to 2.8 μg/mL. Furthermore, compound **472** showed appreciable growth inhibition against the BSY-1 (breast) and MKN7 (stomach) human cancer cell lines [155,156].

Five novel metabolites, trichodenones A–C (**476**–**478**) and harzialactone A–B (**479**–**480**) (Figure 29), were isolated from *Trichoderma harzianum* OUPS-N 115 colonizing in the sponge *Halichondria okadai*, which was collected in the Tanabe Bay of Japan. Compounds **476**–**478** exhibited significant cytotoxicity against cultured P388 cells with ED_50_ values of 0.21, 1.21, and 1.45 μg/mL, respectively [157]. Four unusual cell-associated glycoglycerolipids (**481**–**484**) (Figure 29) were metabolized by a marine-derived bacterium *Microbacterium* sp. associated with the sponge *Halichondria panicea*, which was collected along the Adriaticcoast, Rovinj, Croatia [158]. Novel antibiotics, YM-266183 (**485**) and YM-266184 (**486**) (Figure 29), were detected in the *Bacillus cereus* QN03323, which was isolated from the marine sponge *Halichondria japonica* (collected in Japan). Both compounds were active against Gram-positive bacteria including Methicillin-resistant *Staphylococcus aureus* (MRSA), Methicillin-resistant *Streptococcus epidermidis* (MRSE), and vancomycin-resistant *Enterococci* (VRE) with MIC values ranging from 0.025 to 1.56 μg/mL [159]. Chemical examination of a marine fungus *Clonostachys rogersoniana* strain HJK9, isolated from a sponge *Halicondria japonica* (collected in Numazu, Japan), yielded two new anti-dinoflagellates clonostachysins A–B (**487**–**488**) (Figure 29). Both compounds exhibited a selectively anti-dinoflagellate against *Prorocentrum micans* at 30 μM [160].

The investigation of the marine *Streptomyces* HB202 from the marine sponge *Halichondria panacea* from the Baltic Sea (Germany) yielded eight new phenazines, streptophenazines A–H (**489**–**496**), a new benz[a]anthracene derivative called mayamycin (**497**), and three new streptophenazines I–K (**498**–**500**) (Figure 30). Compound **497** exhibited potent cytotoxic activity against eight human cancer cell lines, and compounds **498**–**500** showed moderate activities against the enzyme phosphodiesterase PDE 4B with IC_50_ values of 11.1, 12.0, and 12.2 μg/mL, respectively. Compounds **491**, **496**, **497**, **499**, and **500** exhibited moderate antibacterial activity against the selected strains [161,162,163]. Chemical examination of the fungal strain *Exophiala* sp. from the marine sponge *Halichondria panicea* (Korea) afforded two unusual compounds, Chlorohydroaspyrones A–B (**501**–**502**) (Figure 30). Both compounds showed weak antibacterial activity [164]. One fungal strain, *Pichia membranifaciens* USF-HO25, derived from the sponge *Halichondria okadai* (collected in Japan) was found to produce two new indole derivatives (**503**–**504**) (Figure 30), which displayed weak DPPH (2,2-diphenyl-1-picrylhydrazyl) radical scavenging activities [165]. Four new γ-pyrones, nocapyrones A–D (**505**–**508**) (Figure 30), were found in an associated actinomycete *Nocardiopsis* strain HB383 in the marine sponge *Halichondria panicea*, which was collected from the Baltic Sea (Germany). No bioactivities were detected [166].

A fungal strain of *Trichoderma harzianum* OUPS-111D-4 from the marine sponge *Halichondria okadai* yielded six novel decalin derivatives, Tandyukisins A–F (**509**–**514**), and three new compounds, trichodermanins C–E (**515**–**517**) (Figure 31). Compounds **513**–**515** exhibited significant cytotoxicity against the cancer cell lines P388, HL-60, and L1210 leukemia cell lines with IC_50_ values ranging from 3.8 to 7.9 µM, respectively [167,168,169,170]. An unusual polyketide with a new carbon skeleton, lindgomycin (**518**) (Figure 31), was metabolized by a marine fungus of *Lindgomycetaceae* strain LF327 isolated from the sponge *Halichondria panicea* (Germany) and showed antibiotic activities with IC_50_ value of 5.1 ± 0.2 µM against methicillin-resistant *Staphylococcus aureus* [171].

Chemical investigation of an associated actinomycete *Streptomyces microflavus*, which was obtained from the sponge *Hymeniacidon perlevis* (collected from Dalian, China), led to the isolation of one new nucleoside derivative named 3-acetyl-5-methyl-2′-deoxyuridine (**519**) (Figure 32) [172]. From the sponge *Hymeniacidon* sp. (East China Sea), an actinomycete *Streptomyces carnosus* strain AZS17 was isolated and shown to produce two novel kijanimicin derivatives named lobophorin C–D (**520**–**521**) (Figure 32). Lobophorin C displayed potent cytotoxic activity against the human liver cancer cell line 7402 with an IC_50_ value of 0.6 μg/mL, while lobophorin D showed significant inhibitory effect on human breast cancer cells MDA-MB 435 with an IC_50_ value of 7.5 μM [173]. A marine sponge-derived fungus strain *Aspergillus versicolor* MF359 isolated from a marine sponge of *Hymeniacidon perleve* (collected from the Bohai Sea, China) afforded three new secondary metabolites, named hemiacetal sterigmatocystin (**522**), acyl-hemiacetal sterigmatocystin (**523**), and 5-methoxydihydrosterigmatocystin (**524**) (Figure 32). Compound **524** showed activity against *S. aureus* and *B. subtilis* with MIC values of 12.5 and 3.125 μg/mL, respectively [174]. From the sponge *Hymeniacidon perleve* (collected from the Bohai Sea, China), fungus *Aspergillus versicolor* Hmp-F48, a new dibenzo [1,4] dioxin **525**, and two new prenylated diphenyl ethers **526**–**527** (Figure 32) were identified and found to show potent cell growth inhibitory activities against the HL-60 cell line with IC_50_ values of 3.26, 6.35, and 19.97 μM, respectively [175]. Chemical examination of the fungal strain *Pseudogymnoascus* sp. F09-T18-1, which is associated with an Antarctic marine sponge belonging to the genus *Hymeniacidon*, yielded four new nitroasterric acid derivatives, pseudogymnoascins A–C (**528**–**530**) and 3-nitroasterric acid (**531**) (Figure 32), which were inactive against a panel of bacteria and fungi [176]. A new minor diketopiperazine alkaloid (**532**) and a natural lactone (**533**) (Figure 32) were detected in the sponge-derived fungus *Simplicillium* sp. YZ-11 originally separated from the marine sponge *Hymeniacidon perleve* collected from Dalian, China [177]. Chemical analysis of the fungus *Trichoderma* sp. HPQJ-34 isolated from the marine sponge *Hymeniacidon perleve* (Zhejiang, China) afforded a new cyclopentenone, 5-hydroxycyclopenicillone (**534**) (Figure 32). Compound **534** has moderate antioxidative, anti-Aβ fibrillization activities, and neuroprotective effects, and it might be a good free radical scavenger [178].

##### Family Suberitidae

Two new cyclic peptides, cyclo-(glycyl-l-seryl-l-prolyl-l-glutamyl) and cyclo-(glycyl-l-prolyl-l-glutamyl) (**535**–**536**) (Figure 33), were produced by a *Ruegeria* strain cultured from the Sponge *Suberites domuncula*. Both compounds showed moderate activity against *Bacillus subtilis* [179]. One fungal strain, *Aspergillus ustus*, derived from the sponge *Suberites domuncula* (collected from the Adriatic Sea) was found to produce seven new drimane sesquiterpenoids (**537**–**543**), five new ophiobolin-type sesterterpenoids (**544**–**548**), and the two new pyrrolidine alkaloids (**549**–**550**) (Figure 33). Compound **540** showed moderate cytotoxic activity against L5178Y with an EC_50_ value of 5.3 µg/mL, while compound **541** showed significant cytotoxic activity against L5178Y, HeLa, and PC12 cells with EC_50_ values of 0.6, 5.9, and 7.2 µg/mL, respectively [180,181]. Three new compounds, bendigoles D–F (**551**–**553**) (Figure 33), were produced by a strain of bacterium, *Actinomadura* sp. SBMs009, a symbiont on the marine sponge *Suberites japonicus*. Compound **551** was the most active inhibitor of GR-translocation (GR: glucocorticoid receptor) and exhibited cytotoxicity against the L929 (mouse fibroblast) cell line with an IC_50_ value approximated to 30 μM, while compound **553** was the most effective inhibitor of NF-κB nuclear translocation with an IC_50_ value of 71 μM [182]. Two new benzophenones, acredinones A–B (**554**–**555**) (Figure 33), were characterized from a marine-sponge-associated *Acremonium* sp. F9A015 fungus, associated with a marine sponge *Suberites japonicas* (Korea). Compounds **554**–**555** inhibited the outward K^+^ currents of the insulin secreting cell line INS-1 with IC_50_ values of 0.59 and 1.0 μM, respectively [183].

#### 2.2.13. Order Tethyida

##### Family Tethyidae

A fungal strain, *Scopulariopsis brevicaulis*, was obtained from the marine sponge *Tethya aurantium* (collected in Croatia), and it was found to produce two novel cyclodepsipeptides, scopularides A–B (**556**–**557**) (Figure 34). Both compounds exhibited significant activity against several tumor cell lines at 10 µg/mL [184]. The new metabolite cillifuranone (**558**) (Figure 34) was isolated from the culture broth of *Penicillium chrysogenum* strain LF066 collected from the Mediterranean marine sponge *Tethya aurantium*. Compound **558** exhibited weak antibiotic bioactivities [185]. Chemical analysis of the marine fungus *Bartalinia robillardoides* strain LF550 derived from the Mediterranean sponge *Tethya aurantium* afforded three new chloroazaphilones—helicusin E (**559**), isochromophilone X (**560**), and isochromophilone XI (**561**)—and one new pentaketide, bartanolide (**562**) (Figure 34). **561** revealed specifically weak activity against *T. rubrum* [186]. Chemical investigation of a marine-derived fungus *Aspergillus* sp. from the marine sponge *Tethya aurantium* (collected from the Adriatic Sea) led to the isolation of a new tryptophan-derived alkaloid and a new meroterpenoid, austalide R (**563**–**564**) (Figure 34). Compound **563** selectively inhibited Vibrio species, and compound **564** showed a broad spectrum of antibacterial activity with MIC values between 0.01 and 0.1 μg/mL [187].

#### 2.2.14. Order Tetractinellida

Off the coast of Korea, a Choristida sponge-derived fungus, *Acremonium strictum*, yielded a novel natural product acremostrictin (**565**) and a novel modified base, acremolin (**566**) (Figure 35). Compound **565** exhibited weak antibacterial and moderate antioxidant activities (IC_50_, 2.1 mM), and compound **566** exhibited weak cytotoxicity against an A549 cell line with an IC_50_ of 45.90 μg/mL [188,189].

##### Family Ancorinidae

A new polyketide, deoxynortrichoharzin (**567**) (Figure 35), was produced by the marine-derived fungus *Paecilomyces* cf. *javanica*, which is a symbiont on the sponge *Jaspis* cf. *coriacea* (collected in the Fiji Islands). Compound **567** did not show any activity in solid-tumor cells in culture [190]. A new bile acid derivative (**568**) (Figure 35) was metabolized by a marine sponge-associated bacterium *Psychrobacter* sp. associated with the marine sponge *Stelletta* sp., which was collected off the coast of Geoje Island, Korea. Compound **568** exhibited moderate suppressive effects on both NO and interleukin-6 (IL-6) production at a concentration of 87.3 μg/mL [191]. New sesquiterpenoids (**569**–**572**) (Figure 35) were isolated from the fungal strain *Acremonium* sp., which was isolated from a marine sponge *Stelletta* sp. (collected in Korea). Compound **569** exhibited weak anti-inflammatory activity in RAW 264.7 murine macrophage cells [192]. A bacterial strain J05B1-11, isolated from the marine sponge *Stelletta* sp. (Korea), yielded a new natural product: Sym-Tetra (**573**) (Figure 35). Compound **573** was non-cytotoxic according to this study [193]. Four new hexylitaconic acid derivatives (**574**–**577**) (Figure 35) were isolated from a sponge-derived fungus *Penicillium* sp., which was isolated from a sponge *Stelletta* sp. collected in Jeju island, Korea [194]. The investigation of the fungus *Aspergillus sydowii* from the sponge *Stelletta* sp. yielded two new metabolites, diorcinolic acid (**578**) and β-D-glucopyranosyl aspergillusene A (**579**) (Figure 35). Compounds **578**–**579** showed mild cytotoxicity against several human cancer cells with IC_50_ values ranging from 50 to 70 μM [195].

##### Family Geodiidae

A marine strain of the fungus *Arthrinium* sp. derived from the Mediterranean sponge *Geodia cydonium* was found to metabolize five new diterpenoids, arthrinins A–D (**580**–**583**) and myrocin D (**584**) (Figure 35). Myrocin D (**584**) inhibited vascular endothelial growth factor A (VEGF-A)-dependent endothelial cell sprouting with IC_50_ values of 2.6 μM [196].

##### Family Neopeltidae

Four quinolone derivatives, **585**–**588** (Figure 36), were isolated from a bacterial strain *Pseudomonas* 1531-E7 associated with the sponge *Homophymia* sp. collected from New Caledonia. Compound **585** was active against *Plasmodium falciparum* and against HIV-1 with ID_50_ values of 1 and 10^−3^ μg/mL, respectively. Compound **586** exhibited mild cytotoxicity against KB cells (IC_50_, 5 μg/mL) and was active against *Plasmodium falciparum* (ID_50_, 3.4 μg/mL). Compound **587** was active against *Plasmodium falciparum* with an ID_50_ value of 4.8 μg/mL. Compound **588** showed antimicrobial activity against *S. aureus* and cytotoxicity toward KB cells with an IC_50_ value less than 2 μg/mL [197].

##### Family Tetillidae

Pseudoalterobactin A–B (**589**–**590**) (Figure 36) were detected in the marine bacterium *Pseudoalteromonas* sp. KP20-4 originally separated from a marine sponge (*Cinachyrella australiensis*) obtained in the Republic of Palau [198]. Three novel isoprenoids, JBIR 46–48 (**591**–**593**) (Figure 36), were isolated from an actinomycetic strain *Streptomyces* sp. SpC080624SC-11, which was associated with the marine sponge *Cinachyra* sp. (Japan) [199]. From a fungal strain *Emericella variecolor* XSA-07-2 derived from a marine sponge *Cinachyrella* sp. (the South China Sea), seven new polyketide derivatives, namely, varioxiranols A–G (**594**–**600**) (Figure 36), a new hybrid PKS-isoprenoid metabolite (**601**) (Figure 36), four new lactones—namely, varioxiranols I–L (**602**–**605**) (Figure 36), and three novel asteltoxin-bearing dimers—namely, diasteltoxins A–C (**606**–**608**) (Figure 37)—were isolated. Three structurally novel asteltoxin-bearing dimers were characteristic of a [2+2] cycloaddition of asteltoxin in different manners. Compounds **594**, **602**–**605**, and **606**–**608** showed weak to moderate cytotoxic activity against the selected tumor cell lines [200,201,202]. Chemical investigation of the marine-derived *Streptomyces* sp. DA22 led to the isolation of a new indole alkaloid, streptomycindole (**609**) (Figure 37). The strain was isolated from the South China Sea sponge *Craniella australiensis*. Compound **609** showed no cytotoxicity against several selected tumor cell lines [203].

##### Family Theonellidae

Chemical examination of a marine-derived *Escherichia coli* from the sponge *Discodermia calyx* (collected in Japan) afforded a novel pyridinium with three indole moieties: tricepyridinium **610** (Figure 37). **610** showed antimicrobial activity against *B. cereus*, *S. aureus*, and *C. albicans* (with MIC values of 0.78, 1.56, and 12.5 μg/mL, respectively) and cytotoxicity to P388 cells with an IC_50_ value of 0.53 μg/mL [204]. Study on the metabolites of the sponge-derived *Streptomyces* sp. GIC10-1 derived from the sponge *Theonella* sp. (Taiwan) afforded a new 16-membered diene macrolide: bafilomycin M (**611**) (Figure 37). Compound **611** exhibited significant cytotoxicity toward K-562, HL-60, SUPT-1, and LNCaP tumor cells with IC_50_ values of 0.060, 0.011, 0.047, and 0.389 μg/mL, respectively [205].

#### 2.2.15. Order Verongiida

##### Family Aplysinidae

Two new betaenone derivatives (**612**–**613**), three new 1,3,6,8-tetrahydroxyanthraquinone congeners (**614**–**616**), and two new metabolites, microsphaerones A–B (**617**–**618**) (Figure 38) were isolated from an undescribed fungus of the genus *Microsphaeropsis*, which was isolated from the Mediterranean sponge *Aplysina aerophoba*. Compounds **612** and **614**–**616** are inhibitors of protein kinase C (PKC)-ε, the cyclin-dependent kinase 4 (CDK4), and the epidermal growth factor receptor (EGF-R) tyrosine kinases with IC_50_ values ranging from 18.5 to 54.0 μM [206,207]. A new compound named hortein (**619**) (Figure 38) was isolated from the fungus *Hortaea werneckii* isolated from the Mediterranean sponge *Aplysina aerophoba*. When tested for antibiotic or insecticidal activity, the bioassays were disappointing [208]. The strain of the fungus *Cladosporium herbarum*, isolated from the sponges *Aplysina aerophoba* collected in the Mediterranean Sea, yielded two new R-pyrones, herbarin A–B (**620**–**621**) (Figure 38). Compounds **620** and **621** showed activity against *Artemia salina* but exhibited no significant antibiotic activity [79]. A new glutarimide (**622**) (Figure 38) was isolated from a marine-derived *Streptomyces anulatus* S71 isolated from a marine sponge *Aplysina aerophoba* (South China Sea) [209].

##### Family Ianthellidae

Six new acremine metabolites, 5-chloroacremine A (**623**), 5-chloroacremine H (**624**), and acremines O–R (**625**–**628**) (Figure 38), were isolated from a marine-derived fungus *Acremonium persicinum*, which was isolated from the sponge *Anomoianthella rubra* collected from Mooloolaba, QLD [210]. Chemical study of an unidentifiable sponge-derived fungus from an Ianthella sponge (Papua New Guinea) afforded two novel cyclic depsipeptides, guangomides A–B (**629**–**630**), together with a new destruxin derivative (**631**) (Figure 38). Compounds **629** and **630** exhibited weak antibacterial activity against *Staphylococcus epidermidis* and *Enterococcus durans* [211].

##### Family Pseudoceratinidae

New compounds termed JBIR 97–99 (**632**–**634**) (Figure 38) were isolated from the culture of *Tritirachium* sp. SpB081112MEf2 associated with the marine sponge, *Pseudoceratina purpurea*, collected from Okinawa Prefecture, Japan. Compounds **632**–**634** showed cytotoxic effects against HeLa cells (IC_50_, 11, 17, and 17 μM, respectively) and ACC-MESO-1 cells (IC_50_, 31, 63, and 59 μM, respectively) [212].

### 2.3. Unidentified

Two new diketopiperazines (**635**–**636**), and orcinotriol (**637**) (Figure 39), a new 1,3-dihydroxyphenol derivative, were isolated from the yeast *Aureobasidium pullulans*, which was separated from an unidentified marine sponge collected in Okinawa [213]. Four novel hexaketide compounds—iso-cladospolide B (**638**), seco-patulolide C (**639**), and the 12-membered macrolides, pandangolide 1–2 (**640**–**641**) (Figure 39)—were isolated from a marine fungal species isolated from a bright orange, encrusting sponge collected in Indonesia. No bioactivity was detected [214]. Chemical examination of a marine-derived fungus *Aspergillus niger* FT-0554 from a marine sponge (collected in Palau Islands) afforded a novel compound: nafuredin (**642**) (Figure 39). Compound **642** inhibited the anaerobic electron transport of Ascaris suum (pig roundworm) [215].

The new compounds varitriol (**643**), varioxirane (**644**), dihydroterrein (**645**), and varixanthone (**646**) (Figure 39) were identified from the fungus *Emericella variecolor*, which was isolated from a sponge collected in the Caribbean waters. In the National Cancer Institute’s 60-cell panel, varitriol (**643**) showed increased potency toward selected renal, CNS, and breast cancer cell lines. Varixanthone (**646**) showed antimicrobial activity against *E. coli*, *Proteus* sp., *B. subtillis*, and *S. aureus*, showing a minimal inhibitory concentration (MIC) of 12.5 μg/mL [216]. Three new chlorine containing compounds (**647**–**649**), three new 14-membered macrolides, named aspergillides A–C (**650**–**652**), a new compound named circumdatin J (**653**), and a stephacidin, 21-hydroxystephacidin A [(+)-2] (**654**) (Figure 39) were isolated from a marine-derived fungus *Aspergillus ostianus* strain TUF 01F313 isolated from a marine sponge collected at Pohnpei. Three new chlorinated compounds (**647**–**649**) showed antibacterial activity against the *Ruegeria atlantica* strain. Compounds **650**–**652** showed cytotoxic activity against mouse lymphocytic leukemia cells (L1210) with LD_50_ values of 2.1, 71.0, and 2.0 μg/mL, respectively [217,218,219,220]. Chemical study of the marine-derived *Pseudomonas* sp. F92S91 from a marine sponge sample collected in Fiji afforded two new α-pyrones, pseudopyronines A–B (**655**–**656**) (Figure 39). Both compounds showed moderate to poor antibacterial activities against Gram-positive bacteria [221]. The examination of an actinomycete of the genus *Streptomyces* isolated from an unidentified marine sponge (Korea) led to the isolation of two new cyclic peptides (**657**–**658**) (Figure 39). Both compounds exhibited weak inhibition against the enzyme sortase B with EC_50_ values of 88.3 and 126.4 μg/mL, respectively [222].

IB-01212, a new cytotoxic cyclodepsipeptide featuring C2 symmetry (**659**) (Figure 40), was isolated from cultures of the marine fungus *Clonostachys* sp. ESNA-A009, which was previously isolated from an unidentified marine sponge collected in Japan. Compound **659** exihibited highly cytotoxic activity to different tumor cell lines [223]. We have identified tropolactones A–D (**660**–**663**) (Figure 40), which are four new cytotoxic meroterpenoids from a marine-derived *Aspergillus* sp. (strain CNK-371) that was isolated from an unidentified sponge collected at Manele Bay, Hawaii. Compounds **660**–**661** showed moderate cytotoxicity against human colon carcinoma (HCT-116) with IC_50_ values of 13.2, 10.9, and 13.9 μM, respectively [224].

Chemical examination of a marine-derived fungus, *Arthrinium* sp., isolated from a marine sponge collected in the Japan Sea, yielded a new inhibitor of p53-HDM2 interaction (**664**) (Figure 40). Compound **664** inhibited the p53-HDM2 binding with an IC_50_ value of 50 μg/mL [225]. A new macrocyclic trichothecene, named roridin R (**665**) (Figure 40), was from the fungus *Myrothecium* sp. TUF 02F6, which was isolated from an unidentified marine sponge collected in Manado, Indonesia. Compound **665** showed significant cytotoxicity against the murine leukemia cell line L1210 with an IC_50_ value of 0.45 μM [226]. Three new polyketide-originated compounds (**666**–**668**) (Figure 40) were isolated from a marine fungus *Mycelia sterilia* derived from a sponge [227]. One new siderophore (**669**) (Figure 40) was isolated from a bacterium isolated from unidentified marine sponges collected in Indonesia. The MIC value of the compound **669** by chrome azurol S (CAS) liquid assay is 156 µg/mL [228]. Chemical investigation of the marine-derived fungus *Fusarium* sp. 05ABR26 led to the isolation of a new β-resorcylic macrolide, 5′-hydroxyzearalenol (**670**) (Figure 40). The strain 05ABR26 was isolated from a sponge collected in Miura Peninsula of Japan. Compound **670** showed no obvious activity against *Pyricularia oryzae* [229]. Three new aminolipopeptides that were designated trichoderins A (**671**), A1 (**672**), and B (**673**) (Figure 40) were from fungus *Trichoderma* sp. 05FI48, which was isolated from the unidentified marine sponge. Trichoderins showed potent anti-mycobacterial activity against *Mycobacterium smegmatis*, *Mycobacterium bovis* BCG, and *Mycobacterium tuberculosis* H37Rv with MIC values ranging from 0.02 to 2.0 μg/mL [230]. Chemical investigation of the marine sponge-associated *Actinomadura* sp. SpB081030SC-15, which was obtained from an unidentified marine sponge (Japan), led to the isolation of a new diterpene compound designated JBIR-65 (**674**) (Figure 40). The bioassays of compound **674** were disappointing [231]. An investigation of the marine sponge-derived *Streptomyces* sp. strain RM72 from an unidentified marine sponge (Japan) yielded three new trichostatin analogues, JBIR 109–111 (**675**–**677**) (Figure 40). The IC_50_ values against HDAC1 of compounds **675**–**677** were 48, 74, and 57 μM, respectively [232]. Three new depsipeptides termed JBIR 113–115 (**678**–**680**) (Figure 40) were isolated from a culture of *Penicillium* sp. fS36, which were isolated from an unidentified marine sponge collected near Takarajima Island, Japan. Compounds **678**–**680** did not show cytotoxicity to human cervical carcinoma HeLa cells lines (IC_50_ > 100 µM) or antimicrobial activity against *Micrococcus luteus* or *Escherichia coli* [233].

Chemical investigation of a marine sponge-derived fungus, *Aspergillus unguis* CRI282-03, derived from an unidentified marine sponge CRI282 (collected in Thailand), afforded three new depsidones (**681**–**683**), a new diaryl ether (**684**), and a new natural pyrone (**685**) (Figure 41). Compounds **681**–**682** showed radical scavenging activity, and compound **683** showed the most potent aromatase inhibitory activity with the IC_50_ value of 0.74 µM. Compound **684** exhibited moderate cytotoxic activity against MOLT-3 cancer cell lines with an IC_50_ value of 8.8 μM [234]. 1-Hydroxy-10-methoxy-dibenz [b, e] oxepin-6,11-dione (**686**) (Figure 41) was purified from the fungus strain of *Beauveria bassiana* TPU942, which was associated with a marine sponge collected in Okinawa, Japan. Compound **686** did not show any apparent activity in the bioassays [235]. Chemical study of the sponge-derived fungus, *Gymnascella dankaliensis*, afforded a new polyketide tyrosine derivative, dankastatin C (**687**) (Figure 41). Compound **687** showed potent cell growth inhibitory activity against the murine P388 cell line with an ED_50_ value of 57 ng/mL [236]. Fractionation of the marine-derived fungus *Metarhizium anisopliae* mxh-99 from an unidentified sponge (Guangxi Province, China) yielded two new naphtho-c-pyrones glycosides, indigotides G–H (**688**–**689**) (Figure 41) [237]. Chemical examination of a marine-derived *Kocuria palustris* strain F-276345, which was isolated from a sponge sample (collected in Florida Keys, United States of America), yielded a new thiazolyl peptide, kocurin (**690**) (Figure 41). Compound **690** displayed activity against methicillin-resistant *Staphylococcus aureus* (MRSA), with an MIC value of 0.25 μg/mL [238]. One new compound (**691**) (Figure 41) was isolated from the marine fungus *Paecilomyces* sp. cultured from the marine sponge (collected along Tinggi Island, Malaysia). Compound **691** inhibited MRSA, with inhibition zones of 8 (±0.07) mm [239].

Chemical examination of the sponge-derived fungus *Penicillium* sp. MWZ14-4, which was isolated from an unidentified sponge (collected from the South China Sea), yielded 10 new fungal metabolites, including three hydroisocoumarins, penicimarins A–C (**692**–**694**), three isocoumarins, penicimarins D–F (**695**–**697**), and four benzofurans, penicifurans A–D (**698**–**701**) (Figure 42). Penicifuran A (**698**) showed moderate inhibitory activity against *Staphylococcus albus* with an MIC value of 3.13 μM and weak activity against *B. cereus* [240]. Two new tetracenedione derivatives, nocatriones A–B (**702**–**703**) (Figure 42), were isolated from the actinomycete *Nocardiopsis* sp. KMF-002, which was previously isolated from an unidentified sponge (Korea). Compounds **702**–**703** may show antiphotoaging activity in UVB-irradiated models [241].

Xylarianaphthol-1, a novel dinaphthofuran derivative (**704**) (Figure 42), was isolated from fungus of order Xylariales strain 05FI52, which was isolated from the unidentified marine sponge collected in Indonesia. Compound **704** activated the p21 promoter, which was stably transfected in MG63 cells dose-dependently [242]. Three new secondary metabolites, amycofuran (**705**), amycocyclopiazonic acid (**706**), and amycolactam (**707**) (Figure 42) were isolated from the sponge-associated rare actinomycete *Amycolatopsis* sp. isolated from a sponge sample gathered from Micronesia. Amycolactam (**707**) displayed significant to moderate cytotoxicity against the gastric cancer cell line SNU638, the colon cancer cell line HCT116, A546, K562, and SK-HEP1 with IC_50_ values of 0.8, 2.0, 13.7, 9.6, and 8.3 μM, respectively [243]. Chemical study of the marine-derived fungus *Penicillium adametzioides* AS-53 from an unidentified marine sponge (collected from Hainan, China) afforded a new spiroquinazoline derivative, N-Formyllapatin A (**708**), two new bisthiodiketopiperazine derivatives, adametizines A–B (**709**–**710**), two new acorane sesquiterpenes, adametacorenols A–B (**711**–**712**), a new dithiodiketopiperazine derivative, peniciadametizine A (**713**), and a highly oxygenated new analogue, peniciadametizine B (**714**) (Figure 42). Compound **709** exhibited lethality against brine shrimp (Artemia salina) with an LD_50_ value of 4.8 μM and moderate antimicrobial potency against several microbes with MIC values ranging from 8 to 32 μg/ mL, respectively, whereas compound **710** only showed weak activity against *S. aureus* (MIC, 64 μg/mL). Compound **712** showed significant selective activity against the NCI-H446 cell line (IC_50_ = 5.0 μM). Compounds **713** and **714** showed inhibitory activity against the pathogenic fungus *Alternaria brassicae* with MIC values of 4.0 and 32.0 μg/mL, respectively [244,245,246]. Chemical examination of a marine fungus *Aspergillus terreus* MXH-23, isolated from an unidentified sponge (collected from Guangdong Province, China), yielded a new butyrolactone derivative, namely butyrolactone VIII (**715**) (Figure 42). Derivative **715** did not show antiviral activity [247].

A new (**716**) sesquiterpenoid and two new (**717**–**718**) (Figure 43) xanthone derivatives were isolated from the fungus *Stachybotry* sp. HH1 ZDDS1F1-2, which was isolated from an unidentified sponge (Xisha Island, China). Compounds **717**–**718** exhibited significant inhibitory activity against cyclooxygenase (COX-2) with IC_50_ values of 10.6 and 8.9 μM, respectively. Besides, compound **718** displayed activities against intestinal virus EV71 with IC_50_ values of 30.1 μM [248]. The examination of the sponge-associated *Micromonospora* sp. NPS2077 isolated from an unidentified marine sponge (collected at Uranouchi Bay, Japan) led to the isolation of a novel β-hydroxyl-δ-lactone compound, neomacquarimicin (**719**) (Figure 43), which exhibited no activities against the *Bacillus subtilis* and *Escherichia coli* [249]. One new bisabolane-type sesquiterpenoid, aspergillusene C (**720**) (Figure 43), was isolated from sponge-associated fungi, the fungal strain ZSDS1-F6, which was identified as *Aspergillus sydowii*; it was isolated from an unidentified marine sponge (Xisha Islands, China). No bioactivities were detected [250]. One new naphthalene derivative (**721**) (Figure 43) was from Fungus *Arthrinium* sp. ZSDS1-F3, which was isolated from an unidentified sponge (Xisha Islands, China) [251]. Chemical study of the marine-derived fungus *Verrucosispora* sp. FIM06054 from a marine sponge sample (the East China Sea) afforded a new compound, FW054-1 (**722**) (Figure 43). Compound **722** showed antiproliferative activity against human tumour cells CNE-2 with IC_50_ values of 6.88 μM [252].

Fractionation of the marine fungus *Aspergillus* sp. OUCMDZ-1583 from an unidentified sponge XD10410 (collected from the Xisha Islands, China) yielded 18 new compounds named aspergones A–Q (**723**–**739**) and 6-O-demethylmonocerin (**740**) (Figure 43). Compounds **723**, **724**, **727**, **732**, **733**, and **736**–**740** showed a-glucosidase inhibition with IC_50_ values of 2.36, 1.65, 1.30, 2.37, 2.70, 1.36, 1.54, 2.21, 2.26, and 0.027 mM, respectively. Besides, compound **740** showed anti-H1N1 activity against the influenza A virus with IC_50_ values of 172.4 and 175.5 μM, respectively (with ribavirin as the positive control; IC_50_, 137.3 μM) [253]. Chemical examination of a marine-derived fungus *Penicillium chrysogenum* SYP-F-2720 from an unidentified sponge (collected off the North Sea coast, China) afforded a novel benzoic acid (**741**) (Figure 43). When administered at 100 mg/kg, compound **741** displayed more significant anti-inflammatory and analgesic activities than aspirin; however, it did not have an ulcerogenic effect [254]. Study of the sponge-derived fungus *Talaromyces minioluteus* PILE 14-5 derived from an unidentified marine sponge (collected in Thailand) afforded four new sesquiterpene lactones (**742**–**745**) (Figure 43). Compounds **742** and **745** exhibited weak cytotoxic activity with IC_50_ values ranging from 50.6 to 193.3 μM [255]. The investigation of an unidentified marine sponge derived fungus, *Aspergillus similanensis* KUFA 0013, yielded a new isocoumarin derivative, similanpyrone C (**746**), a new cyclohexapeptide, similanamide (**747**), and a new pyripyropene derivative, named pyripyropene T (**748**) (Figure 43). Only compound **747** exhibited weak activity against the MCF-7 (breast adenocarcinoma), NCI-H460 (non-small cell lung cancer), and A373 (melanoma) cell lines, and neither of them showed antibacterial activity [256].

The fungus *Corynespora cassiicola* XS-200900I7, isolated from an unidentified sponge XS-2009001 (collected from the Xisha Islands, China), yielded 12 new chromone derivatives, corynechromones A–L (**749**–**760**), two new naphthalenones, corynenones A–B (**761**–**762**), and one new depsidone, corynesidone E (**763**) (Figure 44). The bioassay was disappointing [257,258]. S-Bridged pyranonaphthoquinone dimers, naquihexcins A–B (**764**–**765**) (Figure 44), were metabolized by a sponge-derived *Streptomyces* sp. HDN-10-293 isolated from the marine sponge. Compound **764** could inhibit the proliferation of the adriamycin-resistant human breast cancer cell line MCF-7 ADM with an IC_50_ value of 16.1 μM (adriamycin as positive control, IC_50_ > 20.0 μM) [259]. Two new structurally unique compounds bearing a nitrogen and sulfur-containing tricyclic ring system, ulbactin F–G (**766**–**767**) (Figure 44), were metabolized by a sponge-derived *Brevibacillus* sp. strain TP-B0800 associated with an unidentified sponge collected in Iwate, Japan. Both compounds **766** and **767** inhibit the migration of epidermoid carcinoma A431 cells at non-cytotoxic concentrations with IC_50_ values of 6.4 and 6.1 μM, respectively [260]. Chemical investigation of a marine-derived fungus *Aspergillus aureolatus* HDN14-107 from an unidentified sponge (collected at Xisha Islands, China) led to the isolation of three new meroterpenoids, named austalides S–U (**768**–**770**) (Figure 44). Compound **770** exhibited anti-H1N1 activities against influenza virus A with IC_50_ values of 90 µM [261].

Chemical analysis of the recombinant *Streptomyces albus* PVA94-07 strain derived from an unidentified sponge afforded two new deferoxamine analogues, compounds **771**–**772** (Figure 45). Compounds **771**–**772** exhibited no cytotoxic activity on these tested cell lines [262]. Chemical analysis of the strain *Aspergillus* sp. SCSIO XWS03F03 derived from a sponge (collected in Guangdong Province, China) afforded two new polyketides, aspergchromones A–B (**773**–**774**) (Figure 45). Compounds **773**–**774** showed no antimicrobial activity against *Staphyloccocus aureus* and *Mycobacterium tuberculosis* [263].

## 3. Statistical Analysis of Sponge-Derived Microorganisms

### 3.1. Geographical Distribution of Sponge-Derived Microorganisms

As what we described above, temperate and tropical sea areas have become the main regions with sponge-derived microorganisms related to natural product chemistry. Figure 46 presented a visible-direct sketch map for geographical distribution of sponge-derived microorganisms. The studied species of microbial communities associated with sponges has so far mostly been focused on the Pacific coasts, including the South China Sea, Sea of Japan, Gulf of Thailand, Korean Peninsula, Indonesian Islands, Eastern China Sea, and the Great Barrier Reef, Australia. Of these, the South China Sea and the Sea of Japan are the most active hotspots, and the followings are the Mediterranean Sea, the West Atlantic Ocean, including the Gulf of Mexico and the Caribbean Island, the North Sea, the Black Sea, and then the Red Sea.

### 3.2. Biodiversity of Microbial-Associated Sponge Hosts

The statistical data shows that all the sponge hosts associated with microbes were distributed in the classes of Calcarea, Demospongiae, and unidentified sponges. The class Demospongiae sponges accounts for absolute majority at 77%, the class Calcarea and others sponges account for 1% and 22%, respectively. In the class Demospongiae, the order Haplosclerida provides the most sponge species accounting for 26.75% of the totality, followed by the orders Suberitida, Axinellida, Dictyoceratida, Tetractinellida, and Poecilosclerida etc. (see Figure 47).

### 3.3. Biodiversity of Sponge-Derived Microorganisms

In the process of long-term co-evolution, there may be some sponge-derived microbes with wonderful symbiotic relationships in the sponge ecosystem, which cover fungi, bacteria, actinomycetes, cyanobacteria, and archaea. With no doubt, fungi, bacteria and actinomycetes are the main producers of prolific natural products with therapeutic effects among the sponge-derived microbes (see Figure 48), and the microbes appear to be distributed randomly in the investigated host sponges. Fungi are an important component of sponge-derived microbes—up to 73%—and more than 55 genera strains have been cultured. The fungi studied mainly belonged to the genera *Aspergillus* and *Penicillium*, followed by *Trichoderma*, *Acremonium*, *Arthrinium*, and *Talaromyces*. The genera *Aspergillus* and *Penicillium* obtained from many different sponge species take a percentage of 25% of the total microbes reported, the majority of which displayed wonderful chemistry diversity. Actinomycetes and bacteria account for 16% and 11% of sponge-derived microbes, respectively, which offered many novel and unique compounds. The actinomycetes studied were mainly derived from the genera *Streptomyces*, and the bacteria were mainly derived from the genera *Pseudomonas*, *Pseudoalteromonas*, and *Bacillus* (see Figure 48).

### 3.4. Chemical Diversity and Bioactive Diversity of Sponge-Derived Microorganisms

A total of 774 new compounds, which were assorted into nine types including terpenes, alkaloids, peptides, aromatics, meroterpenoids, macrolides, polyketides, steroids, and miscellaneous (Figure 49), were reported in the 253 studies in the literature. Among them, the number of the aromatics and the alkaloids takes a percentage of 45%, which suggests that we should pay more attention to the study of these two kinds of compounds.

Natural products isolated from sponge-derived microorganisms have interesting pharmaceutical activities such as cytotoxicity, antioxidant, antifungal, antiviral, and antibacterial activities. Some novel compounds showed significant cytotoxic activities in the nM levels, such as two new indolocarbazole alkaloids [13] and linear pentadecapeptides efrapeptin Eα [29]. Some new compounds exhibited activities stronger than the positive controls. Engyodontochone A–B exhibited inhibitory activity that was 10-fold stronger than chloramphenicol against MRSA [69]. A new pyronepolyene C-glucoside exhibited significant inhibitory effects in the cytopathic effect inhibition assay with an IC_50_ value of 91.5 μM (ribavirin as a positive control, IC_50_ 114.8 μM) [86]. Truncateol M showed significant inhibitory effects with an IC_50_ value of 8.8 μM against H1N1 virus (oseltamivir as a positive control, IC_50_ 46.5 μM) [101]. Dankastatin C (**687**) showed potent cell growth inhibitory activity against the murine P388 cell line with an ED_50_ value of 57 ng/mL (5-fluorouracil as a positive control, ED_50_, 78 ng/mL) [236]. As we reported, the diketopiperazine alkaloid amauromine shows affinity for cannabinoid CB1 receptors, which may have potential as a lead structure for drug development [63]. Another noteworthy fact is the discovery of two new compounds—emindole SB betamannoside and 27-O-methylasporyzin C—that may serve as lead structures for the development of GPR18- and CB receptor-blocking drugs [88]. Compounds isolated from the sponge-derived microorganisms have various activities, and in addition to the above, they also exhibit anti-cholesterol activity [112], antiplasmodial activity [81,134], neuroprotective effects [26,178], and so on. The reported bioactivities are limited by many factors and they are not comprehensive, which suggests that these natural compounds should be screened on a wider variety of bioassays in order to unveil their full potential.

## 4. Conclusions

A total of 774 new compounds from sponge-derived microorganisms, covering the last two decades from 1998 to 2017, were reviewed. These new compounds presented abundant chemical diversity except for the well-known types such as terpenes, alkaloids, peptides, aromatics, meroterpenoids, macrolides, polyketides, steroids, and so on.

The total amount of new compounds from sponge-derived microorganisms has increased rapidly and has not yet reached a climax, especially in the last five years (see Figure 50). Among the compounds of the sponge-derived microorganisms obtained, more than 42% of the compounds have detected activities; however, most of them only carried out preliminary active in vitro test experiments.

We should make more efforts to study the pharmacodynamic relationships and pharmacological effects of promising compounds, and conduct clinical trials to complete the drug-like evaluation of the compound. The study on the chemical constituents of sponges and their co-existing microorganisms will promote the study of the relationship between sponges and their co-existing microorganisms, develop and utilize medicinal resources, and systematically study the diversity of biodiversity and ecosystems. Given the structural differences and biodiversity of compounds derived from sponge symbiotic microorganisms, we believe that there are more resources waiting to be mined.

## Figures and Tables

**Figure 1 molecules-25-00853-f001:**
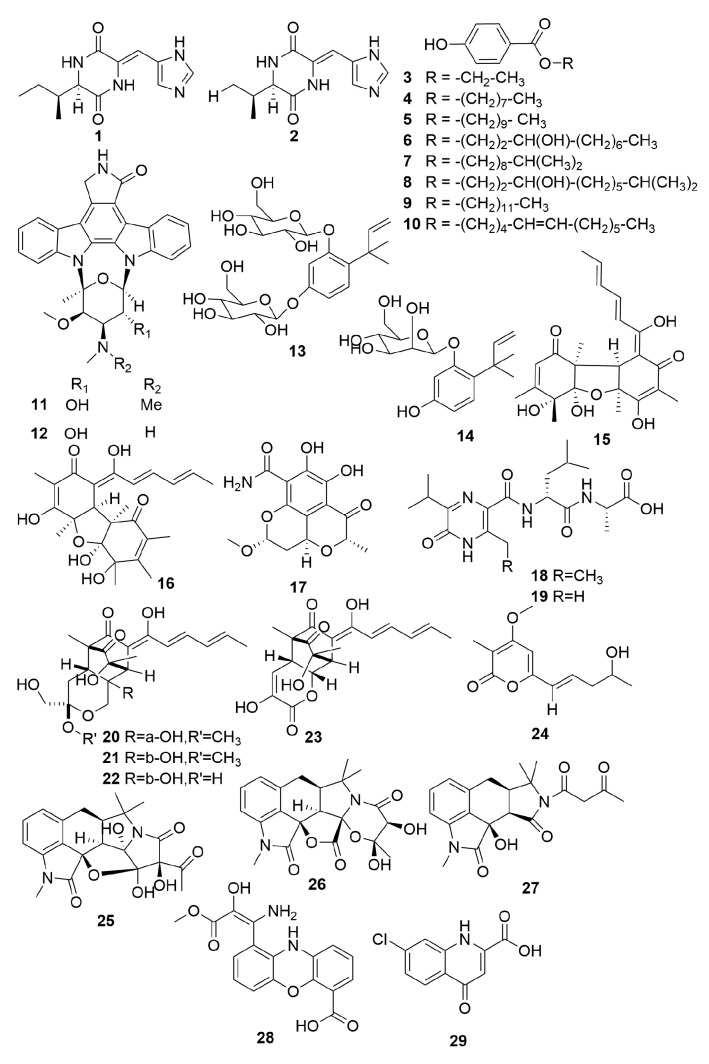
Structures of compounds **1**–**29**.

**Figure 2 molecules-25-00853-f002:**
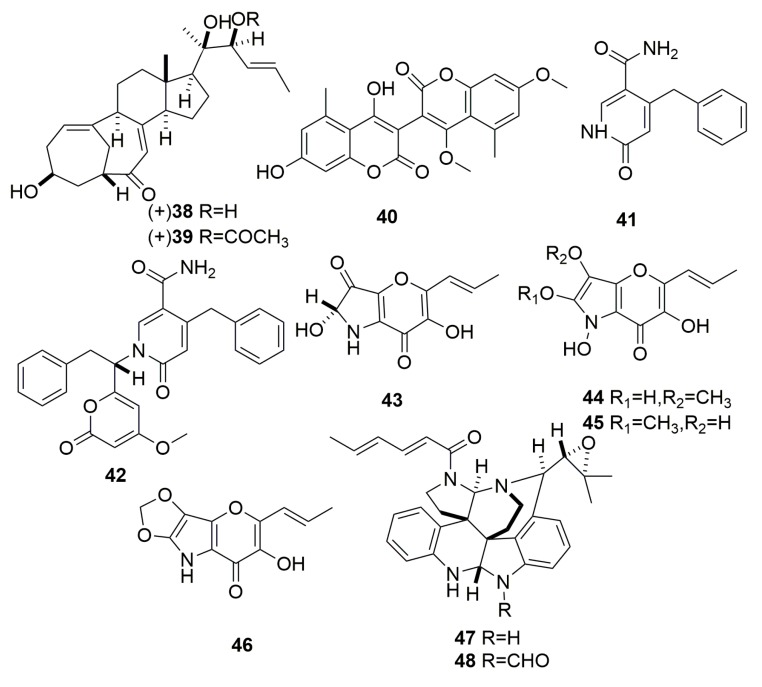
Chemical structures of new compounds **38**–**48**.

**Figure 3 molecules-25-00853-f003:**
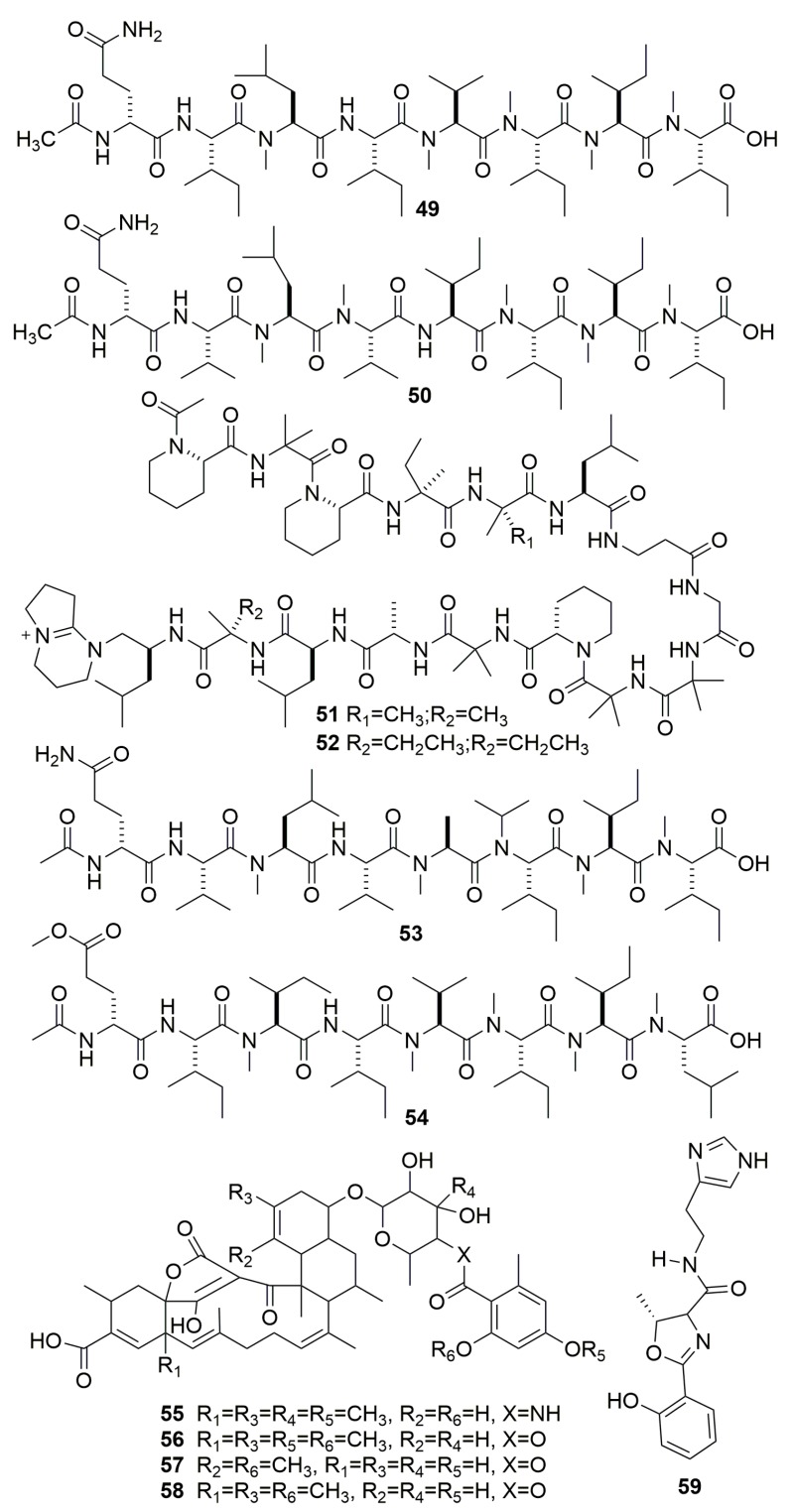
Chemical structures of diverse new molecules **49**–**59.**

**Figure 4 molecules-25-00853-f004:**
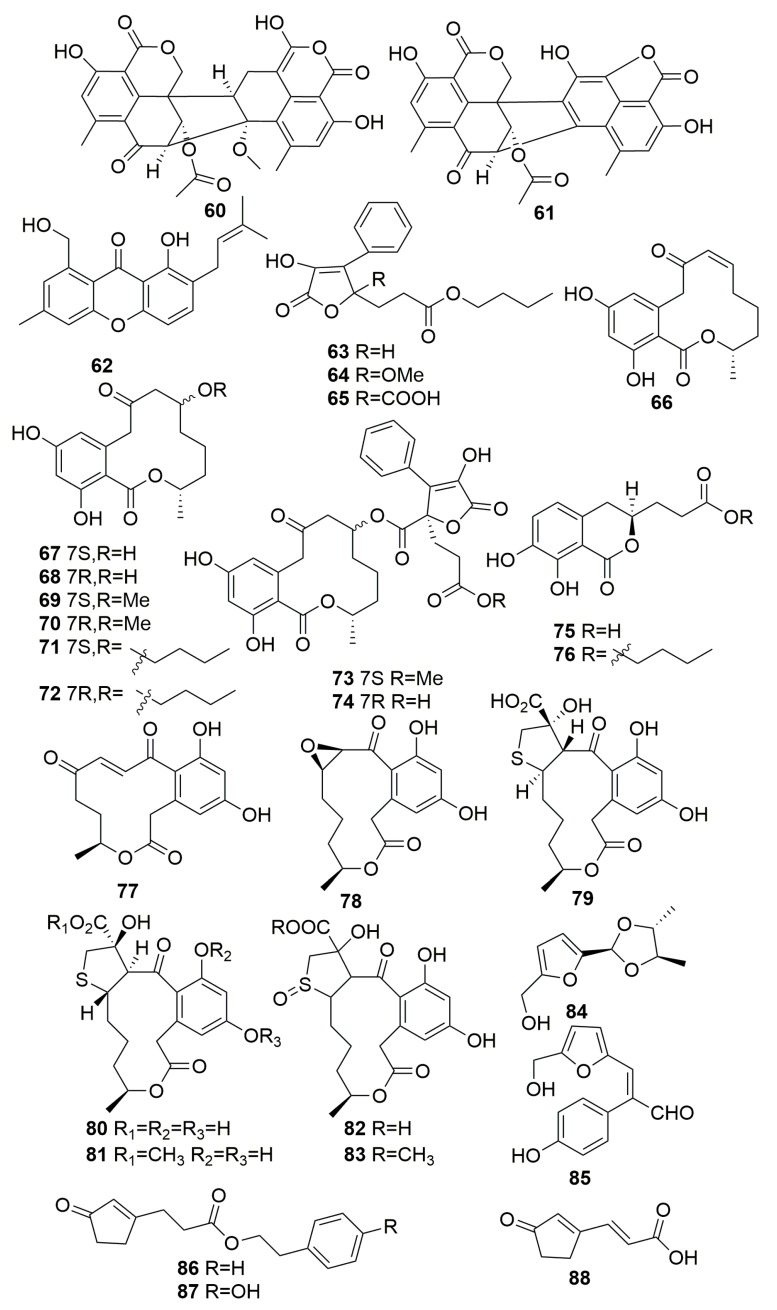
Chemical structures of diverse new molecules **60**–**88**.

**Figure 5 molecules-25-00853-f005:**
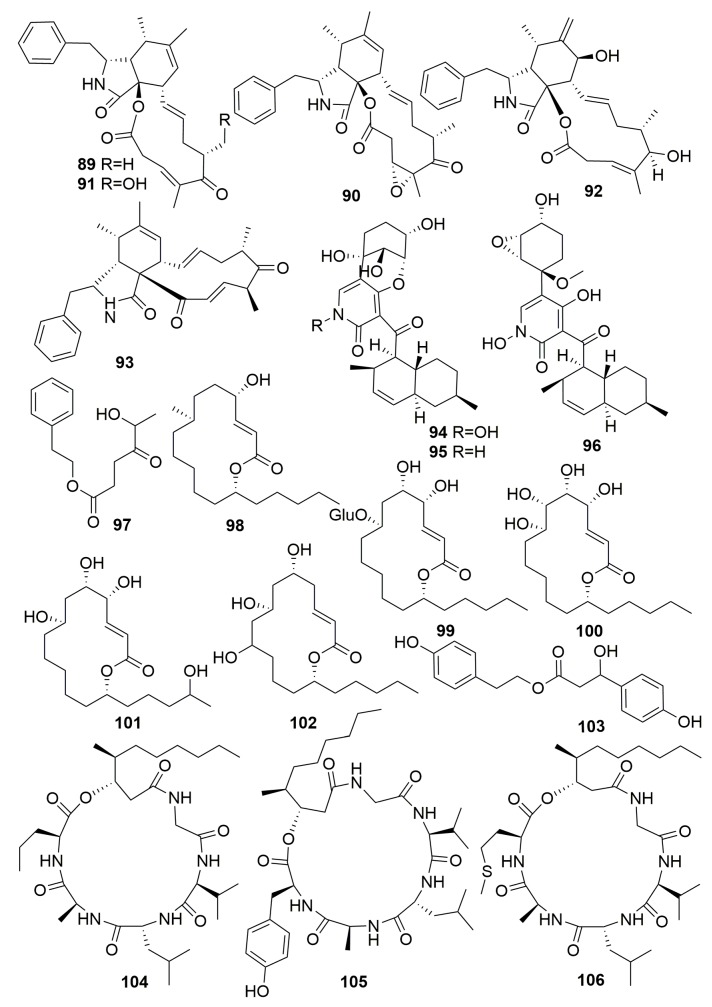
Chemical structures of compounds **89**–**106**.

**Figure 6 molecules-25-00853-f006:**
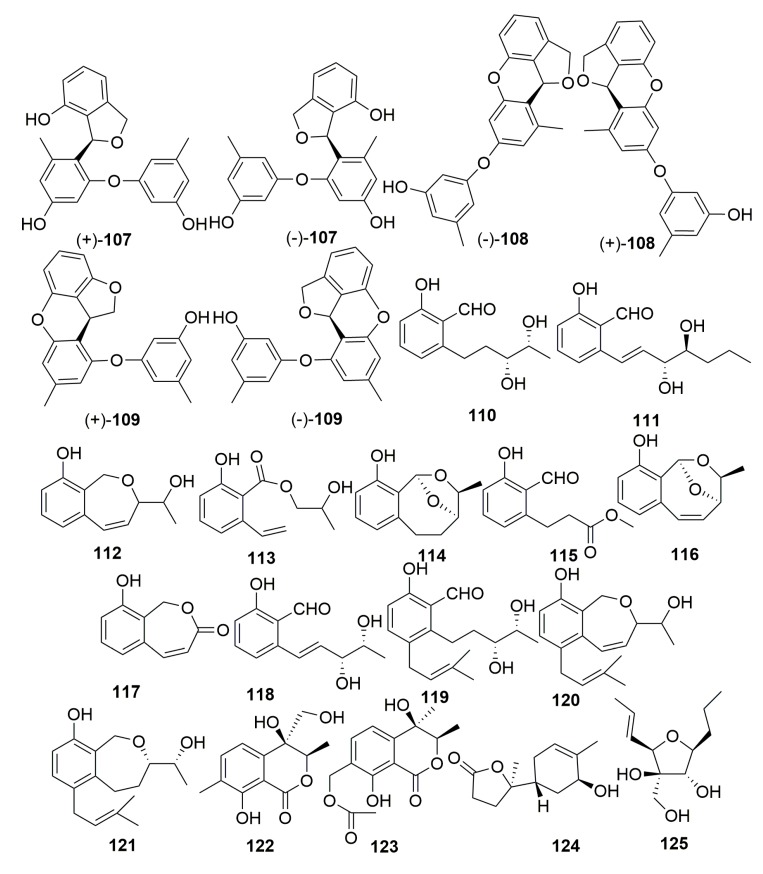
Chemical structures of new marine natural products **107**–**125**.

**Figure 7 molecules-25-00853-f007:**
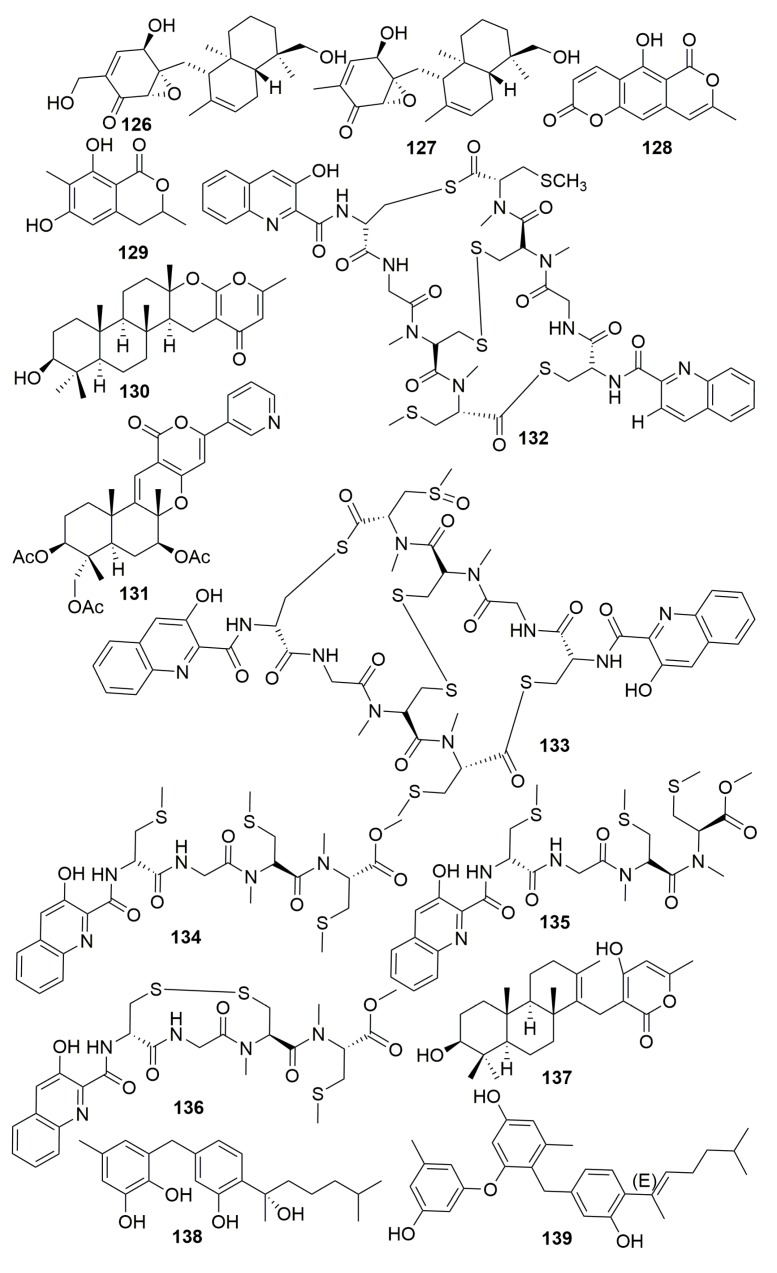
Chemical structures of compounds **126**–**139** derived from sponge-associated microbes.

**Figure 8 molecules-25-00853-f008:**
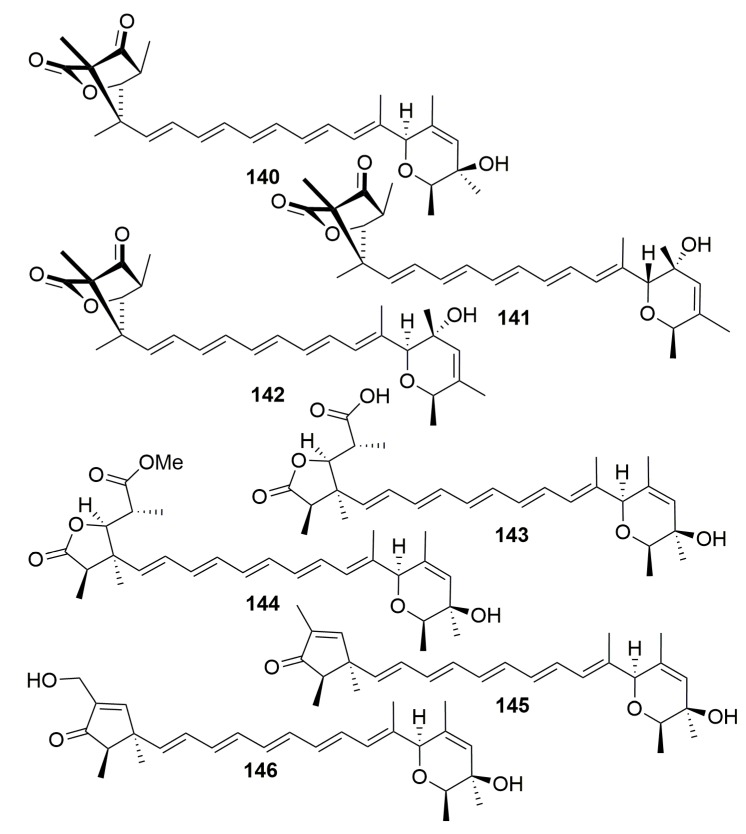
Chemical structures of seven new polyketides **140**–**146**.

**Figure 9 molecules-25-00853-f009:**
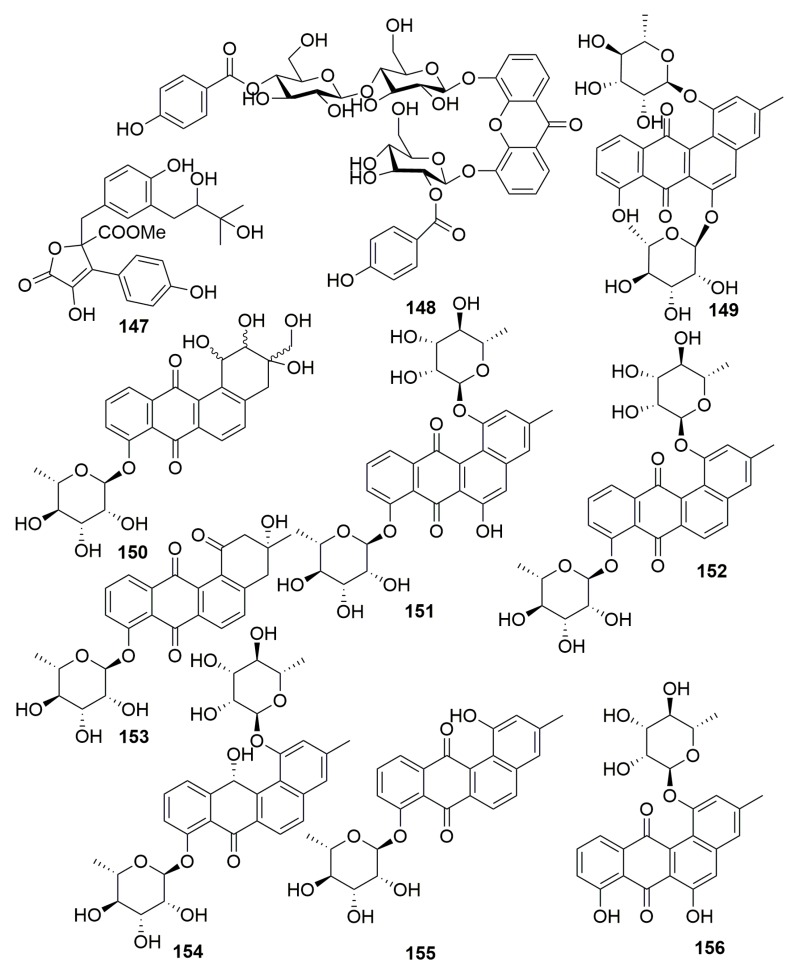
Chemical structures of new marine natural products **147**–**156**.

**Figure 10 molecules-25-00853-f010:**
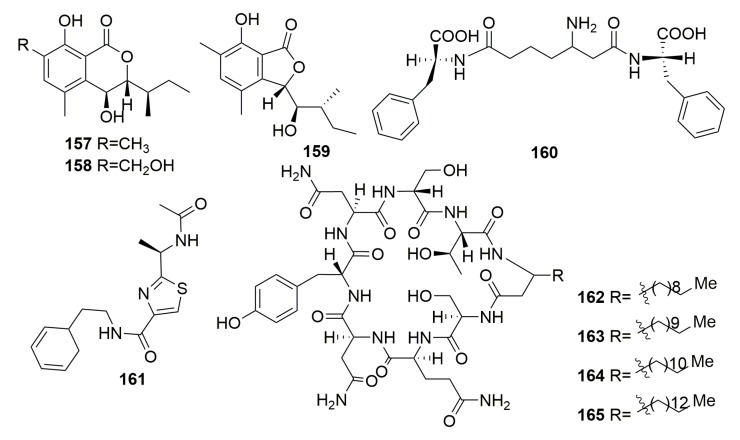
Chemical structures of compounds **157**–**165**.

**Figure 11 molecules-25-00853-f011:**
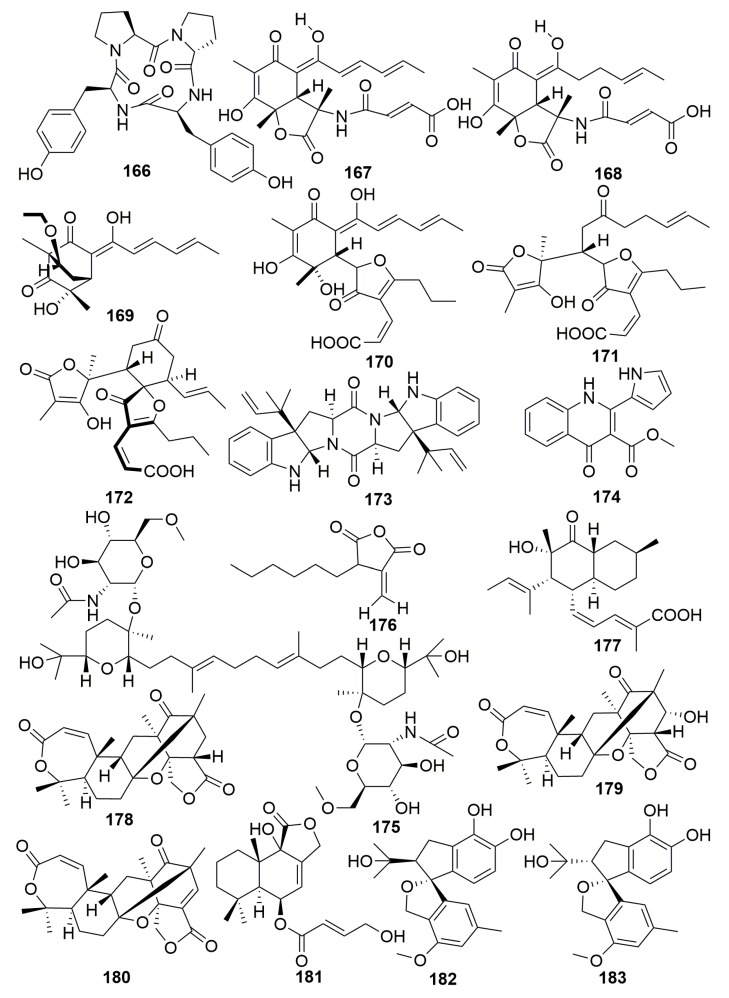
Chemical structures of compounds **166**–**183** produced by the fungus originated from the sponge Irciniidae.

**Figure 12 molecules-25-00853-f012:**
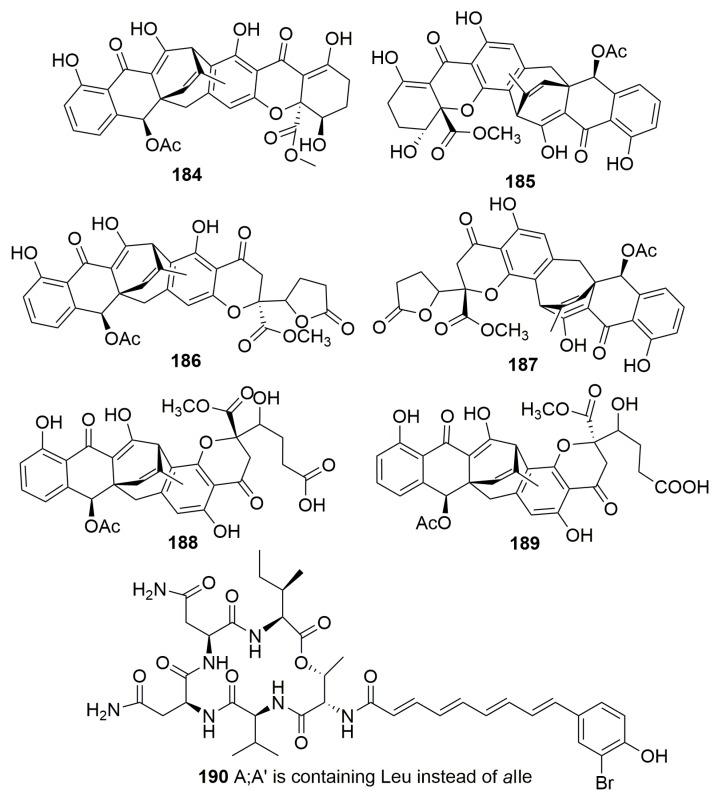
Structures of new marine natural products **184**–**190**.

**Figure 13 molecules-25-00853-f013:**
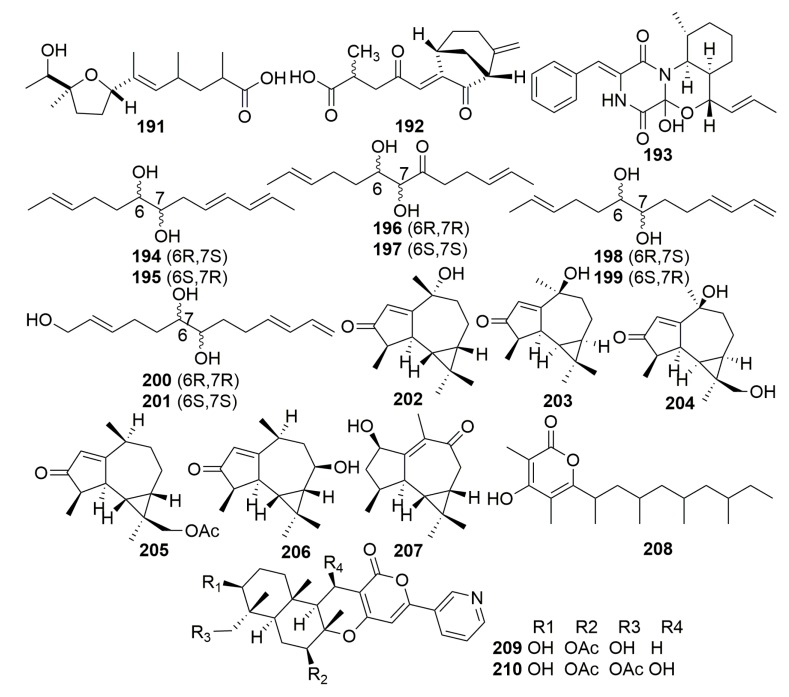
Structures of compounds **191**–**210** isolated from sponge-derived microorganisms.

**Figure 14 molecules-25-00853-f014:**
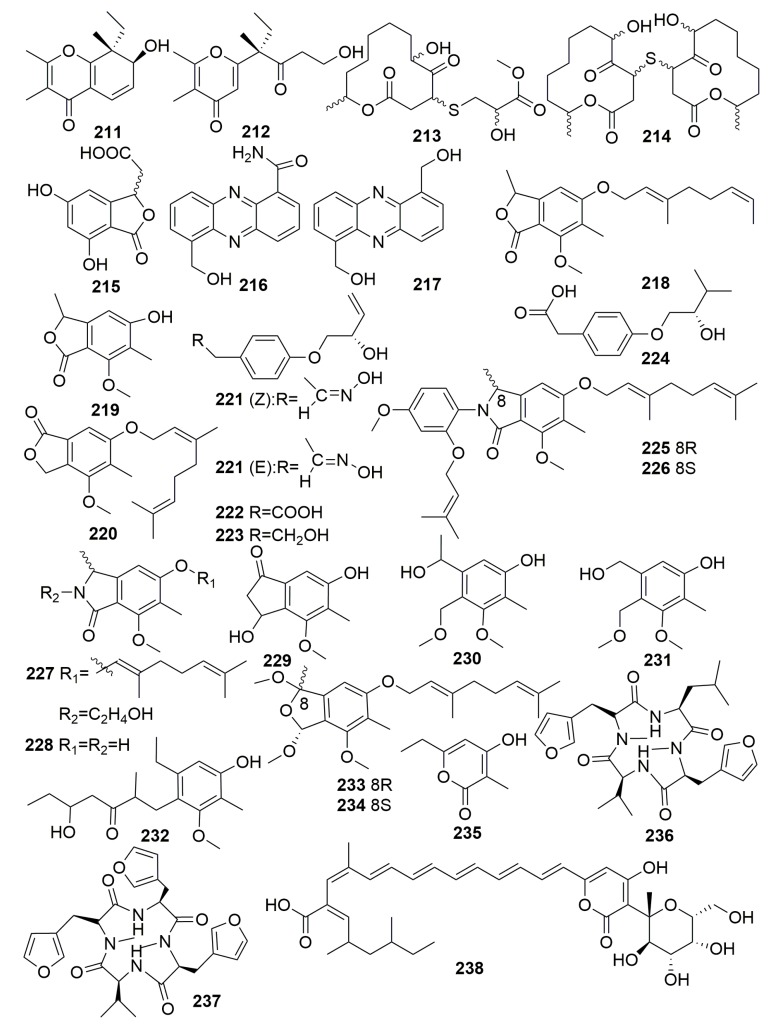
Chemical structures of compounds **211**–**238**.

**Figure 15 molecules-25-00853-f015:**
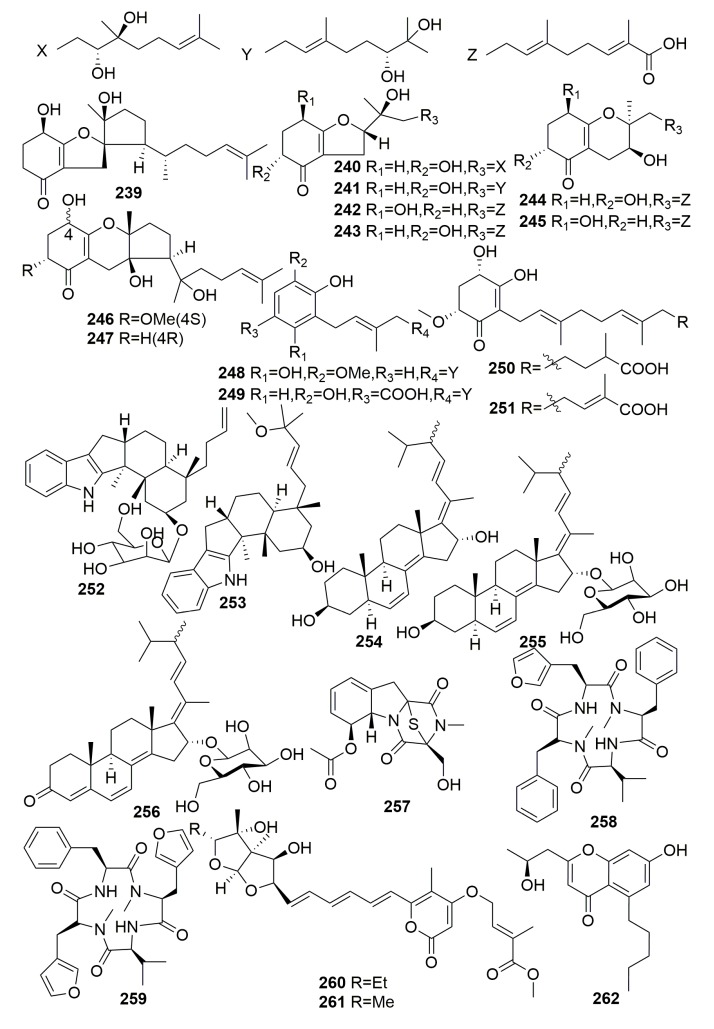
Structures of new marine natural products **239**–**262**.

**Figure 16 molecules-25-00853-f016:**
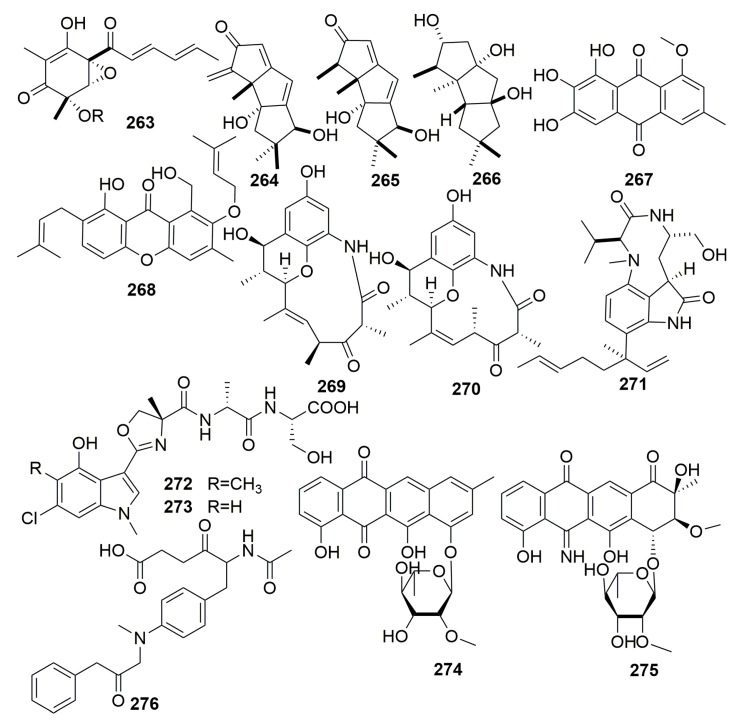
Chemical structures of compounds **263**–**276**.

**Figure 17 molecules-25-00853-f017:**
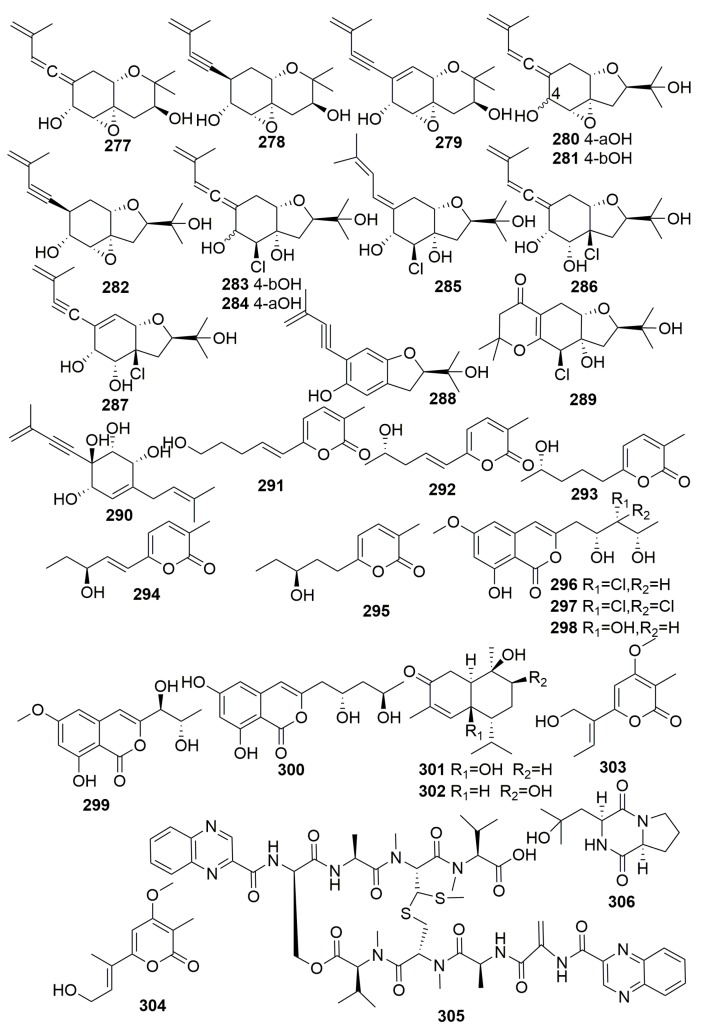
Chemical structures of new molecules **277**–**306**.

**Figure 18 molecules-25-00853-f018:**
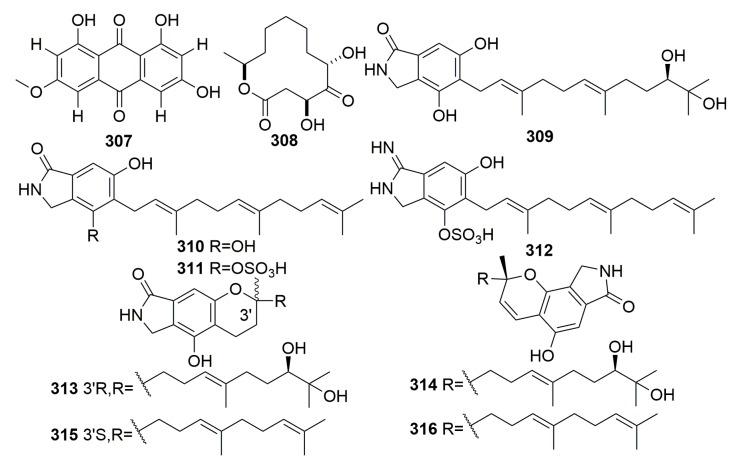
Chemical structures of compounds **307**–**316**.

**Figure 19 molecules-25-00853-f019:**
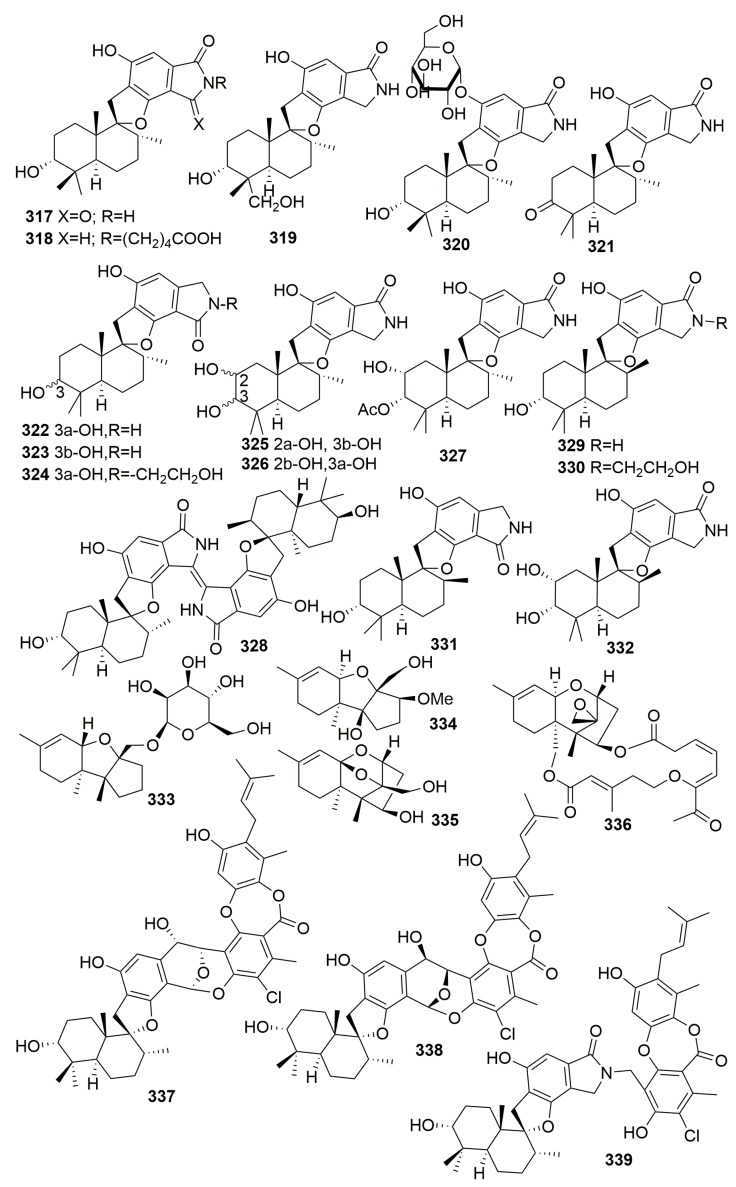
Chemical structures of molecules **317**–**339** isolated from the above fungus *Stachybotrys chartarum*.

**Figure 20 molecules-25-00853-f020:**
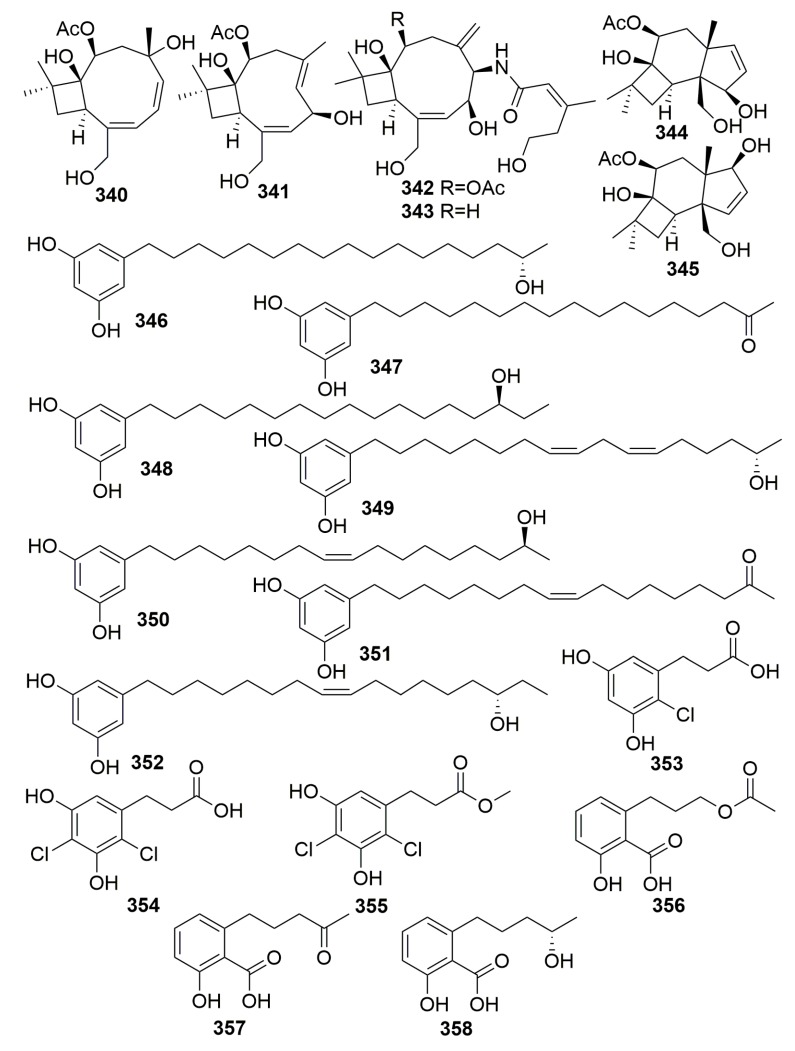
Chemical structures of diverse new molecules **340**–**358** isolated from microorganisms.

**Figure 21 molecules-25-00853-f021:**
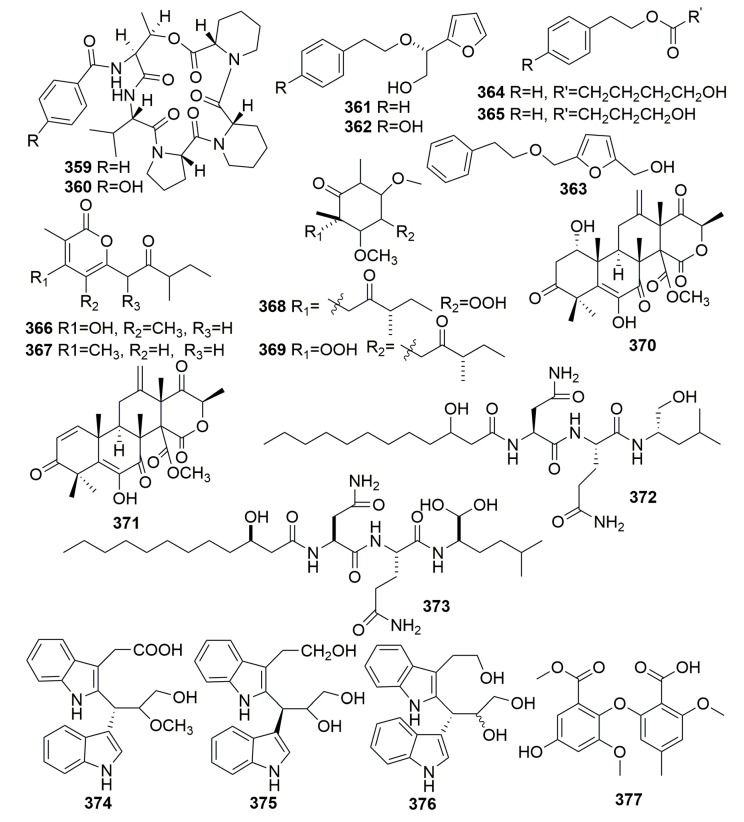
Chemical structures of diverse new molecules **359**–**377** isolated from sponge-derived microorganisms.

**Figure 22 molecules-25-00853-f022:**
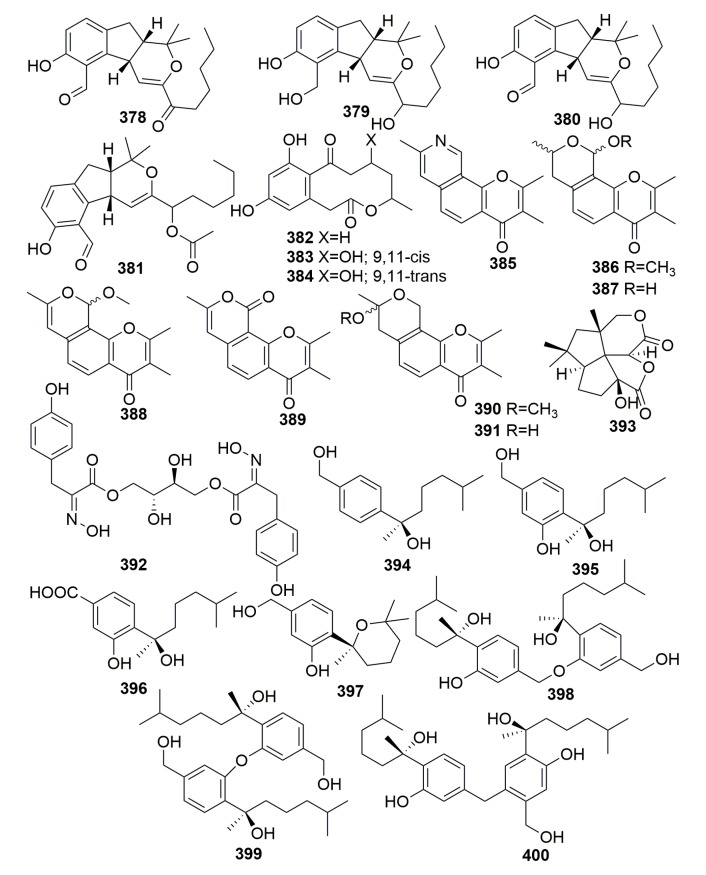
Chemical structures of new marine natural products **378**–**400** isolated from microorganisms.

**Figure 23 molecules-25-00853-f023:**
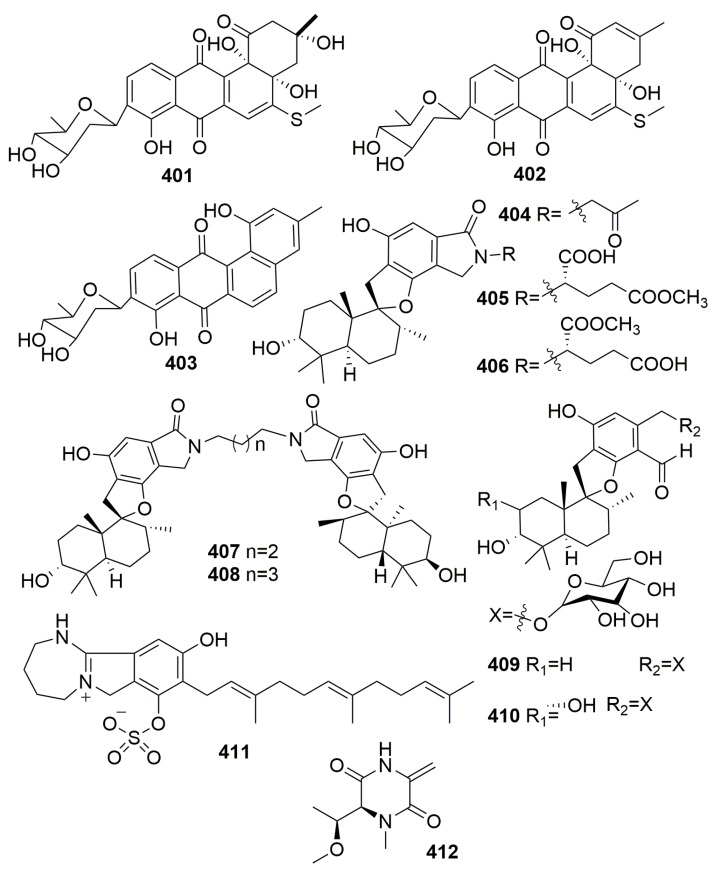
Chemical structures of diverse new molecules **401**–**412** isolated from sponge-derived microbes.

**Figure 24 molecules-25-00853-f024:**
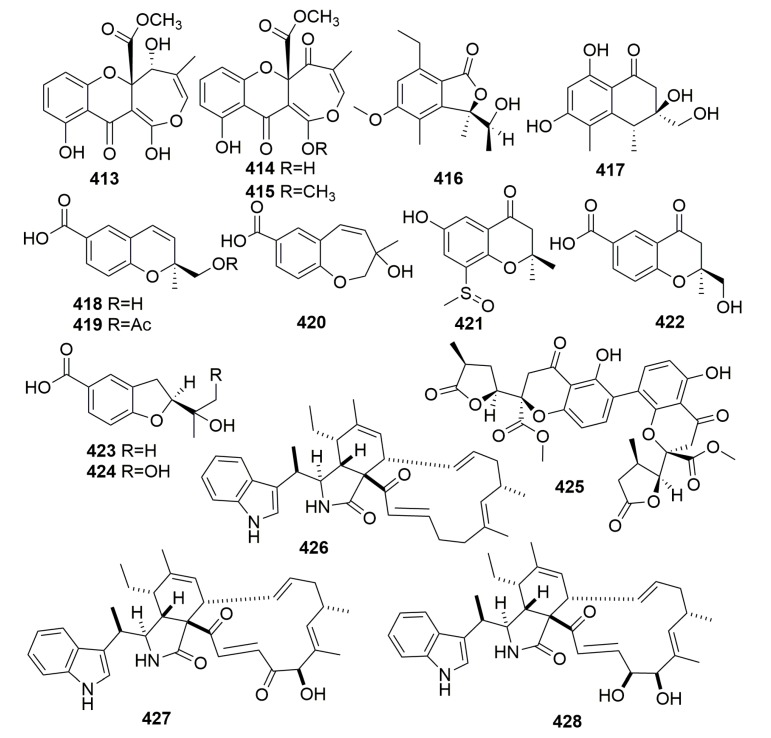
Structures of new marine natural products **413**–**428** derived from microbes associated with the sponge (Acarnidae and Microcionidae).

**Figure 25 molecules-25-00853-f025:**
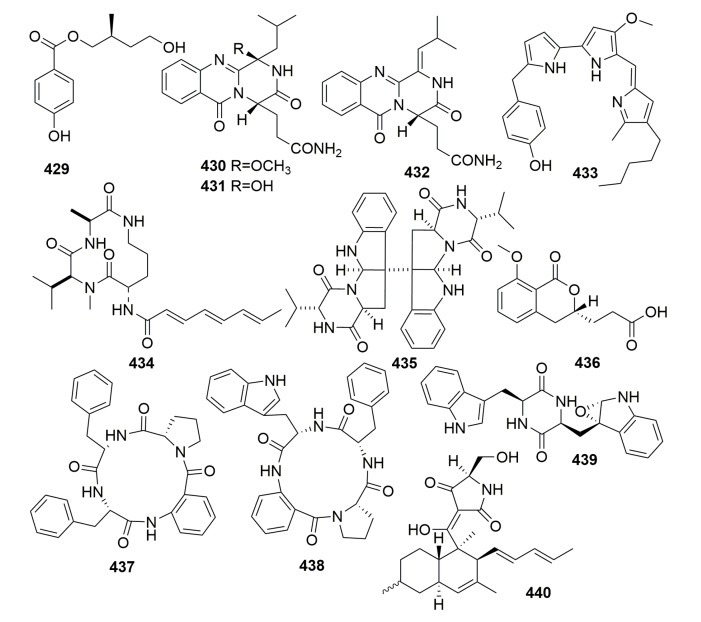
Chemical structures of new molecules **429**–**440**.

**Figure 26 molecules-25-00853-f026:**
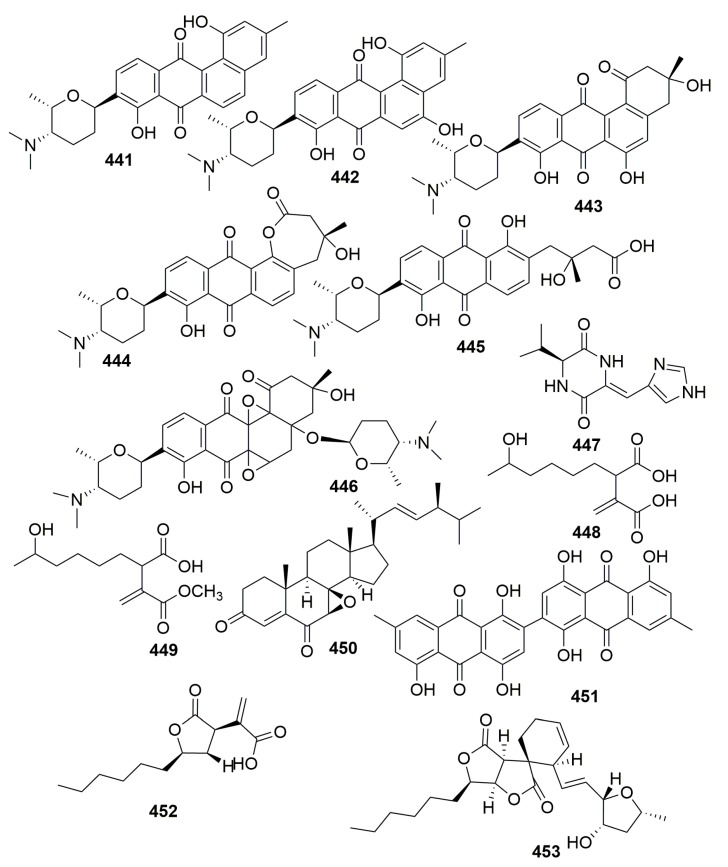
Chemical structures of new marine natural products **441**–**453**.

**Figure 27 molecules-25-00853-f027:**
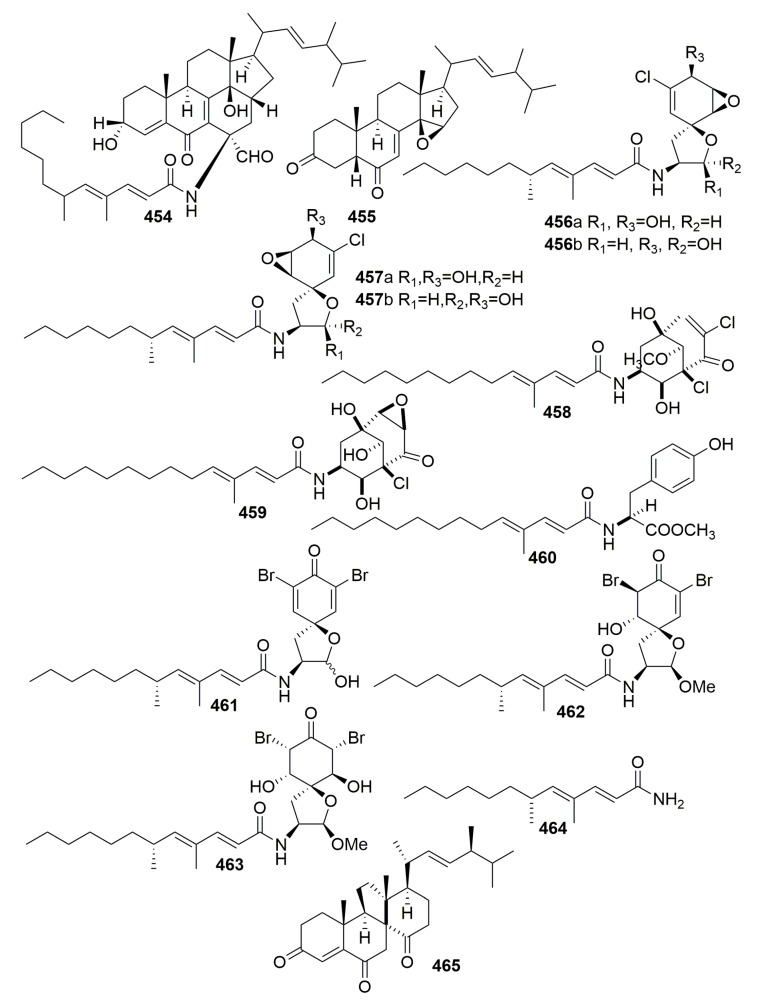
Chemical structures of new compounds **454**–**465**.

**Figure 28 molecules-25-00853-f028:**
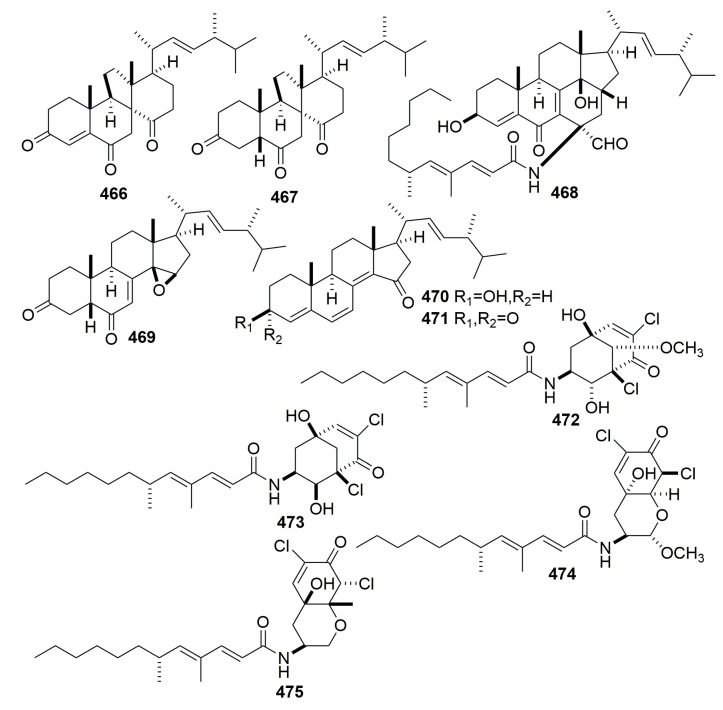
Chemical structures of new compounds **466**–**475**.

**Figure 29 molecules-25-00853-f029:**
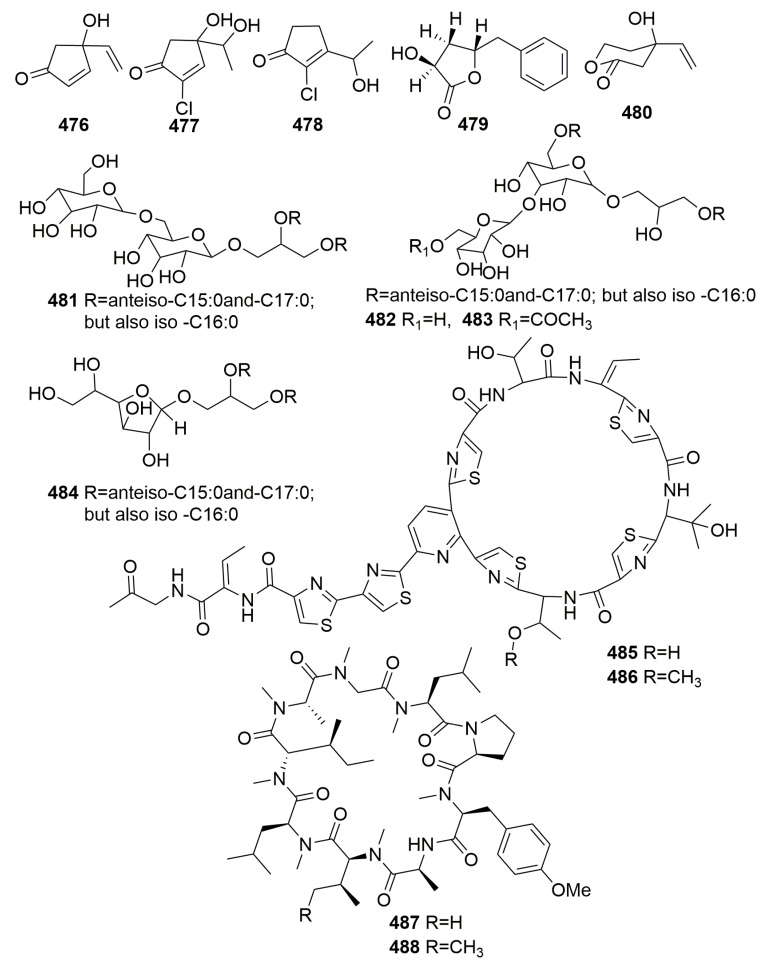
Structures of new molecules **476**–**488** isolated from the microorganisms.

**Figure 30 molecules-25-00853-f030:**
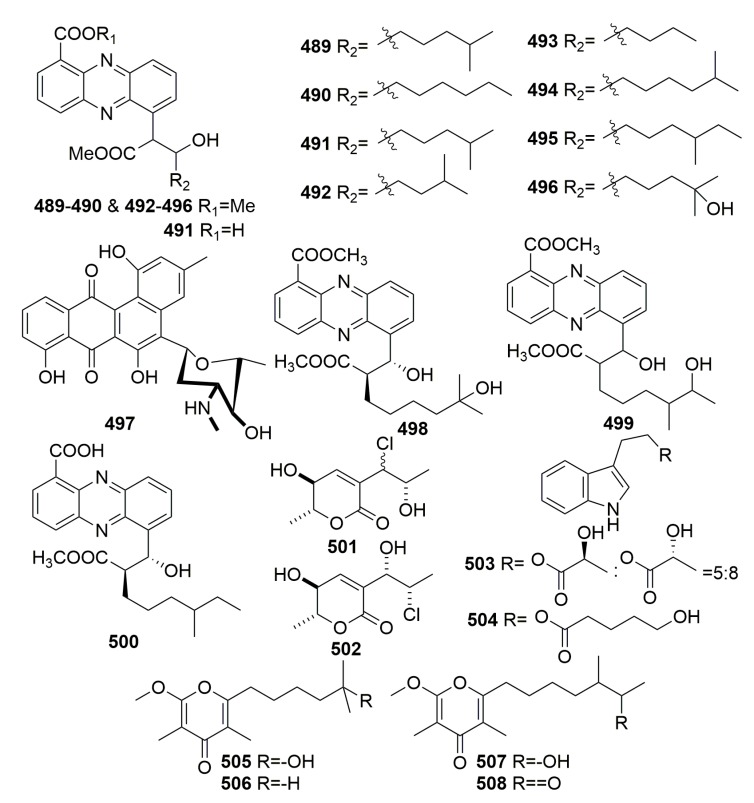
Chemical structures of compounds **489**–**508**.

**Figure 31 molecules-25-00853-f031:**
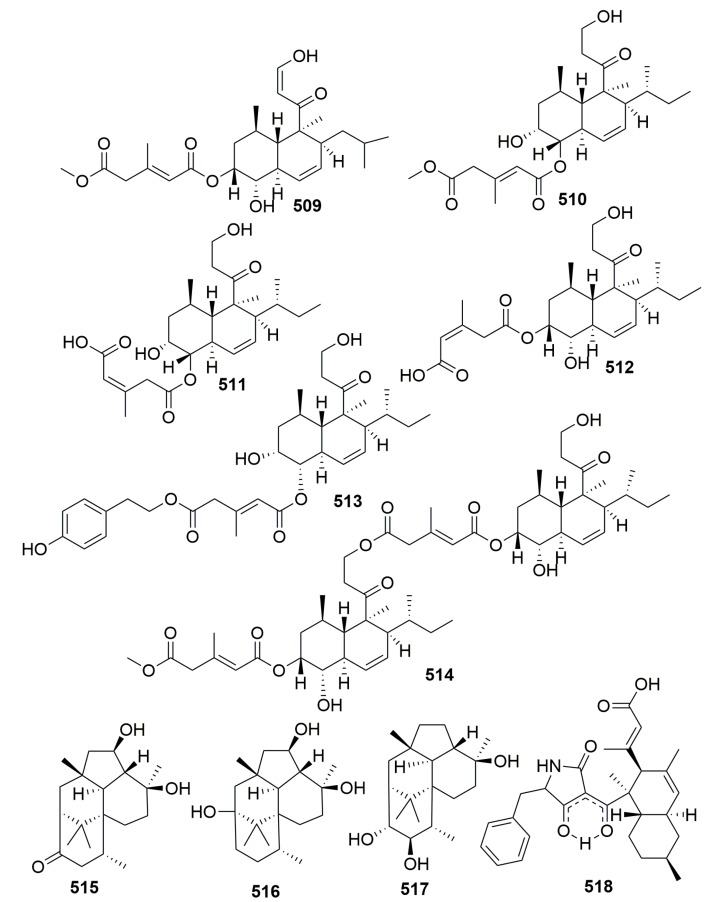
Structures of new marine natural products **509**–**518**.

**Figure 32 molecules-25-00853-f032:**
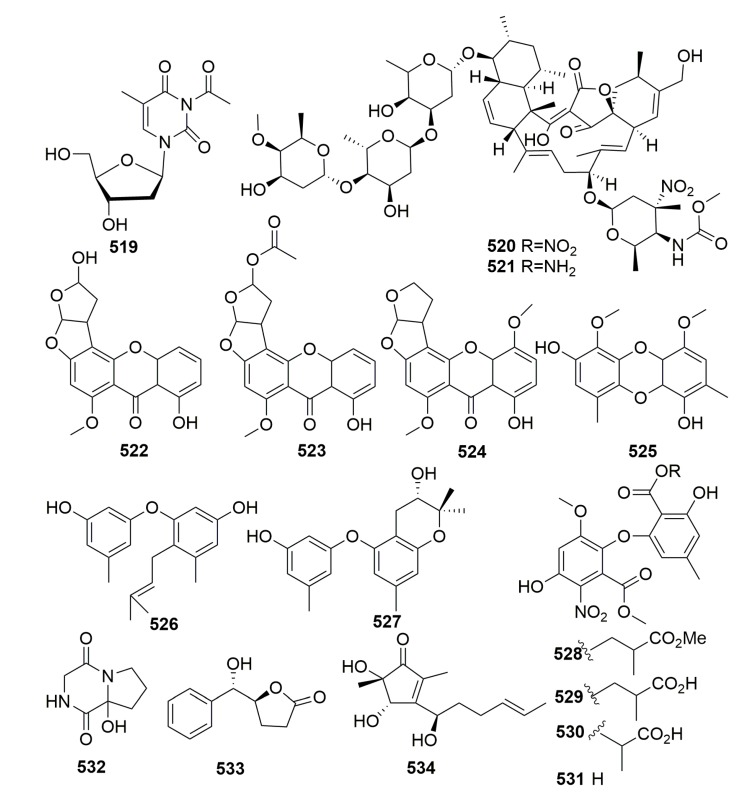
Chemical structures of diverse new molecules **519**–**534**.

**Figure 33 molecules-25-00853-f033:**
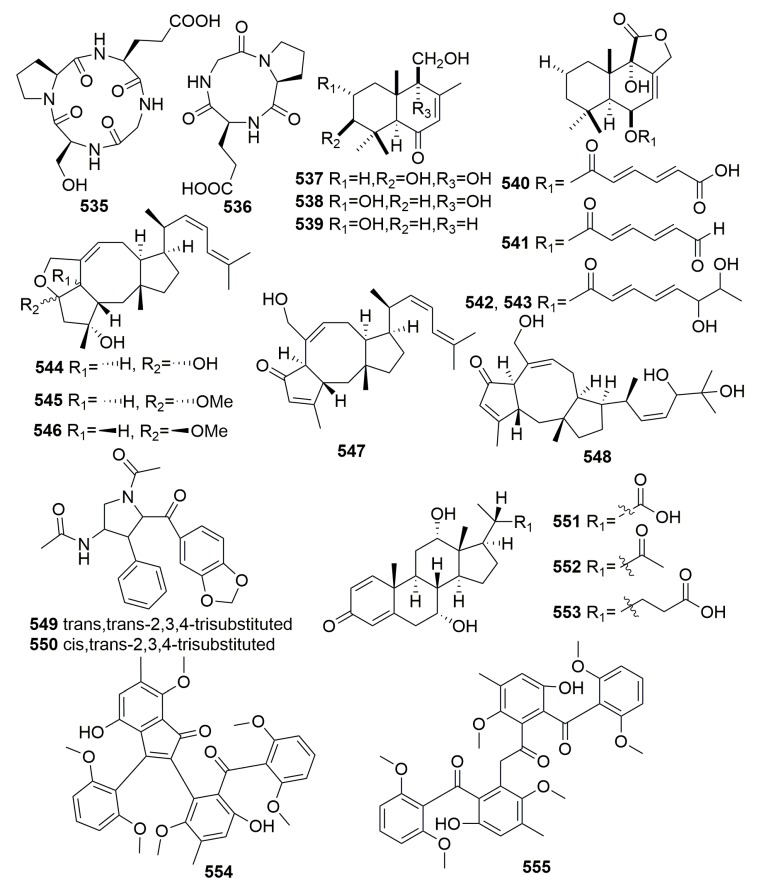
Chemical structures of compounds **535**–**555**.

**Figure 34 molecules-25-00853-f034:**
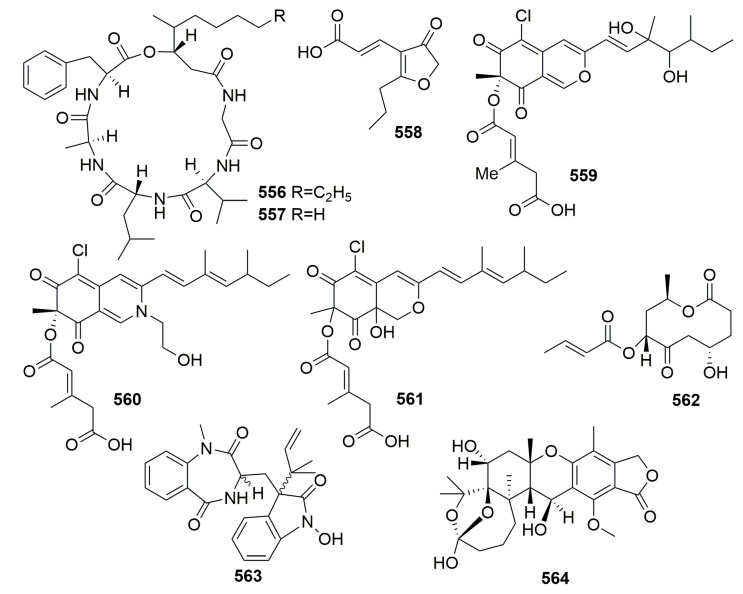
Structures of new marine natural products **556**–**564** derived from microbes associated with the sponge (Tethyidae).

**Figure 35 molecules-25-00853-f035:**
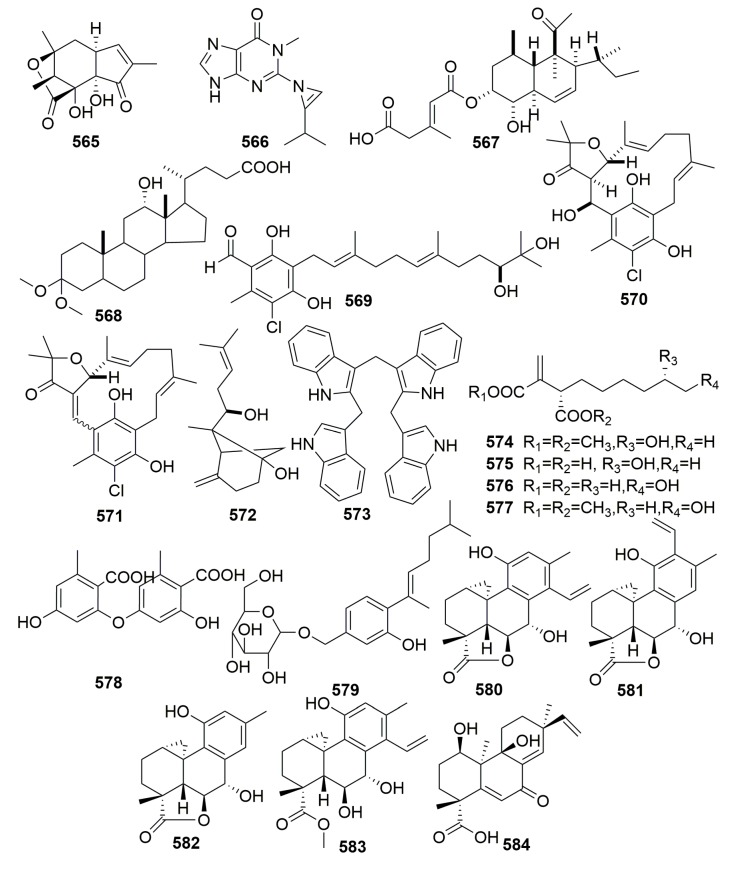
Structures of new compounds **565**–**584**.

**Figure 36 molecules-25-00853-f036:**
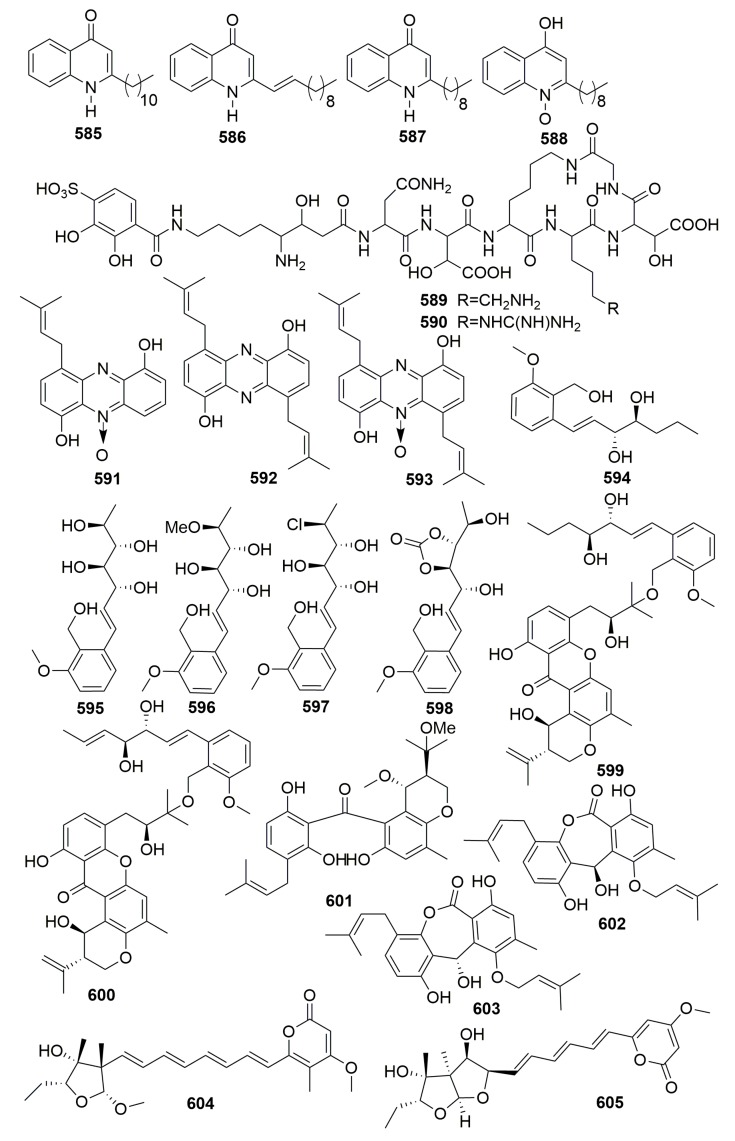
Structures of new marine natural products **585**–**605** derived from microbes associated with the sponge (Neopeltidae and Tetillidae).

**Figure 37 molecules-25-00853-f037:**
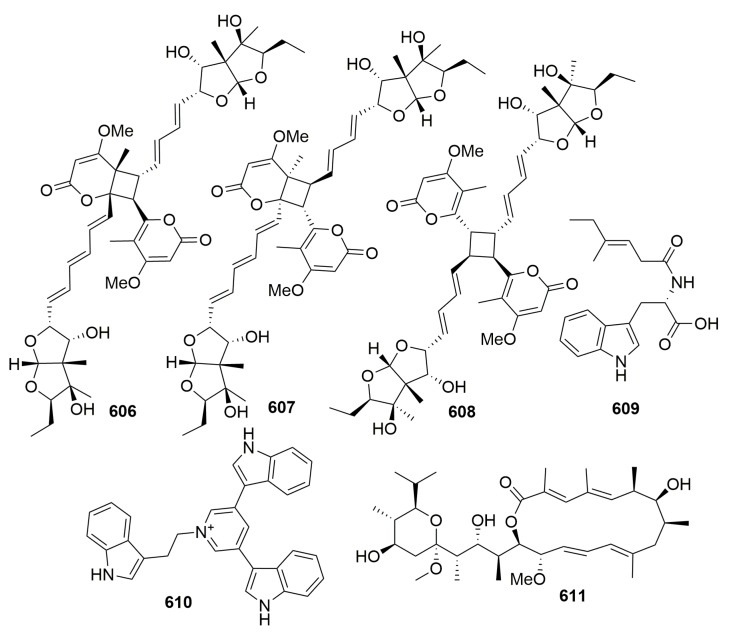
Structures of new marine natural products **606**–**611** derived from microbes associated with the sponge (Tetillidae and Theonellidae).

**Figure 38 molecules-25-00853-f038:**
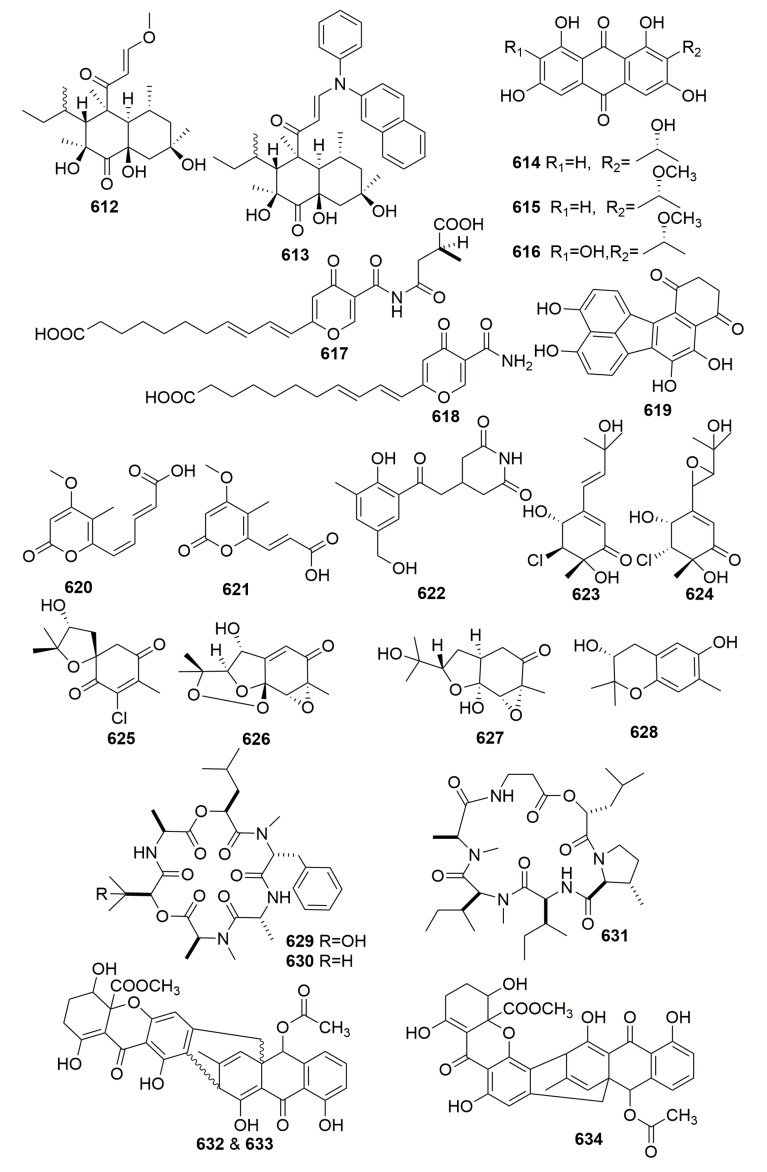
Chemical structures of compounds **612**–**634**.

**Figure 39 molecules-25-00853-f039:**
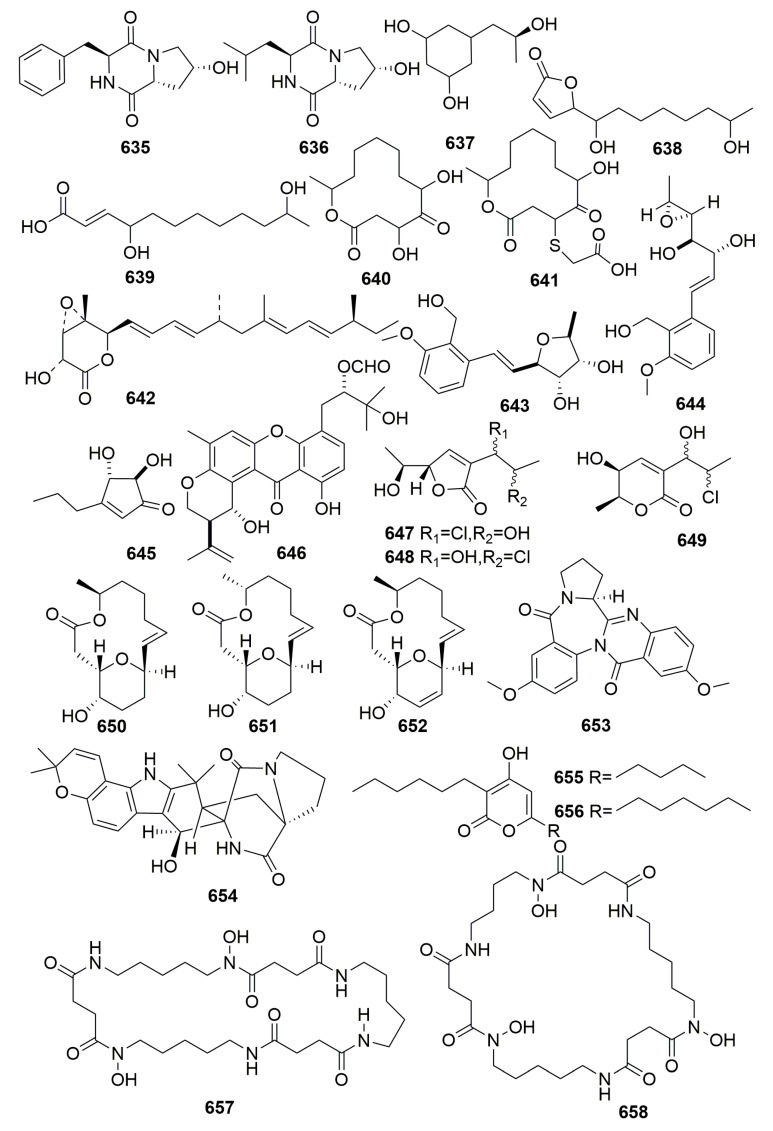
Structures of new molecules **635**–**658**.

**Figure 40 molecules-25-00853-f040:**
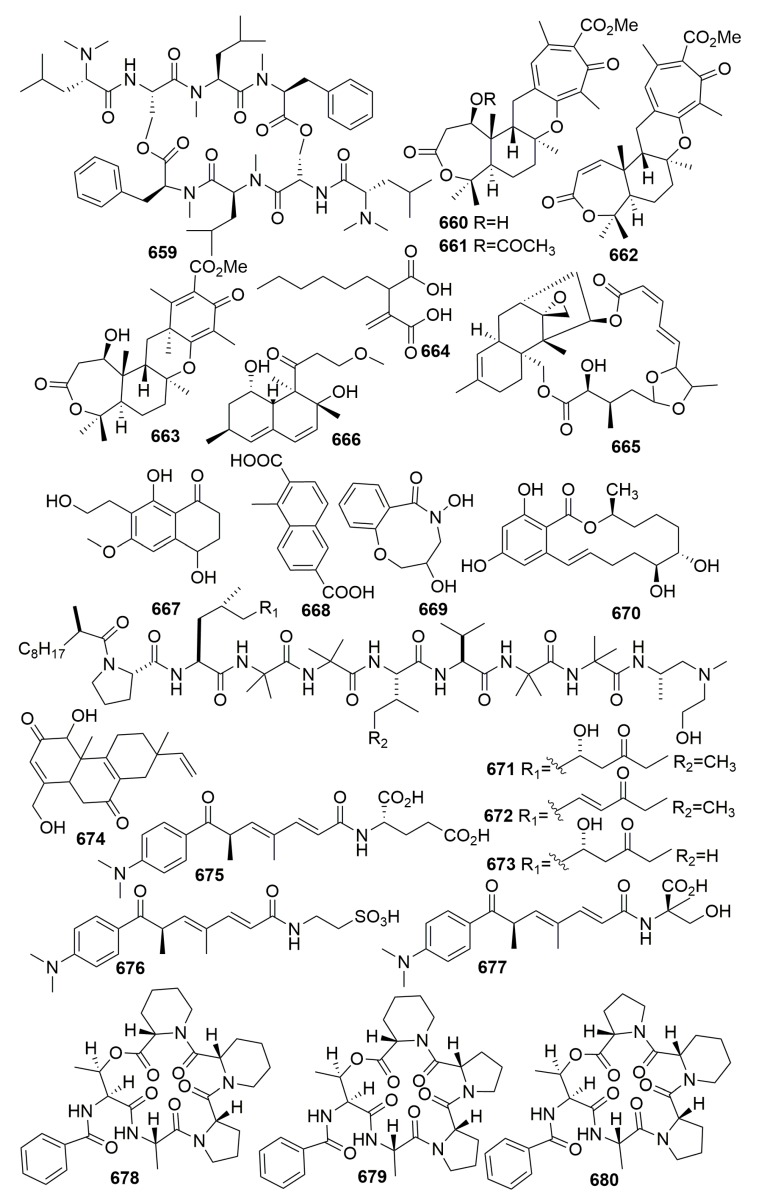
Structures of new compounds **659**–**680**.

**Figure 41 molecules-25-00853-f041:**
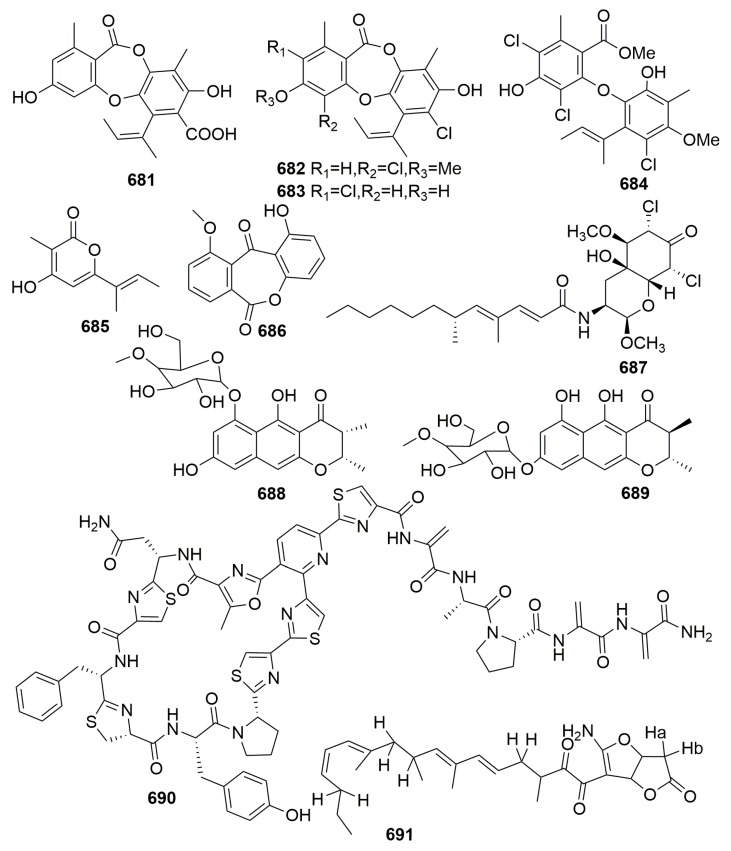
Structures of new marine natural products **681**–**691** derived from sponge (unidentified)-associated microbes.

**Figure 42 molecules-25-00853-f042:**
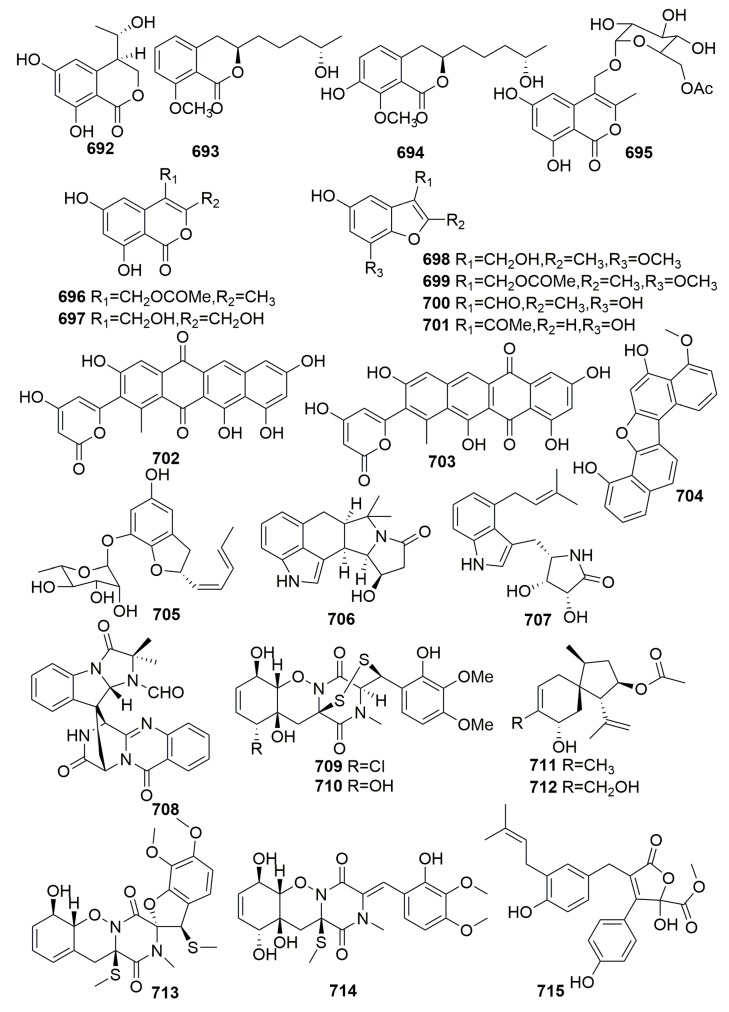
Chemical structures of new marine natural products **692**–**715** derived from sponge-derived microbes.

**Figure 43 molecules-25-00853-f043:**
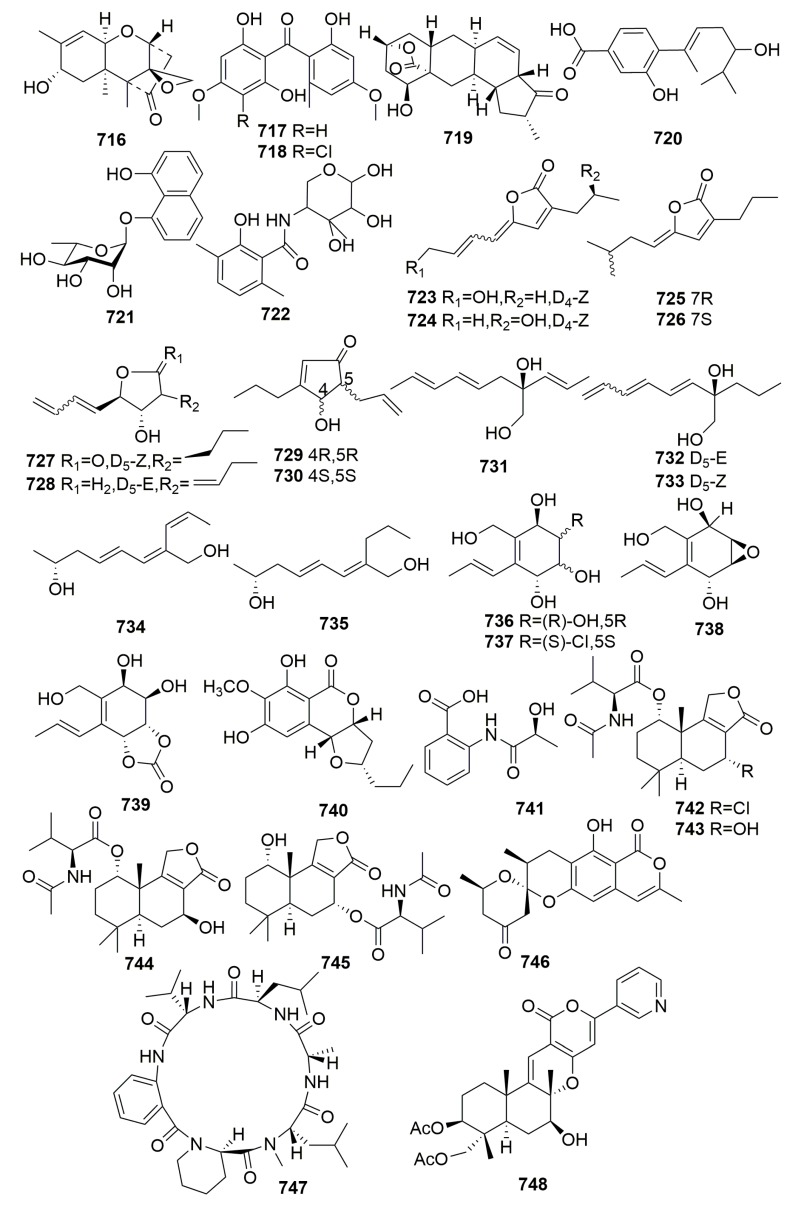
Structures of new compounds **716**–**748**.

**Figure 44 molecules-25-00853-f044:**
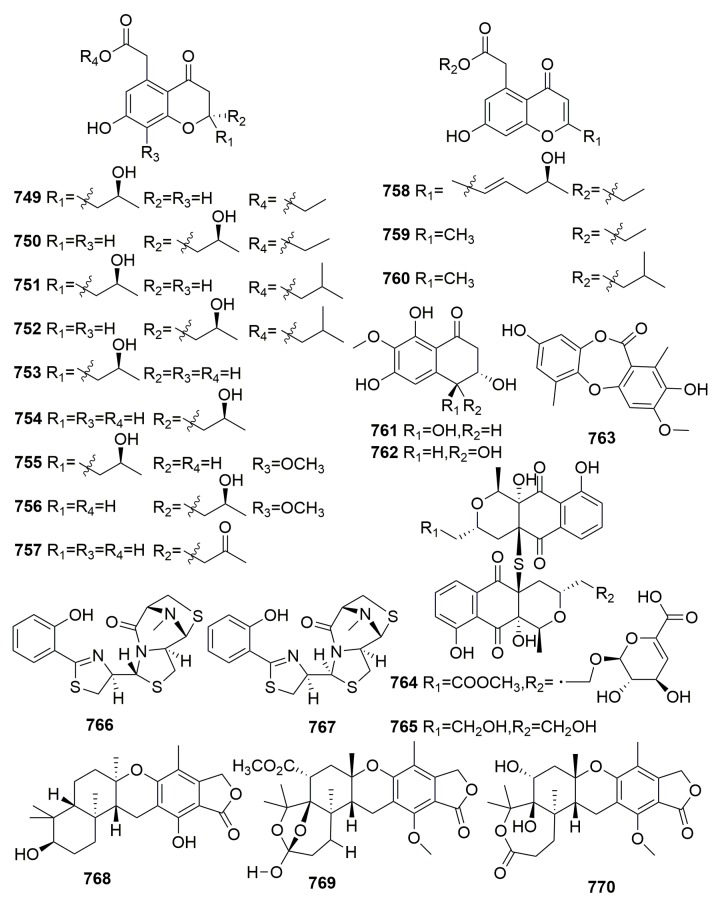
Structures of new molecules **749**–**770**.

**Figure 45 molecules-25-00853-f045:**
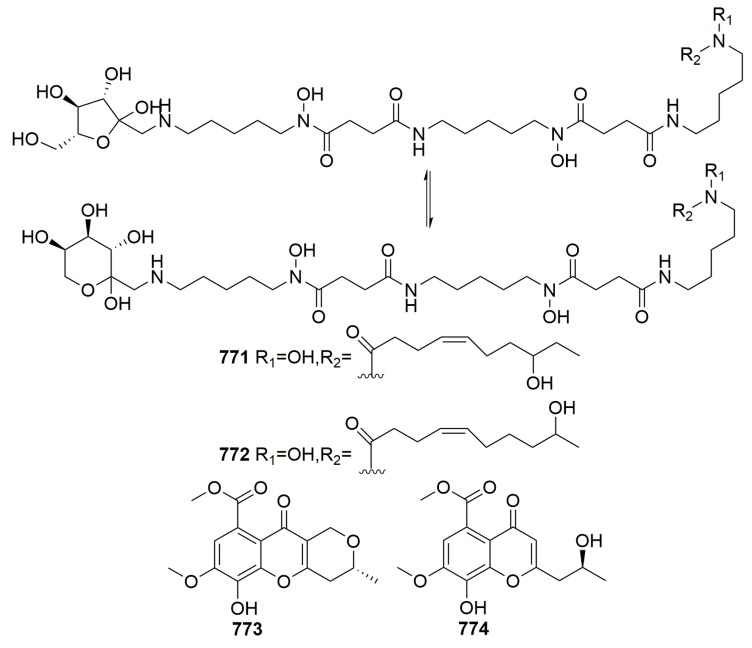
Structures of new marine natural products **771**–**774** derived from microbes, which were associated with the unidentified sponge.

**Figure 46 molecules-25-00853-f046:**
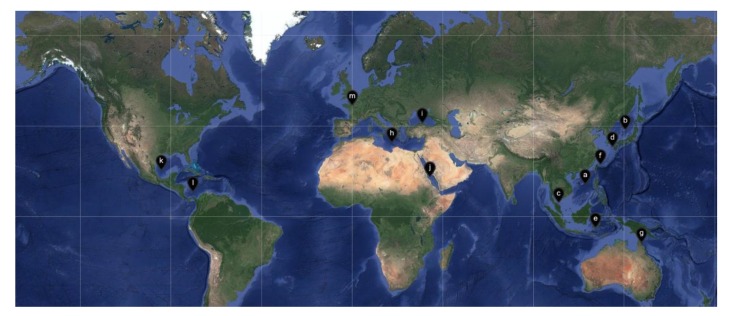
Geographical hotspot distribution for research on sponge-derived microorganisms. The red circles represent the hotspots of research: (**a**) South China Sea; (**b**) Sea of Japan; (**c**) Gulf of Thailand; (**d**) Korean Peninsula; (**e**) Indonesian Islands; (**f**) Eastern China Sea; (**g**) Great Barrier Reef, Australia; (**h**) the Mediterranean Sea; (**i**) the Black Sea; (**j**) the Red Sea; (**k**) the Gulf of Mexico; (**l**) the Caribbean Island; (**m**) the North Sea.

**Figure 47 molecules-25-00853-f047:**
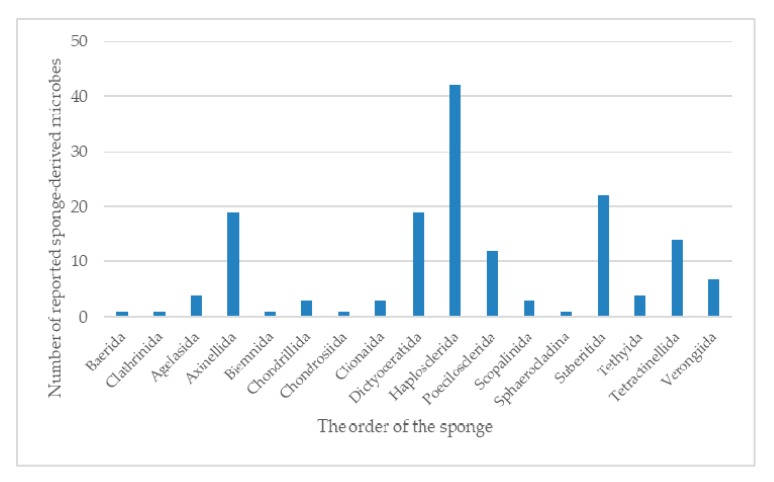
Distribution of research on sponges from which the microbes are derived.

**Figure 48 molecules-25-00853-f048:**
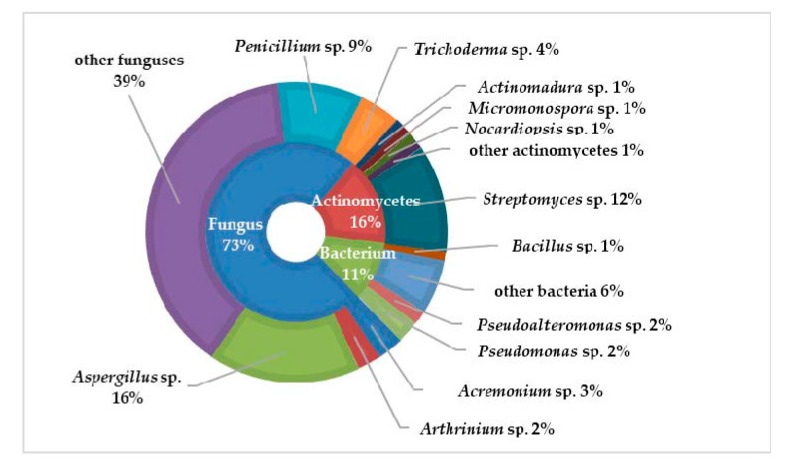
Percentage distribution of studies on sponge-derived microorganisms.

**Figure 49 molecules-25-00853-f049:**
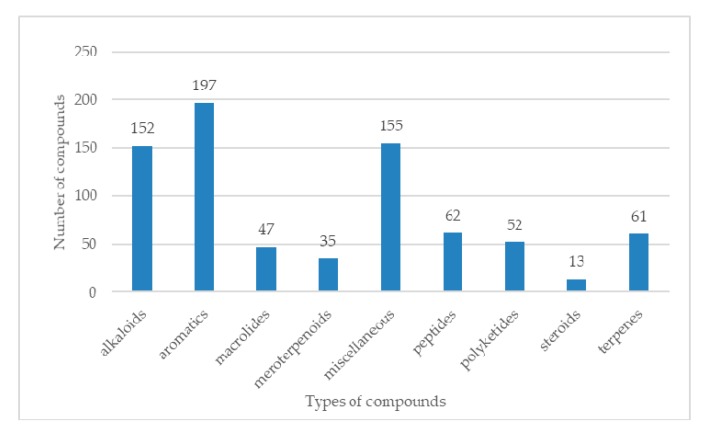
Distribution and activity analysis of natural products from sponge-derived microorganisms based on their putative biogenetic origin.

**Figure 50 molecules-25-00853-f050:**
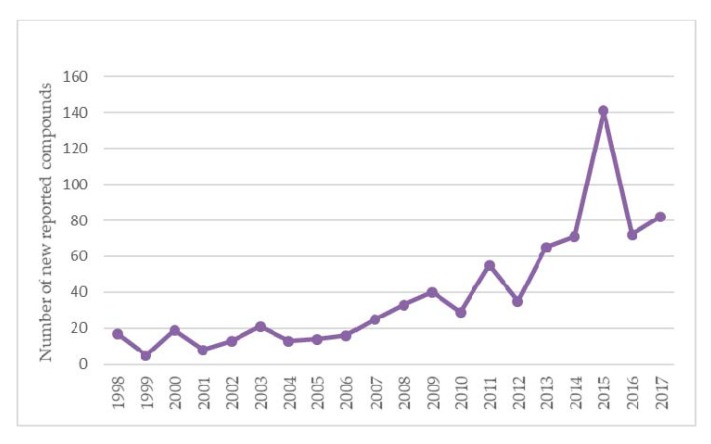
New compounds from sponge-associated microbes in the past two decades.

**Table 1 molecules-25-00853-t001:** Peptaibols (**30**–**37**) isolated from Trichoderma atroviride (NF16). Positions marked in gray differ between compounds.

Title	30	31	32	33	34	35	36	37
1	AcAib	AcAib	AcAib	AcAib	AcAib	AcAib	AcAib	AcAib
2	Ala	Ala	Ala	Ala	Ala	Ala	Ala	Ala
3	Ala	Ala	Ala	Ala	Ala	Ala	Ala	Ala
4	Aib	Aib	Aib	Aib	Aib	Aib	Aib	Aib
5	Iva	Aib	Aib	Aib	Iva	Iva	Aib	Iva
6	Gln	Gln	Gln	Gln	Gln	Gln	Gln	Gln
7	Aib	Aib	Aib	Aib	Aib	Aib	Aib	Aib
8	Aib	Aib	Aib	Aib	Aib	Aib	Aib	Aib
9	Aib	Ala	Ala	Ala	Aib	Ala	Aib	Ala
10	Ser	Ser	Ser	Ser	Ser	Ser	Ser	Ser
11	Leu	Leu	Leu	Leu	Leu	Leu	Leu	Leu
12	Aib	Aib	Aib	Aib	Aib	Aib	Aib	Aib
13	Pro	Pro	Pro	Pro	Pro	Pro	Pro	Pro
14	Leu	Leu	Val	Val	Val	Val	Leu	Val
15	Aib	Aib	Aib	Aib	Aib	Aib	Aib	Aib
16	Ile	Ile	Ile	Ile	Ile	Ile	Ile	Ile
17	Glu	Gln	Gln	Glu-OMe	Glu-OMe	Glu-OMe	Glu-OMe	Gln
18	Gln	Gln	Gln	Gln	Gln	Gln	Gln	Gln
19	Pheol	Pheol	Pheol	Pheol	Pheol	Pheol	Pheol	Pheol

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
