# Peer review of "Biological and Chemical Diversity of Marine Sponge-Derived Microorganisms over the Last Two Decades from 1998 to 2017"

_molecules, 2020, doi:10.3390/molecules25040853_

Round 1

Reviewer 1 Report

Comments to the author:

The present work titled" Biological and Chemical Diversity of Marine Sponge-

Derived Microorganisms over the last two 3 Decades from 1998 to 2017" seems incomplete and not will presented for the following reasons.

It is a listing work without clear objective(s). Classification should be based on chemical structures. Headings and sub-headings are not consistent. Chemical structure's fonts and size are not unified Figure legends are not well presented (use the classical style). No future perspectives. Most of the statistical work exhibits no real benefits. References style should be according to the journal style

Reviewer 2 Report

This review paper covers the research results regarding the metabolites of microorganisms isolated from marine sponges. The information on the metabolites, including structure, biological activity and biological source of producing strain is summarized well. The recommendation is to publish this review after overall review of grammar, expressions, and typos. Some examples are provided below.

English proof reading is needed. For example, the word ‘dissapointed’ is repeatedly used where ‘dissapointing’ should be used. (P16-L311, P22- L398) Some structures should be modified. For example, R should be added in the structure of compounds 47 and 48. The structure of compound 191 is missing an oxygen. Some expressions are not used appropriately. ‘butenolide-resorcylide dimers’ should modified as ‘butenolide-resorcylide conjugates’ (P5-L145), and ‘sulfur-bearing’ would be better to be modified as ‘sulfur-containing’ or ‘thiolane-bearing’ (P5-L150). ‘bridge-cyclic’ should be modified as ‘bridged cyclic’ (P16-L287), and ‘high’ should be modified as ‘higher’ (P16-L290). Errors in spacing or font, such as ‘aninseparable’ (P15-L3) or ‘aIle’ (a should be italic, P15 Figure 12)) are observed. ‘entgloeosteretriol’ also should be modified as ‘ent-gloeosteretriol’ (P20-L362).

Author Response

Response to Reviewer 2 Comments

Point 1: English proof reading is needed. For example, the word ‘dissapointed’ is repeatedly used where ‘dissapointing’ should be used. (P16-L311, P22- L398) 

Response 1: We have rechecked our manuscript carefully and corrected the text. For example, the word ‘disappointed’ has been replaced by ‘disappointing’. (P3-L106, P20-L317, P26-L404, P31-L473, P35-L534, P36-L561, P51-L804, P56-L882, P64-L990)

Point 2: Some structures should be modified. For example, R should be added in the structure of compounds 47 and 48. The structure of compound 191 is missing an oxygen.

Response 2: We have modified the structures. For example, I have added R in the structures of compounds 47 and 48, and I have added an oxygen in the structure of compound 191.

Point 3: Some expressions are not used appropriately. ‘butenolide-resorcylide dimers’ should modified as ‘butenolide-resorcylide conjugates’ (P5-L145), and ‘sulfur-bearing’ would be better to be modified as ‘sulfur-containing’ or ‘thiolane-bearing’ (P5-L150). ‘bridge-cyclic’ should be modified as ‘bridged cyclic’ (P16-L287), and ‘high’ should be modified as ‘higher’ (P16-L290).

Response 3: We have modified the expressions in the review. We have replaced ‘butenolide-resorcylide dimers’ by ‘butenolide-resorcylide conjugates’ (P6-L153), ‘sulfur-bearing’ by ‘sulfur-containing’ (P6-L159), ‘bridge-cyclic’ by ‘bridged cyclic’ (P19-L294), and replaced ‘high’ by ‘higher’ (P19-L297).

Point 4: Errors in spacing or font, such as ‘aninseparable’ (P15-L3) or ‘aIle’ (a should be italic, P15 Figure 12)) are observed. ‘entgloeosteretriol’ also should be modified as ‘ent-gloeosteretriol’ (P20-L362).

Response 4: We have modified the errors in spacing or font, such as ‘an inseparable’ (P19-L286) and ‘aIle’ (P19 Figure 12). I have replaced ‘entgloeosteretriol’ by ‘ent-gloeosteretriol’ (P24-L368).

achment.

Reviewer 3 Report

This work mainly summarizes all new substances from marine sponge-derived microbes discovered the past 20 years from 1998 to 2017 as well as their bioactive properties on the basis of broadly extensive literature search. Generally, this review is highly interesting and important to marine natural product chemists and pharmacologists around the world. However, the authors should pay attention to the following items and revise and improve them:

1-The introduction should be completely revised by adding the composition structure, including the number of class, order and family of marine sponges.

2-All chemical structures of secondary metabolites should be carefully drawn in the same size using ACS 1996 template of ChemOffice. More attention should be payed on their stereo configurations. 

3-As far as the original sources of natural products, the relationship between symbiotic microorganisms and their hosts should be elucidated.

4-The references should be written in the uniform format.

In addition, the English writing should be improved by a native speaker.

Reviewer 4 Report

This is an interesting review whereby the authors present around 774 compounds from microorganisms associated with sponges. While most of the review is rather encyclopedic, the last section of the review was the most informative and presents a nice compendium of the state of the art in the field. There are some corrections and/or suggestions that need to be taken into consideration before this is published, which are outlined below.

Line 102. “28 and 29 exhibited cytotoxic activity…” Do not start sentences with numbers, use names. This happens many times throughout the manuscript (e.g., lines 130, 347, 412, 532). Taking line 102 as an illustrative example, it better reads as: “Compounds 28 and 29 exhibited cytotoxicity…”

Line 158. The Figure titles should read as: “Figures3 and4. Chemical structures of diverse new molecules 49-88. Apply this to all the Figure titles in the review (e.g. line 186).

Line 208. It should read better as: “The antibacterial activities of 138 and 139 were evaluated against eight human pathogenic bacteria [49].”

Line 238. It should be: Pseudomonas alteromonas. The second name should not have a capital letter.

Line 279. An inseparable mixture… Separate words.

Line 311. “But their bioassay was disappointed [78,79].” This sentence is not clear. What is a disappointing bioassay? Please explain and notice the word disappointing, not disappointed. The same happens in line 878. Please define.

Line 364.“Compounds 264 and 266 were found to be anti-microbial active against Bacillus subtilus [94].” This sentence should read: “Compounds 264 and 266were antimicrobial towards Bacillus subtilus [94].” In addition, should it be Bacillus subtilisinstead of Bacillus subtilus???

Lines 498-500. “Compounds (401-403) exhibited potent anti-Plasmodium palcifarum K1 strain with IC50values in a range of 0.0534-2.93 μg/ml”. It will read better as: “Compounds 401-403 exhibited potent antiplasmodial activity towards the P. falciparumK1 strain..” Notice falciparum, not palcifarum!

Line 517. “…with a Fijian Zyzzya sp. sponge.”

Line 574. “…with an IC50value of 0.73 mM…”

Lines 717-718. “…of antibacterial activity with an MIC values ranged from 0.01 to 0.1 μg/ml [187].” This should read as: “… with MIC values between 0.01 and 0.1 mg/ml…”

Line 791. “…are illustrated in Figures 36 and 37.”

Line 815. “…were isolated…” Isolated was misspelled.

Lines 883-4. “Compounds 678-680 did not do well in their bioassays…”  What does it mean not to do well in a bioassay? Please explain since this is not clear!

Line 940. “Derivative 715 did not show antiviral…”

Line 1028 and beyond. Here Figures are abbreviated as Fig. (e.g., lines 1036, 1045) and in the beginning of the review they were designated as Figures. Be consistent here and follow journal styles.

In Figure 49 a series of natural compounds from sponge microorganisms were classified as either active and/or inactive. What was the criteria behind this classification? It was not clear how that distinction was made. What is an active compound and what is an inactive compound?

References style should be revised since in some cases abbreviations were used (e.g., J. Nat. Prod., line 1662) and in other cases not (e.g., Journal of Natural Products, line 1653). Check all references!

Round 2

Reviewer 1 Report

The authors did not provide responses to the previous comments, especially classification based on chemical structures, and future vision. Hence, more revision is required.

Author Response

Response to Reviewer 1 Comments

Point 1: The authors did not provide responses to the previous comments, especially classification based on chemical structures, and future vision. Hence, more revision is required.

Response 1: Thank you very much for considering our manuscript (molecules-706217). Firstly, taxonomy of such reviews can be based on not only the chemical structures, but also the sponge’s taxonomy (such as, Thomas, T. R.; Kavlekar, D. P.; LokaBharathi, P. A. Marine drugs from sponge-microbe association--a review. Mar. Drugs 2010, 8, (4), 1417-68). We focus on the relationship between sponges and symbiotic microbes, so we set the sponge’s taxonomy as the classification. In this way, it will be convenient for research scholars to find the microbes and their new derivatives derived-from the sponges. To be more consistent, we have modified the Headings as “Sponges and Derived Microbes’ Chemical Diversity” (Line 65). Secondly, the relationship between sponges and symbiotic microbes has not yet been elucidated, so we have talked about the future vision in the conclusion concisely (Lines 1101-8). The study on the relationship is an emerging, constantly-progressing, and very tortuous scientific research. If we can elucidate and utilize the relationship, it may solve the problem of drug source and promote the development of new drugs. So, it is very important and challenging to study the relationship between the sponges and associated microbes.